# DyCon: Dynamic Reasoning Control via Evolving Difficulty Modeling

Tengyao Tu [* 1 2]   Yulin Li [* 1]   Huiling Zhen [3]   Libo Qin [1]   Zhoujun Wei [4]   Jinghua Piao [2 5]   Zhuotao Tian [† 1 4]
Yong Li [2 5]   Min Zhang [1 4]

## Abstract

Recent advances in Large Reasoning Models (LRMs) demonstrate remarkable performance improvements by iteratively reflecting, exploring, and executing complex tasks, yet suffer from inefficiencies due to redundant reasoning, known as "overthinking". Existing methods to mitigate this issue either rely on static difficulty estimates or require task-specific training, and thus fail to adapt to the dynamic complexity during reasoning. In this work, we empirically show that the problem difficulty evolves dynamically throughout the reasoning process and is linearly encoded in the LRM's step-level embeddings. Building on this insight, we propose DYCON, a training-free framework that leverages latent step-level representations to explicitly model the evolving task difficulty, enabling the dynamic control of reasoning depth to mitigate the overthinking issue. Extensive experiments conducted on four models ranging from 4B to 32B, and across twelve benchmarks in math reasoning, general question answering, and coding tasks demonstrate that DYCON significantly enhances reasoning efficiency by reducing redundant steps without sacrificing accuracy or generalization. Project page and code are available at https://github.com/yu-lin-li/DyCon.

## 1. Introduction

Recent advances in Large Reasoning Models (LRMs) have shown strong performance on complex reasoning tasks such as mathematical problem-solving and code generation (Guo et al., 2025; Team, 2025; Yang et al., 2025a). These gains mainly arise from the models' ability to iteratively reflect, explore, and execute during reasoning (Chen

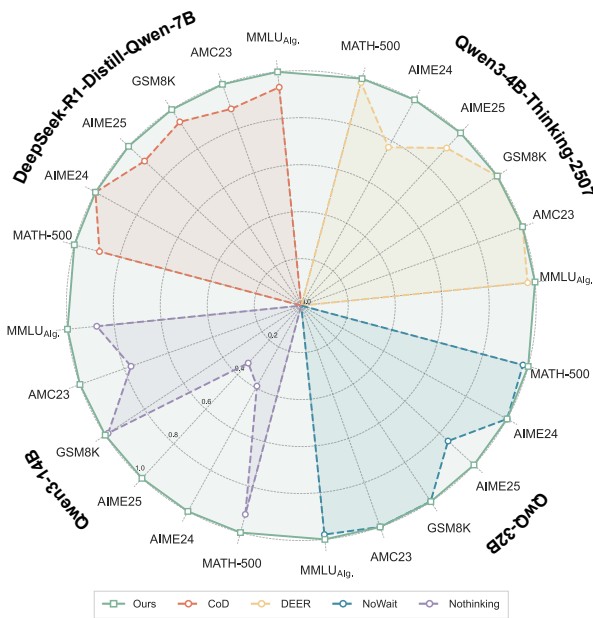

*Figure 1.* **Quantitative comparison.** Our method consistently outperforms prior approaches (Yang et al., 2025b; Wang et al., 2025a; Ma et al., 2025) across multiple mathematical reasoning benchmarks and four model architectures (4B–32B), while reducing token usage without sacrificing accuracy.

et al., 2025). However, existing work reveals that while Chain-of-Thought (CoT) reasoning (Wei et al., 2022) substantially boosts accuracy on difficult problems, current LRMs lack precise control over this mechanism. As a result, they often perform redundant reflection and exploration even on simple or already-solved tasks, a phenomenon termed "overthinking." (Chen et al., 2024) This inefficiency unnecessarily lengthens reasoning traces and can introduce additional hallucinations (Sun et al., 2025), posing a critical bottleneck for practical LRM deployment. Addressing overthinking essentially involves terminating reasoning once sufficient exploration has been achieved. Although several methods have been proposed to identify suitable termination points, they typically fall short in adapting effectively to varying problem difficulties. Specifically, TrimR (Lin et al., 2025a) and FlashThink (Jiang et al., 2025) rely on external models to assess reasoning sufficiency. However, these strategies apply uniform criteria across all inputs, ig-

---
[*]Equal contribution [1]Harbin Institute of Technology (Shenzhen) [2]Zhongguancun Academy [3]Huawei Noah's Ark Lab [4]Shenzhen Loop Area Institute [5]Tsinghua University. Correspondence to: [†] Zhuotao Tian <tianzhuotao@hit.edu.cn>.

*Proceedings of the 43rd International Conference on Machine Learning*, Seoul, South Korea. PMLR 306, 2026. Copyright 2026 by the author(s).

noring problem-specific difficulty and thus failing to adapt termination points accordingly. Alternative methods (Yang et al., 2025b; Fu et al., 2025) leverage handcrafted metrics to gauge the model's certainty and determine when to terminate reasoning. While intuitive, these methods depend heavily on human priors and empirical thresholds, limiting their generalizability across problems of varying complexity. Another direction (Zhang et al., 2025a; Lou et al., 2025; Huang et al., 2025c) employs Supervised Fine-Tuning (SFT) or Reinforcement Learning (RL) with specially curated datasets to train models to implicitly infer problem difficulty and decide where the reasoning process terminates. Despite their potential, such methods are sensitive to the quantity and quality of data and prone to mode collapse (Lou et al., 2025). Hence, a key question arises: *How can we explicitly model task difficulty to adaptively determine when to terminate or extend the reasoning process, thereby enhancing reasoning efficiency for simpler problems while ensuring comprehensive exploration for complex ones?*

**Key observations.** Though recent works (Sheng et al., 2025; Nguyen et al., 2025; Zhao et al., 2025) have attempted to estimate problem difficulty, they typically assign static difficulty scores before the reasoning process begins based on embeddings derived from the initial question or the `<think>` token. Consequently, these approaches are constrained to the sample-level estimation and fail to capture how difficulty dynamically evolves throughout the reasoning process itself.

However, as illustrated in Fig. 2(a), we observe that the problem difficulty is not static but evolves dynamically during reasoning. When the reasoning path remains valid, the difficulty gradually decreases as the CoT progressively decomposes and clarifies the problem. Conversely, if reasoning deviates, misleading or distracting CoT content causes difficulty to remain high or even increase. This observation motivates us to explore a fine-grained, step-level metric capable of explicitly modeling and accurately capturing the dynamic variations in problem difficulty during reasoning.

Furthermore, the results shown in Fig. 2(b) indicate that the step-level difficulty information in LRMs can be encoded within embeddings at each reasoning step, exhibiting a linear correlation with actual problem difficulty. This suggests that LRMs inherently possess latent knowledge regarding dynamically evolving difficulty in their embedding spaces. Inspired by this finding, we ask: *Can this latent knowledge be leveraged to adaptively assess difficulty, both across different samples and throughout the reasoning process, thereby facilitating more efficient reasoning?*

**Our Solution.** In this work, we introduce **DyCon**, a training-free, evolving difficulty-aware mechanism for efficient reasoning. DYCON leverages latent knowledge in

LRM representations to model both inter-sample and intra-reasoning difficulty dynamics. We fit a linear regressor on a small-scale seen dataset to map reasoning-step embeddings to problem difficulty. During inference, this regressor estimates difficulty at each reasoning step, capturing fine-grained complexity shifts. Guided by these estimates, DY-CON dynamically adjusts the logits for reflection keywords. If the estimated difficulty is low, indicating adequate reasoning, logits of reflection keywords are reduced to expedite convergence. Conversely, if the estimated difficulty is high, these logits are increased to encourage deeper reflection. This mechanism enables dynamic, latent knowledge-guided control over reasoning length, improving reasoning efficiency on simpler tasks without compromising exploration on complex ones.

Extensive experiments across four models ranging from 4B to 32B, and on twelve benchmarks covering math reasoning, general question answering, and coding tasks, demonstrate the effectiveness and strong generalization capabilities of DYCON. To summarize, our contributions are as follows:

- We empirically verify that problem difficulty in LRMs evolves dynamically during reasoning. Our analysis reveals a linear correlation between step embeddings and step-level difficulty, indicating that LRMs inherently possess latent knowledge capable of explicitly modeling this evolving difficulty.

- To achieve a dynamic control of the reasoning behavior, we propose DYCON, a training-free evolving difficulty-aware dynamic reasoning control mechanism. By employing a lightweight linear regressor to estimate difficulty from step embeddings, DYCON dynamically adjusts the logits of reflection-related keywords based on this latent knowledge, effectively balancing exploration and efficiency during reasoning.

- Extensive experiments across different models and tasks demonstrate that DYCON effectively reduces redundant reasoning without compromising accuracy, exhibiting its strong generalizability and robustness across varying problem complexities and domains.

## 2. Background and Motivation

### 2.1. Preliminaries

This study addresses the problem of efficient reasoning by explicitly modeling step-level difficulty, enabling adaptive adjustments in reasoning behavior to mitigate overthinking. In this section, we introduce the preliminaries required to elaborate on the motivation and details of our method.

**Inference of LRMs.** Given an input question $q$, a Large Reasoning Model (LRM) generates a sequence of tokens

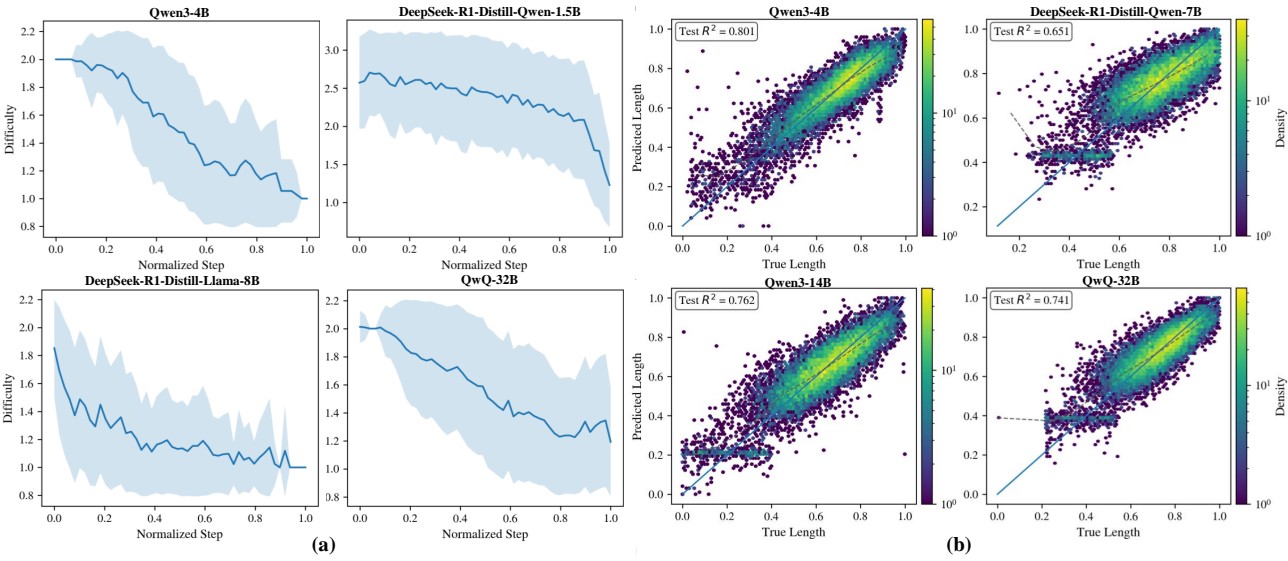

*Figure 2.* **Dynamic evolution and latent encoding of problem difficulty during reasoning.** (a) The dynamic evolution of self-assessed difficulty across normalized reasoning steps. The blue curves indicate mean difficulty ratings, while shaded areas represent standard deviations. Problem difficulty exhibits a consistent declining trend, confirming its dynamic nature throughout reasoning. (b) Linear regression predictions of normalized problem difficulty from step embeddings. With remaining reasoning length as the proxy for evolving difficulty, predictions closely match actual difficulty with high R² scores (*i.e.*, the coefficient of determination in statistics), demonstrating a strong linear relationship and confirming that step embeddings encode latent difficulty knowledge.

$\mathbf{y} = (y_1, \ldots, y_T)$ autoregressively:

$$p_\theta(\mathbf{y} \mid q) = \prod_{t=1}^{T} p_\theta(y_t \mid q, y_{<t}), \quad (1)$$

where $p_\theta(y_t \mid q, y_{<t}) = \text{softmax}(\mathbf{z}_t)$ and $\mathbf{z}_t \in \mathbb{R}^{|\mathcal{V}|}$ denotes the pre-softmax logit vector over the vocabulary $\mathcal{V}$ at decoding step $t$. Let $z_{t,i}$ be the logit of token $i \in \mathcal{V}$ at step $t$. The average logit at step $t$ is given by:

$$\mu_t = \frac{1}{|\mathcal{V}|} \sum_{i \in \mathcal{V}} z_{t,i}. \quad (2)$$

Our study focuses on the reasoning part of the output, which is enclosed between the tokens <think> and </think>. Following (Wang et al., 2025a), we consider each occurrence of \n\n as the boundary between steps. $t_s$ and $t_{\text{end}}$ denote the token indexes of the $s$-th step boundary and the ending token </think>, respectively.

**Representations of reasoning steps.** To enable fine-grained control over reasoning behavior, we investigate the latent representations of individual reasoning steps. Consider an LRM consisting of $L$ layers, where the $d$-dimensional hidden state at layer $\ell$ and token position $t$ is denoted as $\mathbf{h}_t^{(\ell)} \in \mathbb{R}^d$. Due to the causal attention mask employed during decoding, the hidden state $\mathbf{h}_{t_s}^{(\ell)}$ at each step boundary (*i.e.*, \n\n) inherently encodes contextual information from preceding steps (Chen et al., 2025). Therefore,

we define the step embedding $\mathbf{e}_s^{(\ell)}$ for the $s$-th reasoning step at layer $\ell$ as follows:

$$\mathbf{e}_s^{(\ell)} := \mathbf{h}_{t_s}^{(\ell)}. \quad (3)$$

**Proxy for estimating step-level difficulty.** Harder tasks require deeper exploration, while simpler tasks benefit from quicker convergence. Prior work typically uses overall reasoning length as a proxy for task difficulty (Sheng et al., 2025; Su et al., 2025b). However, difficulty often varies throughout the reasoning process, and different stages may present distinct challenges. Therefore, fine-grained control necessitates estimating difficulty at the step-level. To achieve this, we propose a step-level proxy defined at each step boundary:

$$r_s := t_{\text{end}} - t_s, \quad (4)$$

where $t_s$ denotes the index at the $s$-th step boundary (*i.e.*, \n\n) and $t_{\text{end}}$ is the index of the </think> token.

Intuitively, $r_s$ measures the remaining length from the current step boundary to the end of the reasoning trace. A larger $r_s$ indicates that substantial reasoning remains, suggesting a more challenging situation, while a smaller $r_s$ indicates that the reasoning process is closer to termination.

## 2.2. Key Observations

Existing efficient reasoning methods (Lin et al., 2025a; Yang et al., 2025b) focus on identifying optimal termination

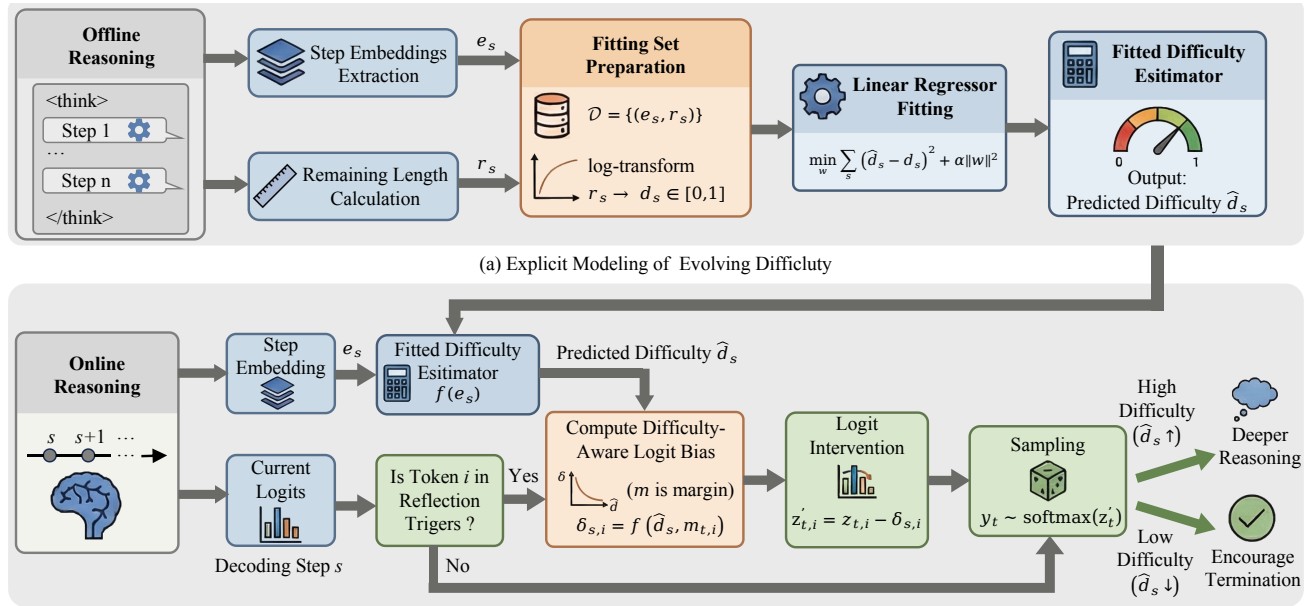

(a) Explicit Modeling of Evolving Difficulty

(b) Difficulty-Aware Dynamic Reasoning Control

*Figure 3.* **Overview of DyCon.** (a) Explicit Modeling of Evolving Difficulty: In offline reasoning, step embeddings are extracted from model outputs to construct a fitting set with remaining length information. These lengths are log-transformed and normalized, creating a bounded difficulty target used to fit a linear regressor as the difficulty estimator. (b) Difficulty-Aware Dynamic Reasoning Control: During online reasoning, this estimator dynamically predicts step-level difficulty, guiding logit interventions to reduce the probabilities of reflection-related tokens based on evolving difficulty. This adaptive mechanism promotes deeper reasoning when difficulties are high and encourages early termination in simpler scenarios, optimizing the reasoning depth effectively.

points to avoid unnecessary reasoning steps. These methods assume that problem difficulty remains static throughout reasoning (Sheng et al., 2025; Zhao et al., 2025). However, we observe that problem difficulty evolves dynamically during the reasoning process and find that large reasoning models (LRMs) inherently encode such evolving difficulty as latent knowledge within their internal representations. We detail our observations below.

**Difficulty evolves with the reasoning progress.** Theoretically, problem difficulty may decrease if the model follows a productive reasoning path, whereas ineffective paths could increase difficulty by introducing noise or confusion. To empirically validate this assumption, we conduct experiments on level 5 problems from the MATH-500 (Lightman et al., 2023) benchmark, which typically demand extended CoT and thus enable fine-grained analysis.

Specifically, after each reasoning step, we prompt the model to self-assess current difficulty on a 3-point scale: 1 (almost solved), 2 (some uncertainty remains), or 3 (missing key insight) (see Appendix D.6 for details). As shown in Fig. 2(a), the average self-assessed difficulty, normalized and aggregated across all samples, displays a clear decreasing trend with fluctuations. Notably, this phenomenon consistently emerges across four distinct model families (1.5B–32B parameters). Consequently, accurate identification of termi-

nation points requires careful monitoring of difficulty evolution. Practical exploitation of this phenomenon for reasoning control thus requires explicit, fine-grained difficulty modeling.

**Latent knowledge encoded in step embeddings.** Prior studies (Su et al., 2025a) suggest that internal reasoning states are reflected in hidden states of LRMs. We hypothesize that step embeddings similarly encode latent difficulty knowledge.

To investigate this, we take the remaining reasoning length as a difficulty proxy (Sec. 2.1), and sample 600 samples from the MATH (Hendrycks et al., 2021) training set, fitting a linear regressor to predict normalized difficulty based on corresponding step embeddings (detailed in Sec. 3.2). As illustrated in Fig. 2(b), predictions from the fitted regressor closely match the actual difficulty values across a held-out, unseen test set and three distinct model families ranging from 4B to 32B. The consistently high $R^2$ scores (*i.e.*, the coefficient of determination in statistics) indicate that step embeddings effectively capture latent difficulty information, exhibiting a nearly linear relationship.

Consequently, the linear relationship between step embeddings and problem difficulty offers an effective foundation for explicit, fine-grained modeling of difficulty evolution. Leveraging this latent knowledge enables computationally

efficient difficulty estimation, thus facilitating dynamic control over model reasoning behavior.

## 3. Method

### 3.1. Overview

In this section, we introduce DYCON, a dynamic reasoning control mechanism guided by evolving difficulty estimation. Inspired by the observations described in Sec. 2.2, DYCON consists of two steps: (i) explicitly modeling step-level difficulty that evolves throughout the reasoning trajectory by leveraging latent knowledge captured within the hidden representations of the LRM (Sec. 3.2); and (ii) dynamically adjusting the reasoning behavior based on estimated difficulty, thereby mitigating unnecessary exploration once sufficient reasoning depth has been achieved. (Sec. 3.3).

### 3.2. Explicit Modeling of Evolving Difficulty

As discussed in Sec. 2.2, step embeddings naturally encode evolving difficulty information. Therefore, DYCON introduces a lightweight difficulty estimator that maps hidden step embeddings directly to step-level difficulty. Crucially, DYCON does not alter the original LRM parameters $\theta$; instead, we fit a simple linear regressor on a small-scale seen dataset to decode the latent difficulty signals inherently captured by the model.

**From remaining length to evolving difficulty.** We randomly sample 600 instances from the MATH (Hendrycks et al., 2021) training set. For each instance, we run the LRM to generate its Chain-of-Thought (CoT) output enclosed by `<think>···</think>`. Following the definitions in Sec. 2.1, at each step boundary (*i.e.*, \n\n), we record: (i) the step embedding $\mathbf{e}_s$, and (ii) the corresponding remaining length $r_s$, forming a step-level fitting set:

$$\mathcal{D} = \left\{ \left( \mathbf{e}_s, r_s \right) \right\}. \quad (5)$$

However, directly using $r_s$ as a regression target may be suboptimal because it typically exhibits a heavy-tailed distribution: a small number of steps can have extremely large remaining lengths, disproportionately influencing the regression (see Tab. 8). To mitigate this, we first apply a log-transform to compress the scale of remaining lengths, followed by normalization to derive a bounded difficulty target $d_s$ for fitting:

$$\tilde{r}_s = \ln(1 + r_s), \qquad d_s = \frac{\tilde{r}_s - \tilde{r}_{\min}}{\tilde{r}_{\max} - \tilde{r}_{\min}} \in [0, 1], \quad (6)$$

where $\tilde{r}_{\min}$ and $\tilde{r}_{\max}$ are computed over the fitting set. By construction, a larger $d_s$ corresponds to a more difficult reasoning step (indicating more reasoning remains), whereas a smaller $d_s$ indicates an easier step.

**Linear decoding of latent difficulty knowledge.** To leverage the linear encoding of evolving difficulty within the step embeddings (Sec. 2.2), we fit a ridge regressor to estimate the difficulty based on the step embeddings. Specifically, with $\mathbf{e}_s \in \mathbb{R}^d$ denoting the extracted embedding of step $s$, we can model the normalized step difficulty $d_s$ via a linear decoder that yields estimated difficulty $\hat{d}_s$:

$$\hat{d}_s = f(\mathbf{e}_s) = \mathbf{w}^\top \mathbf{e}_s + b, \quad (7)$$

The learnable parameters $\mathbf{w} \in \mathbb{R}^d$ and $b \in \mathbb{R}$ are optimized via ridge regression:

$$\min_{\mathbf{w}, b} \sum_{(\mathbf{e}_s, d_s) \in \mathcal{D}} \left( \hat{d}_s - d_s \right)^2 + \alpha \|\mathbf{w}\|_2^2, \quad (8)$$

where $\alpha \geq 0$ controls the strength of $\ell_2$ regularization. We note that we extract embeddings from a specific layer of the model, and both the embedding layer and the ridge regularization weight $\alpha$ are determined automatically by maximizing the $R^2$ score on a held-out validation set without manual tuning (see Appendix A.4 for more details).

**Test-time difficulty estimation.** During test-time reasoning, whenever a new step boundary is generated, we compute its step embedding $\mathbf{e}_s$ and estimate the difficulty $\hat{d}_s = f(\mathbf{e}_s)$, which tracks the evolution of difficulty along the reasoning trajectory, enabling dynamic, difficulty-aware reasoning control.

### 3.3. Difficulty-Aware Dynamic Reasoning Control

With the step-level estimated difficulty $\hat{d}_s$ available during inference, DYCON dynamically controls the LRM's reasoning behavior to mitigate overthinking. The control follows a simple yet effective principle: for steps identified as low-difficulty, the model is encouraged to terminate the reasoning; conversely, for steps assessed as high-difficulty, the model's reasoning capacity should be preserved for deeper reflection and exploration.

Existing works (Wang et al., 2025a) terminate reasoning by suppressing the probabilities of reflection keywords. Inspired by them, to achieve difficulty-aware dynamic reasoning control, we propose reducing the token logits of the reflection keywords based on estimated difficulties. Specifically, we define a set $\mathcal{S} \subset \mathcal{V}$ of token IDs corresponding to reflection-related keywords, as detailed in Appendix D.2. Then, at each decoding step $s$, we compute a difficulty-conditioned logit bias for each $i \in \mathcal{S}$, and subtract the bias from the logits of reflection-triggers, *i.e.*, the tokens belonging to the reflection keywords:

$$z'_{t,i} = \begin{cases} z_{t,i} - \delta_{s,i}, & i \in \mathcal{S}, \\ z_{t,i}, & \text{otherwise}, \end{cases} \quad (9)$$

*Table 1.* **Performance on math reasoning benchmarks.** Following prior work (Jaech et al., 2024; Guo et al., 2025), we evaluate our method on small-scale benchmarks using multiple independent sampling trials to assess stability; detailed results are provided in Appendix B.5. Since TrimR (Lin et al., 2025a), FlashThink (Jiang et al., 2025), and ThinkPilot (Li et al., 2025a) are not publicly released, we re-implemented these methods based on their published descriptions.

| Method | MATH-500 | | AIME24 | | AIME25 | | GSM8K | | AMC23 | | MMLU$_{algebra}$ | |
| --- | --- | --- | --- | --- | --- | --- | --- | --- | --- | --- | --- | --- |
| | Pass@1↑ | #Tok↓ | Pass@1↑ | #Tok↓ | Pass@1↑ | #Tok↓ | Pass@1↑ | #Tok↓ | Pass@1↑ | #Tok↓ | Pass@1↑ | #Tok↓ |
| **DeepSeek-R1-Distill-Qwen-7B** | | | | | | | | | | | | |
| Baseline (Guo et al., 2025) | 92.0 | 3955 | 50.0 | 13008 | 36.7 | 15245 | 90.6 | 1214 | 87.5 | 6193 | 90.0 | 2387 |
| CoD (Xu et al., 2025a) | 81.8 | 1976 | 53.3 | 11419 | 33.3 | 14333 | 85.4 | 301 | 80.0 | 4810 | 85.0 | 1091 |
| Nothinking (Ma et al., 2025) | 80.0 | 1020 | 16.7 | 4222 | 23.3 | 4385 | 82.1 | 242 | 72.5 | 1141 | 74.0 | 760 |
| Thinkpilot (Li et al., 2025a) | 78.0 | 715 | 13.3 | 1229 | 10.0 | 1961 | 86.7 | 327 | 60.0 | 1042 | 74.0 | 705 |
| DEER (Yang et al., 2025b) | 89.8 | 2143 | 49.2 | 9839 | 36.7 | 7257 | 90.6 | 917 | 85.0 | 4451 | 79.0 | 1493 |
| SEAL (Chen et al., 2025) | 91.6 | 2943 | 43.3 | 11092 | 26.7 | 11092 | 88.8 | 889 | 77.5 | 5267 | 80.0 | 1507 |
| Manifold Steering (Huang et al., 2025d) | 88.4 | 2239 | 53.3 | 8457 | – | – | 87.6 | 440 | 87.5 | 4440 | – | – |
| Controlling Thinking Speed (Lin et al., 2025b) | 90.0 | 2818 | 50.0 | 12588 | 40.0 | 10997 | 86.4 | 478 | 82.5 | 5433 | 90.0 | 1719 |
| NoWait (Wang et al., 2025a) | 89.6 | 2702 | 40.0 | 7281 | 26.7 | 9302 | 89.1 | 794 | 85.0 | 4376 | 89.0 | 1347 |
| Ours | 92.0 | 3216 | 53.3 | 10906 | 36.7 | 12415 | 91.1 | 880 | 90.0 | 3801 | 91.0 | 1488 |
| Δ vs. Baseline | (+0.0) | (-18.7%) | (+3.3) | (-16.2%) | (+0.0) | (-18.6%) | (+0.5) | (-27.5%) | (+2.5) | (-38.6%) | (+1.0) | (-37.7%) |
| **Qwen3-4B-Thinking-2507** | | | | | | | | | | | | |
| Baseline (Yang et al., 2025a) | 96.2 | 6749 | 83.3 | 21493 | 76.7 | 22708 | 95.9 | 1494 | 100 | 11073 | 94.0 | 3496 |
| CoD (Xu et al., 2025a) | 95.6 | 4484 | 83.3 | 18652 | 80.0 | 21246 | 95.7 | 952 | 100 | 8973 | 95.0 | 3209 |
| Thinkpilot (Li et al., 2025a) | 88.6 | 2911 | 43.3 | 7913 | 30.0 | 8814 | 94.7 | 878 | 75.0 | 5085 | 83.0 | 1306 |
| Nothinking (Ma et al., 2025) | 95.2 | 4362 | 73.3 | 16556 | 73.3 | 19177 | 95.0 | 1137 | 97.5 | 7738 | 94.0 | 2331 |
| DEER (Yang et al., 2025b) | 94.6 | 5508 | 66.7 | 12728 | 70.0 | 13342 | 95.7 | 1037 | 100 | 9521 | 92.0 | 1945 |
| NoWait (Wang et al., 2025a) | 92.6 | 5062 | 53.3 | 12393 | 53.3 | 13322 | 94.8 | 1070 | 92.5 | 8204 | 95.0 | 2068 |
| Ours | 96.2 | 6092 | 86.7 | 18867 | 76.7 | 21100 | 95.7 | 1098 | 100 | 9162 | 95.0 | 2122 |
| Δ vs. Baseline | (+0.0) | (-9.7%) | (+3.4) | (-12.2%) | (+0.0) | (-7.1%) | (-0.2) | (-26.5%) | (+0.0) | (-17.3%) | (+1.0) | (-39.3%) |
| **QwQ-32B** | | | | | | | | | | | | |
| Baseline (Team, 2025) | 96.0 | 4267 | 73.3 | 13364 | 60.0 | 16462 | 96.8 | 1505 | 97.5 | 7166 | 95.0 | 2133 |
| CoD (Xu et al., 2025a) | 94.8 | 3662 | 63.3 | 11029 | 46.7 | 13289 | 96.5 | 617 | 92.5 | 6321 | 97.0 | 1345 |
| Nothinking (Ma et al., 2025) | 95.6 | 3989 | 66.7 | 11507 | 70.0 | 15312 | 96.5 | 1331 | 97.5 | 7472 | 96.0 | 1431 |
| DEER (Yang et al., 2025b) | 94.6 | 3316 | 70.0 | 10087 | 50.0 | 11598 | 96.3 | 977 | 95.0 | 5782 | 96.0 | 1395 |
| FlashThink (Jiang et al., 2025) | 93.2 | 3144 | 60.0 | 10034 | 40.0 | 11861 | 96.5 | 910 | 92.5 | 6702 | – | – |
| TrimR (Lin et al., 2025a) | 93.8 | 3830 | 56.7 | 8345 | 43.3 | 8827 | 93.7 | 1319 | 90.0 | 6055 | – | – |
| SEAL (Chen et al., 2025) | 93.0 | 3667 | 63.3 | 12064 | 56.7 | 12089 | 96.3 | 1231 | 97.5 | 6448 | 95.0 | 1541 |
| NoWait (Wang et al., 2025a) | 93.6 | 2902 | 73.3 | 9405 | 56.7 | 11871 | 96.7 | 983 | 100.0 | 4536 | 95.0 | 1302 |
| Ours | 95.8 | 3345 | 73.3 | 12794 | 66.7 | 13640 | 96.8 | 995 | 100 | 5654 | 97.0 | 1266 |
| Δ vs. Baseline | (-0.2) | (-21.6%) | (+0.0) | (-4.3%) | (+6.7) | (-17.1%) | (+0.0) | (-33.9%) | (+2.5) | (-21.1%) | (+2.0) | (-40.6%) |
| **Qwen3-14B** | | | | | | | | | | | | |
| Baseline (Yang et al., 2025a) | 95.0 | 4962 | 76.7 | 12746 | 70.0 | 16613 | 96.3 | 1693 | 97.5 | 6671 | 96.0 | 2545 |
| CoD (Xu et al., 2025a) | 93.8 | 3535 | 63.3 | 11426 | 46.7 | 12391 | 96.2 | 670 | 92.5 | 6371 | 94.0 | 1381 |
| Nothinking (Ma et al., 2025) | 87.4 | 940 | 30.0 | 5123 | 23.3 | 5115 | 94.9 | 260 | 75.0 | 1818 | 84.0 | 547 |
| Thinkpilot (Li et al., 2025a) | 86.8 | 854 | 26.7 | 7841 | 23.3 | 3488 | 94.9 | 274 | 72.5 | 1561 | 88.0 | 538 |
| Dynasor-CoT (Fu et al., 2025) | 93.8 | 4023 | 73.3 | 10369 | 60.0 | 12159 | 95.6 | 1483 | 95.0 | 6582 | 91.0 | 1733 |
| DEER (Yang et al., 2025b) | 94.0 | 3316 | 76.7 | 7619 | 66.7 | 11135 | 95.3 | 840 | 95.0 | 4763 | 87.0 | 1380 |
| NoWait (Wang et al., 2025a) | 94.6 | 3305 | 76.7 | 10181 | 60.0 | 12276 | 95.8 | 1125 | 97.5 | 4935 | 93.0 | 1729 |
| Ours | 95.0 | 3645 | 76.7 | 10536 | 70.0 | 14537 | 96.3 | 1166 | 97.5 | 5240 | 96.0 | 2073 |
| Δ vs. Baseline | (+0.0) | (-26.6%) | (+0.0) | (-17.3%) | (+0.0) | (-12.5%) | (+0.0) | (-31.1%) | (+0.0) | (-21.4%) | (+0.0) | (-18.5%) |

and sample the next token from the intervened distribution

$$y_t \sim \text{softmax}(\mathbf{z}'_t). \tag{10}$$

Different from prior studies (Lin et al., 2025a; Yang et al., 2025b), our strategy does not enforce termination. Instead, it dynamically reduces the probabilities of reflection triggers based on the reasoning depth, enabling fine-grained and adaptive control over the reasoning behavior of LRMs. Next, we need to derive the logit bias $\delta_{s,i}$ based on the estimated difficulty $\hat{d}_s$.

**Difficulty-aware logit bias.** To generate the difficulty-aware logit bias $\delta_t$, given the logits $\mathbf{z}_t$ of the $s$-th step boundary, we first compute the mean logit $\mu_t$ as in Eq. (2) and define the positive margin $m_{t,i}$ as

$$m_{t,i} := [z_{t,i} - \mu_t]_+ = \max(z_{t,i} - \mu_t, 0). \tag{11}$$

This formulation ensures the logit bias is applied only to reflection-triggers whose logits exceed the average, thereby preserving normal reasoning patterns. Otherwise, the reasoning cannot proceed as shown in Appendix B.4.

We then define the bias magnitude $\delta_{t,i}$ using a threshold $\tau$, which is consistent across all models and tasks:

$$\delta_{s,i} = (1 - \hat{d}_s) \cdot \begin{cases} \sqrt{m_{t,i}}, & \hat{d}_s \geq \tau, \\ m_{t,i}, & \hat{d}_s < \tau. \end{cases} \tag{12}$$

Although our central objective is to mitigate overthinking, an essential challenge lies in removing redundant reflections without disrupting the model's normal reasoning process, particularly when solving difficult problems that inherently require deeper reflection. Thus, to protect the integrity of normal reasoning, our formulation scales the bias magnitude by $1 - \hat{d}_s$, ensuring weaker suppression for high-difficulty steps and stronger suppression for low-difficulty ones. Furthermore, when difficulty surpasses the threshold ($\hat{d}_s \geq \tau$), we introduce the square root of the margin $m_{t,i}$ to additionally reduce the bias magnitude. This design ensures gentler suppression in challenging scenarios, preserving essential reflective exploration without unintended interference. The sensitivity analysis and necessity of introducing the threshold $\tau$ are illustrated in Fig. 5.

*Table 2.* **Generalization capabilities on non-mathematical benchmarks.**

| Method | GPQA-D | | StrategyQA | | CommonSenseQA | | LiveCodeBench | | TriviaQA | |
|---|---|---|---|---|---|---|---|---|---|---|
| | Pass@1↑ | #Tok↓ | Pass@1↑ | #Tok↓ | Pass@1↑ | #Tok↓ | Pass@1↑ | #Tok↓ | Pass@1↑ | #Tok↓ |
| **DeepSeek-R1-Distill-Qwen-7B** | | | | | | | | | | |
| Baseline | 38.4 | 7518 | 88.0 | 359 | 64.7 | 746 | 57.5 | 8504 | 19.2 | 1295 |
| Ours | 47.0 (+8.6) | 5304 (-29.4%) | 88.3 (+0.3) | 304 (-15.3%) | 65.6 (+0.9) | 540 (-27.6%) | 57.0 (-0.5) | 8061 (-5.2%) | 19.6 (+0.4) | 619 (-52.2%) |
| **Qwen3-4B-Thinking-2507** | | | | | | | | | | |
| Baseline | 66.2 | 9210 | 91.7 | 1533 | 78.3 | 2616 | 88.8 | 9771 | 33.5 | 1041 |
| Ours | 68.2 (+2.0) | 6205 (-32.6%) | 91.7 (+0.0) | 1520 (-0.8%) | 79.4 (+1.1) | 2190 (-16.3%) | 89.0 (+0.2) | 8581 (-12.2%) | 34.1 (+0.6) | 729 (-30.0%) |
| **QwQ-32B** | | | | | | | | | | |
| Baseline | 67.7 | 7732 | 95.1 | 276 | 85.3 | 724 | 91.7 | 6641 | 71.7 | 630 |
| Ours | 67.7 (+0.0) | 5699 (-26.3%) | 95.3 (+0.2) | 238 (-13.8%) | 85.5 (+0.2) | 584 (-19.3%) | 92.0 (+0.3) | 5688 (-14.4%) | 71.7 (+0.0) | 592 (-6.0%) |
| **Qwen3-14B** | | | | | | | | | | |
| Baseline | 65.1 | 7654 | 94.3 | 279 | 83.6 | 1038 | 90.0 | 7125 | 66.4 | 508 |
| Ours | 65.2 (+0.1) | 6180 (-19.3%) | 96.0 (+1.7) | 210 (-24.7%) | 83.9 (+0.3) | 946 (-8.9%) | 89.0 (-1.0) | 5972 (-16.2%) | 66.8 (+0.4) | 440 (-13.4%) |

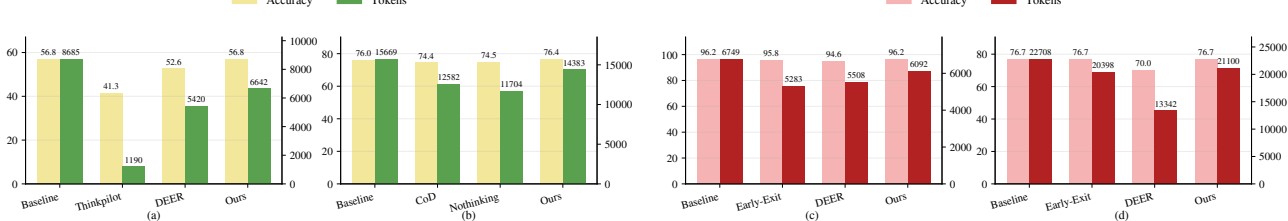

*Figure 4.* **(a–b)** Olympiad performance of (a) R1-Qwen-7B and (b) Qwen3-4B. **(c)** Early-exit evaluation on Math-500 for Qwen3-4B. **(d)** Early-exit evaluation on AIME2025 for Qwen3-4B.

# 4. Experiment

Evaluation is conducted on benchmarks spanning multiple reasoning domains. **Mathematical reasoning datasets**: Math-500 (Lightman et al., 2023), AIME2024 (AI-MO, 2024a), AIME2025 (OpenCompass, 2025), AMC23 (AI-MO, 2024b), GSM8K (Cobbe et al., 2021), Olympiad Bench (He et al., 2024), MMLU$_{\text{algebra}}$ (Hendrycks et al., 2020). **Scientific reasoning datasets**: GPQA-Diamond (Rein et al., 2024). **Code reasoning datasets**: LiveCodeBench (Jain et al., 2024). **Implicit reasoning datasets**: StrategyQA (Geva et al., 2021). **Commonsense reasoning datasets**: CommonSenseQA (Talmor et al., 2019). **Knowledge-intensive question answering datasets**: TriviaQA (Joshi et al., 2017). For each backbone, a regressor is fitted offline on 600 randomly sampled problems from Math (Hendrycks et al., 2021) and remains fixed across all evaluations. A sensitivity analysis of this choice is reported in Fig. 5(c). Additional experimental settings and baseline details are provided in Appendix D.

## 4.1. Main Results

As shown in Table 1, Table 4, Table 2, and Fig. 4(a–b), our method consistently outperforms all baselines, achieving up to **40.6% token reduction** and **6.7% absolute accuracy gains** on mathematical benchmarks, and up to **52.5% token reduction** and **8.6% absolute accuracy gains** on non-mathematical benchmarks. The gains generalize beyond Qwen backbones to alternative architectures such as

*Table 3.* **Difficulty Awareness Ablation.** R1-Qwen-7B.

| Setting | Math-500 | | AIME2024 | | GSM8K | |
|---|---|---|---|---|---|---|
| | Pass@1↑ | #Tok↓ | Pass@1↑ | #Tok↓ | Pass@1↑ | #Tok↓ |
| Baseline | 92.0 | 3955 | 50.0 | 13008 | 90.6 | 1214 |
| Ours | **92.0** | 3216 | **53.3** | 10906 | **91.1** | 880 |
| Static | 88.4 (-3.6) | 2735 | 33.3 (-16.7) | **7738** | 89.1 (-1.5) | **798** |
| Entropy-based | 91.0 (-1.0) | 3603 | 53.3 | 11911 | 90.2 (-0.4) | 1172 |

LLaMA family models (Dubey et al., 2024), demonstrating strong cross-architecture effectiveness. Moreover, even when the regressor is fitted only on Math and applied to other datasets without adaptation, the method maintains high accuracy and strong efficiency gains. This suggests that the temporal evolution patterns of reasoning difficulty learned from mathematical trajectories are qualitatively similar across diverse reasoning domains, enabling effective transfer. Detailed regressor analysis is provided in Appendices A.4 and A.5. Further results on the domain generalizability of the regressor are reported in Appendix A.6.

## 4.2. Ablation Study

**Importance of the regressor.** Table 3 shows that replacing adaptive difficulty awareness with a static coefficient leads to a substantial degradation in accuracy. Static suppression indiscriminately over-suppresses challenging instances, reducing the method to a conventional efficiency strategy that fails to balance accuracy and cost. While token-level entropy (as an alternative proxy) captures local uncertainty, it lacks a global view of the reasoning trajectory and fails to distinguish globally complex problems. In contrast, our trajectory-level representation enables a temporally consis-

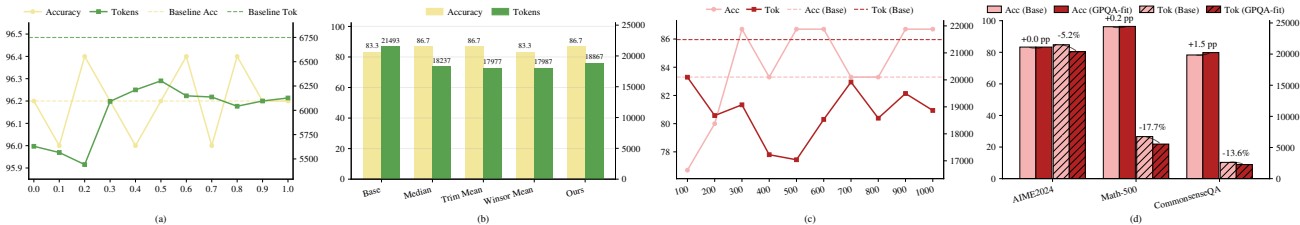

*Figure 5.* **Detailed analysis of Qwen3-4B.** (a) Hyperparameter sensitivity on MATH-500. (b) Comparison of different logits-statistic variants on AIME 2024. (c) Sensitivity of regressor fitting to sample size on AIME 2024. (d) Performance of the regressor fitted on data from different domains.

*Table 4.* **Performance on R1-Llama-8B.**

|  | Math-500 | | AIME2024 | |
|---|---|---|---|---|
| Setting | Pass@1↑ | #Tok↓ | Pass@1↑ | #Tok↓ |
| Baseline | 86.6 | 4333 | 40.0 | 14504 |
| NoThinking (Ma et al., 2025) | 66.2 | 2680 | 36.7 | 9447 |
| Ours | **86.6** | 3633 | **53.3** | 11876 |

*Table 5.* **Ablation on regressor type.** Results with Qwen3-4B.

|  | Math-500 | | AIME2024 | | AIME2025 | |
|---|---|---|---|---|---|---|
| Regressor | Pass@1↑ | #Tok↓ | Pass@1↑ | #Tok↓ | Pass@1↑ | #Tok↓ |
| Baseline | 96.2 | 6749 | 83.3 | 21493 | 76.7 | 22708 |
| OLS | 96.2 | 5807 | 83.3 | **17314** | 83.3 | 20880 |
| Ridge | 96.2 | 6092 | **86.7** | 18867 | 76.7 | 21100 |
| Random Forest | 95.6 | **5657** | 76.7 | 19117 | 76.7 | 22528 |
| Elastic Net | **96.6** | 5803 | 83.3 | 17698 | 76.7 | 20910 |
| Gradient Boosted Trees | 96.4 | 5892 | 83.3 | 18372 | 76.7 | 20559 |

tent difficulty assessment, which is essential for reliable control during reasoning.

**Impact of an alternative efficient strategy.** We further evaluate alternative efficiency strategies by integrating difficulty awareness into an early-exit mechanism (Fig. 4(c–d)). While this variant outperforms existing early-exit baselines, its reliance on discrete stopping decisions inherently limits the granularity of control. In contrast, our soft difficulty-aware mechanism provides continuous, trajectory-level control over computation, enabling finer-grained adjustment and consistently yielding a more favorable balance between efficiency and accuracy. See Appendix B.1 for implementation details. Additional results on using a GRU-based policy to guide efficient reasoning and on the bidirectional DyCon strategy are discussed in Appendices B.2 and B.11, respectively.

**Sensitivity to the hyperparameter.** Fig. 5(a) analyzes the sensitivity to the hyperparameter that balances the linear and square-root distance terms. Increasing the weight on the square-root term leads to more conservative inference and higher token usage, whereas increasing the weight on the linear term improves efficiency with a modest reduction in accuracy.

**Impact of aggregation operator choices.** As shown in Fig. 5(b), we replace the mean with the median, trimmed mean, and winsorized mean for aggregating token-level

*Table 6.* **Vocabulary Ablation.** Qwen3-4B on Math-500.

| Method | Pass@1↑ | #Tok↓ |
|---|---|---|
| Baseline | 96.2 | 6749 |
| NoWait vocab (Wang et al., 2025a) | 96.2 | 6092 |
| SEAL vocab (Chen et al., 2025) | **96.4** | **5753** |

states. The results show comparable performance across aggregation choices, indicating low sensitivity to the specific operator. Stability analysis is shown in Appendix B.3.

**Impact of regressor data and model choice.** Fig. 5(c–d) studies the effect of regressor fitting data scale and source, while Table 5 compares different regressor architectures for difficulty prediction. We observe that insufficient fitting data substantially degrades predictive accuracy and downstream performance, whereas performance improvements largely saturate at around 300 samples. Moreover, regressors trained on GPQA exhibit strong cross-domain transferability, generalizing well to Math and other benchmarks. We further discuss the noise introduced by using reasoning length as a difficulty proxy in Appendix B.8, where removing samples with redundant reasoning is shown to degrade DyCon's performance.

Across regressor types, DyCon remains broadly robust, with Elastic Net yielding further improvements on Math-500. In contrast, Random Forest leads to degraded performance, consistent with its inferior predictive quality ($R^2 = 0.6398$ compared to approximately 0.8 for other regressors). Overall, these results highlight that accurate difficulty regression is a key factor for reliable difficulty estimation and effective downstream control. Further analyses of regressor fitting, more complex nonlinear regressors such as MLPs, and additional experiments on iteratively refining the regressor with DyCon-generated trajectories are provided in Appendices A.4, B.12, and B.10, respectively.

Across regressor types, DyCon remains broadly robust, with Elastic Net yielding further improvements on Math-500. In contrast, Random Forest leads to degraded performance, consistent with its inferior predictive quality ($R^2 = 0.6398$ compared to approximately 0.8 for other regressors). Over-

all, these results highlight that accurate difficulty regression is a key factor for reliable difficulty estimation and effective downstream control. Detailed analyses of regressor fitting are provided in Appendix A.4, and additional studies on more complex nonlinear regressors, such as MLPs, are presented in Appendix B.12.

**Impact of the vocabulary design.** Table 6 shows that replacing our suppression vocabulary with the SEAL (Chen et al., 2025) reflection list yields comparable or even superior performance. This result suggests that DyCon is largely insensitive to the exact choice of suppression vocabulary and remains effective as long as reflective terms are appropriately suppressed. More detailed analyses of vocabulary optimization and token sensitivity are provided in Appendix B.7, and cross-lingual analyses are presented in Appendix B.9.

## 5. Conclusion

This paper shows that LLMs continuously encode difficulty signals, which we leverage for adaptive inference. Our proposed method, **DyCon**, is training-free and improves efficiency while preserving performance. Extending DyCon to multi-modal scenarios is a promising future direction.

## Acknowledgements

This work was supported by the Guangdong Basic and Applied Basic Research Foundation (2025A1515011546), the Shenzhen Science and Technology Program (JCYJ20240813105901003, ZDCY20250901113000001), the Zhongguancun Academy under Grant No. C20250201, and the National Natural Science Foundation of China (NSFC) via Grant No. 92570120.

## Impact Statement

This paper proposes DYCON, a training-free dynamic control mechanism for Large Reasoning Models to improve inference efficiency by reducing redundant reasoning while preserving accuracy. By modeling evolving problem difficulty from latent representations, our approach adaptively reallocates computation during reasoning, lowering inference-time cost and improving accessibility under constrained computational budgets.

Potential risks are similar to those of general-purpose reasoning language models. Increased efficiency may lower the cost of misuse, and latent difficulty estimation may be unreliable on out-of-distribution or adversarial inputs, potentially leading to premature termination or insufficient reasoning. The method introduces no new data collection or training and inherits the biases and limitations of the underlying pretrained models. Responsible deployment should rely on existing safety and moderation mechanisms.

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

# Contents

# A. Further Discussion on Motivation

## A.1. System 1 or System 2: Which Reasoning Mode Is Needed?

**Summary.**    In this section, we examine whether reasoning-oriented language models should uniformly rely on slow, deliberative System 2 reasoning, or instead adaptively switch between System 1–like and System 2–like reasoning modes according to problem difficulty. Our validation study shows that explicit reasoning-termination signals can substantially reduce token consumption, especially when injected during the reasoning process, but such compression also leads to notable accuracy degradation on challenging benchmarks such as AIME and Olympiad. In contrast, simpler datasets such as GSM8K are much less affected, suggesting that easy problems often do not require extended deliberation, whereas hard problems depend critically on sustained reasoning. These findings motivate difficulty-adaptive inference: a reasoning model should estimate task difficulty either before or during generation, and dynamically allocate cognitive effort by using fast heuristic responses for simple instances while preserving deliberate reasoning for complex ones.

The dual-process theory of cognition distinguishes between fast, automatic System 1 processes and slow, deliberative System 2 reasoning (Kahneman, 2011). Recent reasoning-oriented language models draw inspiration from this framework by encouraging step-by-step deliberation through chain-of-thought supervision (Wei et al., 2022), thereby inducing behaviors characteristic of System 2 reasoning in large language models.

Reasoning language models have achieved remarkable success in domains that require complex computation and multi-step reasoning, such as mathematics and programming. However, this paradigm also introduces new challenges, including overthinking (Chen et al., 2024), underthinking (Wang et al., 2025d), and reasoning drift.

In essence, effective reasoning models should adaptively allocate cognitive effort: they should rely on System 1–like processing to produce fast and direct responses for simple problems, rather than repeatedly re-evaluating trivial cases, while engaging System 2–like deliberation for complex problems to ensure correctness through careful and sustained reasoning.

Motivated by the prevalence of overthinking, numerous studies have proposed methods to shorten the reasoning trajectories of reasoning-oriented models. Among these approaches, NoThinking (Ma et al., 2025) represents the most aggressive form of compression, as it injects an explicit termination cue to prematurely halt the chain-of-thought, forcing a reasoning model to behave in a manner analogous to System 1 processing rather than System 2 deliberation. While effective in reducing reasoning length, this strategy also incurs the largest performance degradation in terms of accuracy. To systematically evaluate this trade-off, we conduct a validation study comparing NoThinking, a standard baseline, and a variant that inserts a reasoning termination cue immediately after the first step of the model's reasoning process. The results are summarized in Table 7.

We observe that both NoThinking and NoThinking Variant substantially reduce token consumption in reasoning models. Notably, NoThinking Variant achieves a markedly stronger compression effect, reducing the average token usage by 64.28% relative to the baseline.

This result suggests that injecting an explicit reasoning-termination semantic during the reasoning process is more effective than introducing such a signal prior to the onset of reasoning, as it better preserves the model's instruction-following behavior while suppressing redundant deliberation. At the same time, we observe a pronounced accuracy degradation on more challenging benchmarks, such as **AIME** and **Olympiad**, whereas performance on simpler datasets (e.g., **GSM8K**) remains largely unaffected. This contrast suggests that complex reasoning problems critically rely on extended deliberative processes to maintain accuracy, while simpler problems can often be solved correctly with substantially reduced reasoning depth.

These observations raise a fundamental question: can a reasoning model identify problem difficulty either before or during the reasoning process, and dynamically adapt its cognitive strategy accordingly—employing a fast, heuristic-driven *System 1* mode for simpler problems, while reserving more deliberate *System 2* reasoning for harder ones?

In response to the above question, we argue that a reasoning model does not need to commit to a single cognitive strategy throughout the entire inference process. Instead, either *before* processing a problem or *during* reasoning, once the model judges the task to be sufficiently simple, it can directly switch to a fast, heuristic-driven mode of reasoning. As illustrated in Figure 6, an explicit or implicit estimation of task difficulty enables *per-question* adaptive inference, allowing the model to dynamically balance efficiency and accuracy by combining fast and slow thinking in a principled manner.

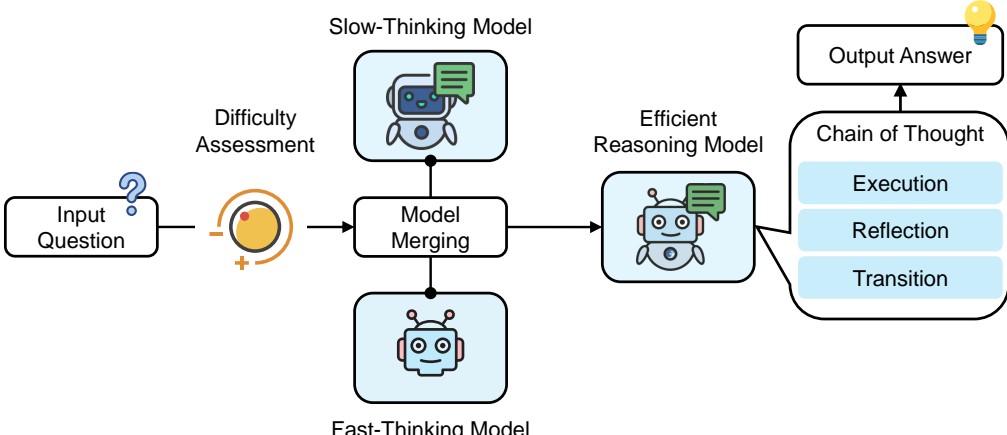

*Figure 6.* **Difficulty-adaptive reasoning.** We illustrate the central hypothesis: a reasoning model may infer problem difficulty either *before* or *during* generation, and accordingly switch its cognitive mode—using a fast, heuristic *System 1* strategy for easy instances, while allocating more deliberate *System 2* reasoning for hard ones.

*Table 7.* Comparison of accuracy (ACC) and average token usage (Tok) across reasoning control strategies. Reported $\Delta$ values indicate relative percentage changes with respect to the Baseline. **NoThinking** inserts the explicit termination cue "Okay, I have finished thinking.</think>" at step 0, while **NoThinking Variant** inserts "Okay, I have finished thinking.</think>" at step 1, allowing minimal initial deliberation before terminating the reasoning process.

| Method | Math500 | | AIME2024 | | AIME2025 | | Olympiad | | GSM8K | | AMC23 | |
|---|---|---|---|---|---|---|---|---|---|---|---|---|
| | Pass@1 | #Tok | Pass@1 | #Tok | Pass@1 | #Tok | Pass@1 | #Tok | Pass@1 | #Tok | Pass@1 | #Tok |
| **Qwen3-4B-Thinking-2507** | | | | | | | | | | | | |
| Baseline (Yang et al., 2025a) | **96.2** | 6768 | **83.3** | 21493 | **76.7** | 22708 | 76.0 | 15669 | 95.9 | 1494 | 100.0 | 11073 |
| NoThinking (Ma et al., 2025) | 95.2 | 4362 | 73.3 | 16556 | 73.3 | 19177 | 74.5 | 11704 | 95.0 | 1137 | 97.5 | 7738 |
| $\Delta$ vs. Baseline (%) | −1.04 | −35.55 | −12.00 | −22.97 | −4.43 | −15.55 | −1.97 | −25.31 | −0.94 | −23.90 | −2.50 | −30.11 |
| NoThinking Variant | 91.8 | **1975** | 63.3 | **10817** | 50.0 | **13506** | 66.7 | **6014** | 93.8 | **431** | 92.5 | **3555** |
| $\Delta$ vs. Baseline (%) | −4.57 | −70.82 | −24.01 | −49.67 | −34.81 | −40.51 | −12.24 | −61.62 | −2.19 | −71.15 | −7.50 | −67.90 |

## A.2. Who Decides Difficulty? A Model-Centric Perspective

**Summary.** In this section, we argue that task difficulty should be understood from a model-centric perspective rather than as a fixed, model-agnostic property. Since different models possess different capacities and reasoning abilities, the same problem may be difficult for a smaller model but easy for a stronger one. Therefore, difficulty should be defined relative to the model's own competence and internal uncertainty, and should be assessed dynamically during inference. Building on this view, we investigate whether difficulty awareness emerges throughout the reasoning process rather than only before generation begins. By segmenting model reasoning into discrete steps and analyzing step-level hidden states on the MATH dataset, we find that difficulty-related information is continuously encoded in the model's internal representations across reasoning steps. This suggests that models can maintain and update an intrinsic perception of task difficulty during reasoning, supporting the feasibility of dynamic, model-aware difficulty estimation.

A central question that follows is whether a model can meaningfully perceive problem difficulty. We argue that difficulty assessment should be an intrinsic, model-dependent process rather than being imposed by an external or universal discriminator. Different models possess distinct capacities, inductive biases, and reasoning strengths; consequently, the same problem may require explicit multi-step reasoning for a smaller model (e.g., 1.5B parameters), while being solvable almost immediately by a larger or more capable one (e.g., 32B parameters). This heterogeneity implies that there is no single, model-agnostic notion of difficulty. Instead, difficulty should be understood as a relative concept, defined by the model's own competence and internal uncertainty, and evaluated dynamically during inference. EPIC (Nguyen et al., 2025) employs a contrastive learning paradigm to select appropriate reasoning strategies for a given query. Its learned mapper is able to separate hard and easy mathematical problems in the latent space, indicating that problem difficulty can be effectively encoded and distinguished at the representation level. Sheng et al. (2025) further observe that special indicator tokens <think> at the onset of reasoning encode the model's internal perception of problem difficulty, suggesting that difficulty

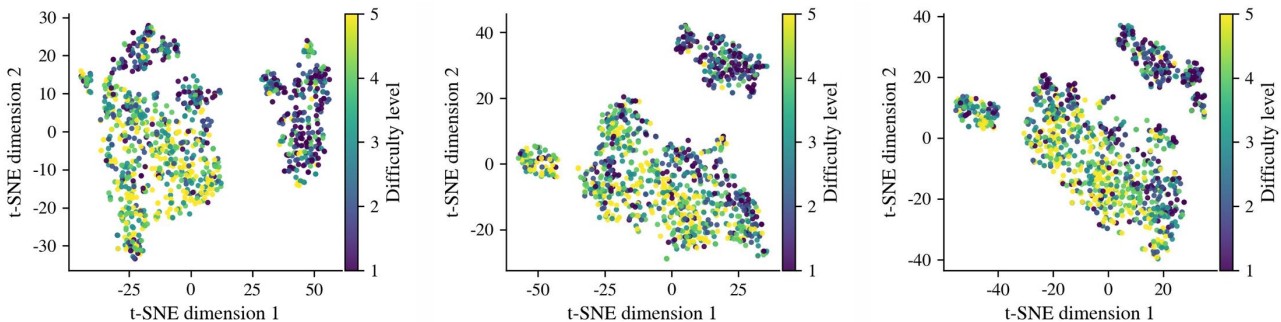

*Figure 7.* **t-SNE visualization of hidden representations colored by difficulty level.** From left to right, each panel shows the t-SNE projection of the hidden states extracted at the first, second, and third reasoning steps (defined by the delimiter \n\n), respectively. Colors indicate the ground-truth difficulty level (Level 1–Level 5). We observe that such difficulty information is continuously encoded throughout reasoning, and this phenomenon consistently holds across different model families.

awareness is already present before or at the early stages of the reasoning process. However, the aforementioned studies assess task difficulty either *before* the model begins reasoning or at the very early stages of the reasoning process. In contrast, human difficulty assessment is inherently *dynamic* and unfolds during reasoning: a problem that initially appears difficult may become easier as reasoning progresses, while a seemingly simple problem may later reveal unexpected complexity. This observation motivates a central question of our work: *does a model's assessment of task difficulty also emerge and evolve during the reasoning process itself?*

Following prior work, we segment the reasoning process of a large language model into a sequence of discrete reasoning steps, each separated by the delimiter \n\n. Formally, we denote the resulting sequence of reasoning steps as

$$\mathcal{S} = \{S_0, S_1, S_2, \ldots, S_n\}. \tag{13}$$

where each $S_s$ represents the model's intermediate reasoning state at step $s$. The final answer is generated after completing the last reasoning step $S_n$.

To investigate whether a model exhibits an awareness of task difficulty during the reasoning process, we conduct experiments on the Math dataset (Hendrycks et al., 2021), which provides discrete difficulty annotations ranging from Level 1 to Level 5. Following our step-based formulation, we associate the hidden state extracted at each reasoning step—segmented by the delimiter \n\n—with the corresponding difficulty level of the problem.

Formally, let $\ell \in \{1, 2, 3, 4, 5\}$ denote the ground-truth difficulty level of a given problem, and let

$$\mathbf{h}_s \in \mathbb{R}^d. \tag{14}$$

represent the hidden state at reasoning step $S_s$. We analyze the relationship between $\mathbf{h}_s$ and $\ell$ across different reasoning steps $s$.

As illustrated in Figure 7, we find that difficulty related information is not only encoded before or at the very beginning of the reasoning process, but is instead continuously embedded in the model's hidden representations throughout reasoning.

### A.3. Generation Length as a Generalizable Difficulty Indicator

**Summary.** In this section, we show that remaining generation length provides a continuous, fine-grained, and generalizable indicator of the model's perceived task difficulty. Unlike discrete difficulty annotations, which are often unavailable outside specific benchmarks, reasoning length naturally reflects the amount of computation a model allocates to solving a problem. Through hidden state visualizations, we observe that representations associated with shorter remaining lengths largely align with low-difficulty regions, while harder instances requiring longer reasoning trajectories occupy distinct regions of the representation space. This structure remains stable as diverse mathematical datasets and the non-mathematical GPQA benchmark are progressively incorporated, suggesting that remaining-length encoding captures a robust, task-agnostic difficulty signal rather than a dataset-specific artifact. Building on this property, we further demonstrate that a simple

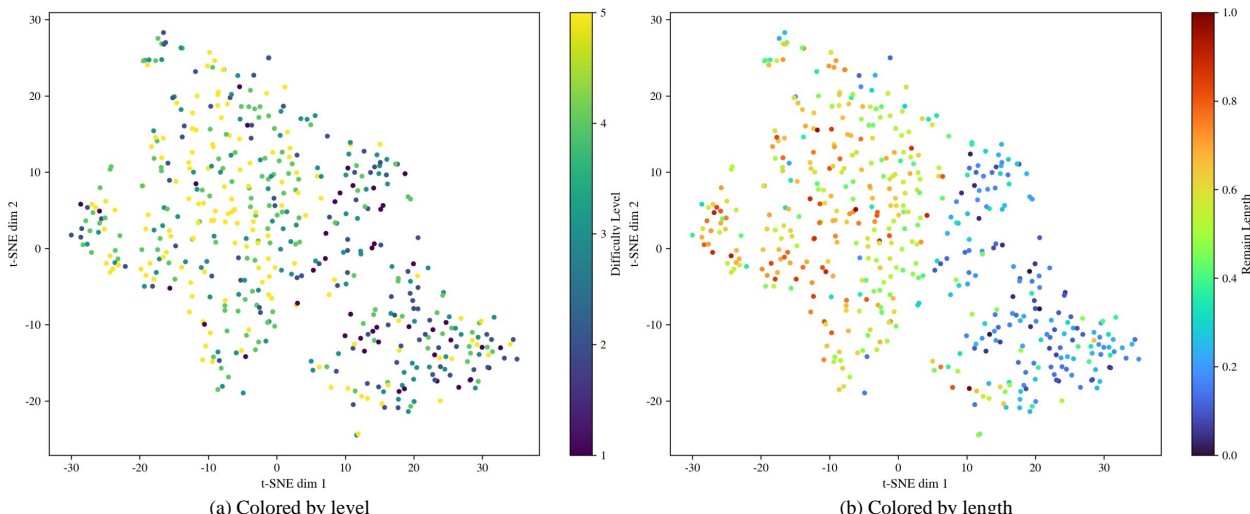

(a) Colored by level          (b) Colored by length

*Figure 8.* Visualization of Layer-28 Hidden States at the First Reasoning Break for Qwen3-4B-Thinking-2507 on Math-500. Left: colored by difficulty level; Right: colored by remaining generation length (tokens).

unsupervised difficulty classifier trained from this signal can produce intuitive difficulty distributions across datasets, supporting its potential use for difficulty estimation, dataset characterization, and future curriculum design.

Given that the model exhibits a continuous perception of difficulty throughout the reasoning process, a natural question is whether this perceived difficulty evolves along the reasoning trajectory. From a latent-variable perspective, for challenging problems, the inferred difficulty may decrease as intermediate reasoning states accumulate sufficient evidence toward a solution, whereas for simpler problems, the difficulty may be assessed as low from the outset. Such a trajectory-dependent, continuous notion of difficulty offers a more principled and expressive representation than discrete multi-class formulations.

This naturally leads to a new question: how can one construct a continuous metric that reflects the model's perceived task difficulty? While explicit difficulty annotations are available for certain mathematical benchmarks, such labels are absent in most real-world datasets, posing a challenge for generalization. A practical and broadly applicable solution is to use the model's reasoning length as a proxy for difficulty. Reasoning length is a continuous variable, and for problems of a similar type, more difficult instances tend to induce longer reasoning trajectories, whereas easier instances require substantially fewer reasoning steps. For example, on relatively simple benchmarks such as GSM8K, the average reasoning length is around 1,000 tokens, whereas on more challenging benchmarks such as AIME, it approaches 20,000 tokens.

As shown in Fig. 8, we color the hidden states of Qwen3-4B-Thinking-2507 using two different criteria: difficulty level and remaining generation length. We observe that the model not only continuously encodes difficulty-related information, but also captures signals associated with the remaining generation length. Notably, regions corresponding to low-difficulty instances exhibit a substantial overlap with those associated with shorter remaining generation lengths, which aligns well with intuitive expectations. This observation suggests that remaining generation length can serve as a more continuous and fine-grained proxy for difficulty awareness, providing a principled and effective signal for modeling task difficulty.

As shown in Fig. 9, we investigate whether this property is specific to Math-500 or persists under broader data distributions. We find that as datasets are progressively and cumulatively incorporated, the model's ability to encode remaining generation length remains consistently observable across all mathematical benchmarks.

In particular, difficult instances from Olympiad and AIME2025 concentrate in the same region of the representation space, while easier instances from GSM8K and Math-500 cluster in a distinct and aligned region, exhibiting a clear directional separation.

Furthermore, we extend this analysis to a non-mathematical benchmark, GPQA, and observe that difficult GPQA instances are mapped to the same region as difficult Olympiad problems. These results indicate that the encoding of remaining generation length reflects a persistent, task-agnostic property of the model, rather than a dataset-specific artifact, thereby

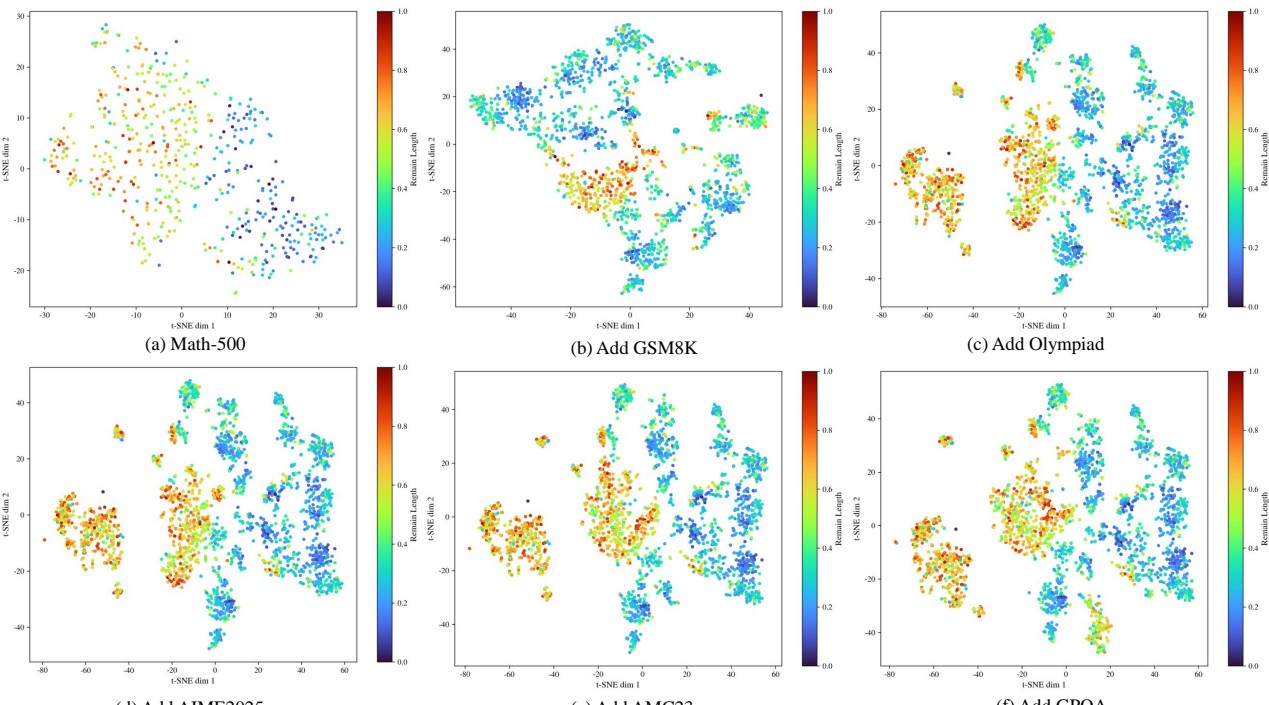

*Figure 9.* Progressive generalization of remaining-length encoding across cumulatively added datasets. Hidden states of Qwen3-4B-Thinking-2507 at the first reasoning break (Layer 28) are colored by remaining generation length. (a) Math-500; (b) Math-500 + GSM8K; (c) Math-500 + GSM8K + Olympiad; (d) Math-500 + GSM8K + Olympiad + AIME2025; (e) Math-500 + GSM8K + Olympiad + AIME2025 + AMC23; (f) Math-500 + GSM8K + Olympiad + AIME2025 + AMC23 + GPQA. The overall geometric structure remains stable as additional datasets are incorporated, indicating that remaining generation length captures a robust and highly transferable signal.

providing a principled foundation for the strong generalization capability of our method.

Building on this representational property, we show that it is possible to train a difficulty classifier in an unsupervised manner, without relying on any explicit difficulty annotations, and to assign difficulty labels to large-scale datasets. Specifically, we fit a simple binary logistic regression classifier on mathematical benchmarks and find that it can effectively annotate difficulty information across diverse datasets.

As illustrated in Fig. 10, the resulting difficulty distributions are highly intuitive. Math-500, which is designed to be difficulty-balanced, yields an approximately balanced distribution between easy and hard instances. In contrast, GSM8K, a relatively simple benchmark, is dominated by instances classified as easy, while OLYMPIAD, a substantially more challenging benchmark, exhibits a distribution heavily skewed toward the hard class. This property suggests that the learned difficulty signal can be leveraged to support large-scale dataset characterization, and may further serve as a useful signal for model pretraining or curriculum design in future work.

### A.4. Fitting a Regressor for Continuous Difficulty Estimation

**Summary.** In this section, we fit a lightweight regressor to estimate continuous task difficulty from hidden-state representations using the model's remaining generation length as supervision. To obtain a stable regression target, we apply a logarithmic transformation followed by min–max normalization, which effectively mitigates the heavy right-tailed distribution of generation lengths. We then train Ridge regressors on step-level hidden states and find that difficulty-related signals become stronger in intermediate-to-late layers across multiple reasoning models. Empirically, the log-transformed and normalized target consistently outperforms raw or directly normalized remaining length, with the best models achieving strong validation performance. Finally, we introduce an automatic validation-based procedure for selecting both the optimal hidden layer and regularization strength, ensuring robust regressor fitting without manual tuning or test-set leakage.

Building on the properties of large language models discussed above, we fit a regressor to predict a continuous difficulty-

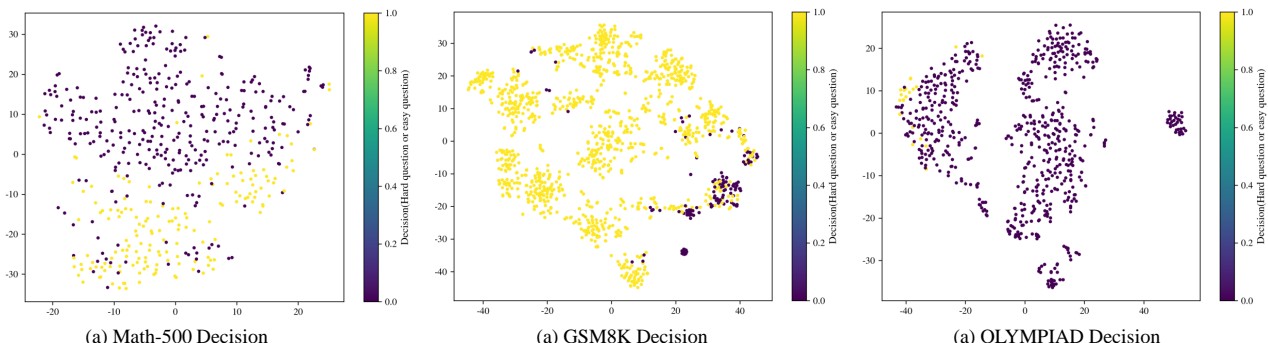

(a) Math-500 Decision        (a) GSM8K Decision        (a) OLYMPIAD Decision

*Figure 10.* Unsupervised difficulty classification results across datasets using a logistic regression classifier.

related signal. Our regression target is derived from the model's remaining generation length. Specifically, given the raw remaining length $y$, we first apply a logarithmic transformation followed by min–max normalization:

$$\tilde{y} = \frac{\log(1+y) - y_{\min}}{y_{\max} - y_{\min}}, \quad y_{\min} \triangleq \min_i \log(1+y_i), \ y_{\max} \triangleq \max_i \log(1+y_i). \tag{15}$$

where $y_{\min}$ and $y_{\max}$ denote the minimum and maximum remaining lengths observed in the dataset, respectively.

This transformation is motivated by the empirical distribution of generation lengths, which is strongly right-skewed and contains a small number of extremely long outputs. Applying a logarithmic transform effectively compresses the tail of the distribution, yielding a more stable and well-conditioned regression target. We fit a Ridge regression model to predict the normalized remaining-length signal from hidden-state representations. Let $\mathbf{h}_i \in \mathbb{R}^d$ denote the hidden state extracted from the selected layer at the first reasoning break for sample $i$, and let $\tilde{y}_i$ be the corresponding regression target defined in Eq. (X). The Ridge regressor is trained by minimizing the following objective:

$$\mathbf{w}^* = \arg\min_{\mathbf{w}} \sum_{i=1}^{N} \left( \mathbf{w}^\top \mathbf{h}_i - \tilde{y}_i \right)^2 + \lambda \|\mathbf{w}\|_2^2, \tag{16}$$

where $\mathbf{w} \in \mathbb{R}^d$ denotes the regression weights and $\lambda$ is the regularization coefficient. The $\ell_2$ regularization term mitigates overfitting and stabilizes training when the hidden representations are high-dimensional and potentially correlated.

As shown in Fig. 11, we fit a regressor to predict the remaining length from the hidden states at each layer. We observe that difficulty-related signals are progressively strengthened with increasing depth, and reach their maximum at intermediate-to-late layers.

As shown in Table 8, we report detailed regression performance under different target normalization strategies. We find that applying a logarithmic transformation followed by min–max normalization consistently yields the best performance, which is consistent with the heavy right-tailed distribution of model-generated remaining lengths. The best models achieve an $R^2$ of approximately 0.8, indicating that difficulty-related signals are strongly encoded in the hidden representations. This provides a stable and reliable signal source for downstream difficulty-aware control.

**Automatic layer and hyperparameter selection.** To avoid manual tuning, we automatically select both the hidden layer and the regressor hyperparameter via a validation-based grid search. Specifically, we first identify the set of *common layers* that are available across all sampled trajectories, ensuring a consistent hidden dimensionality. For each candidate layer, we extract the corresponding hidden states and train a Ridge regression model to predict the (optionally normalized) remaining length.

We perform a grid search over the Cartesian product of candidate layers and Ridge regularization strengths, and evaluate each configuration on a held-out validation split. The best configuration is selected by maximizing the validation $R^2$, with validation MAE used as a tie-breaker. After selecting the optimal layer and regularization strength, we refit the regressor on

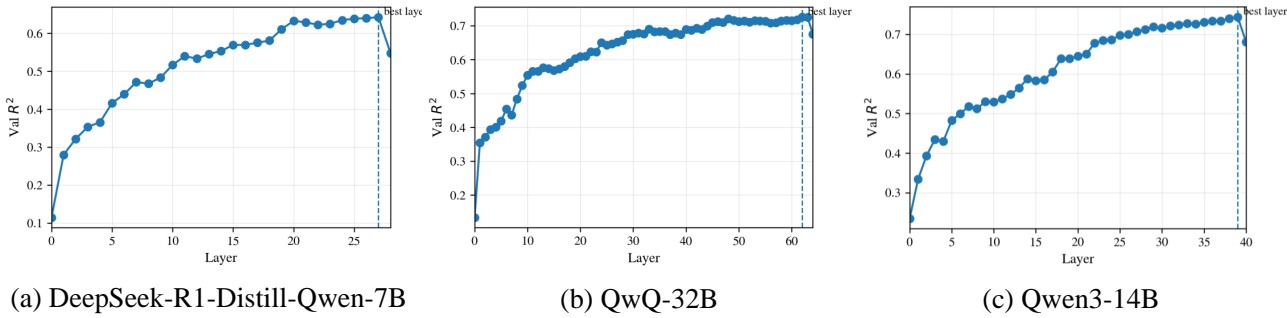

|  (a) DeepSeek-R1-Distill-Qwen-7B  |  (b) QwQ-32B  |  (c) Qwen3-14B |

*Figure 11.* Layer-wise validation $R^2$ of the remaining-length regressor across different models. Panels (a)–(c) correspond to DeepSeek-R1-Distill-Qwen-7B, QwQ-32B, and Qwen3-14B, respectively. For each layer, the best ridge regularization strength is selected based on validation performance.

*Table 8.* Summary of the remain-length regressor across models. For each model, we report the best-performing layer, the target normalization range, and the corresponding regression performance. Normalization options include Log1p + Min–Max, Min–Max, and Raw (no transform).

| Norm. | Best Layer | $y_{min}$ | $y_{max}$ | $R^2$ | MAE |
|---|---|---|---|---|---|
| **DeepSeek-R1-Distill-Qwen-7B** | | | | | |
| Log1p + Min–Max | 27 | 0.0 | 9.702 | **0.642** | **0.063** |
| Min–Max | 27 | 0.0 | 16364 | 0.522 | 0.101 |
| Raw | 27 | – | – | 0.501 | 1735 |
| **QwQ-32B** | | | | | |
| Log1p + Min–Max | 62 | 0.0 | 10.394 | **0.726** | **0.052** |
| Min–Max | 62 | 0.0 | 32682 | 0.635 | 0.058 |
| Raw | 62 | – | – | 0.606 | 2043 |
| **Qwen3-14B** | | | | | |
| Log1p + Min–Max | 39 | 2.302 | 10.392 | **0.744** | **0.055** |
| Min–Max | 39 | 9.0 | 32612 | 0.649 | 0.064 |
| Raw | 39 | – | – | 0.619 | 1939 |
| **Qwen3-4B-Thinking-2507** | | | | | |
| Log1p + Min–Max | 25 | 1.945 | 10.731 | **0.802** | **0.052** |
| Min–Max | 25 | 6.000 | 45753 | 0.642 | 0.053 |
| Raw | 25 | – | – | 0.618 | 2577 |

the combined training and validation set and report performance on a held-out test set. This procedure enables data-driven selection of both the representational layer and model capacity, while preventing test-set leakage.

### A.5. Trend Analysis of Difficulty Estimation

**Summary.** In this section, we show that the difficulty estimator learned from MATH transfers well to unseen reasoning benchmarks and preserves meaningful dataset-level difficulty trends. Across different base models, the regressor consistently assigns lower difficulty scores to simpler benchmarks such as GSM8K, moderate scores to Math-500, and higher scores to AIME- and Olympiad-style benchmarks. This alignment with ground-truth difficulty indicates that the regressor captures generalizable temporal patterns of reasoning difficulty rather than merely memorizing the training distribution, further supporting its reliability as the difficulty signal used by DyCon.

In this section, we further examine whether the learned difficulty estimator captures meaningful difficulty trends across benchmarks. While the main experiments demonstrate that DyCon can improve reasoning efficiency and accuracy, it is also important to verify that the underlying difficulty regressor produces reasonable estimates beyond the training distribution. To this end, we fit the regressor on the MATH dataset and evaluate it on multiple benchmarks with different levels of reasoning complexity. The results show that the predicted difficulty scores are closely aligned with the corresponding ground-truth difficulty scores, and that the regressor can effectively recover the expected dataset-level difficulty ordering.

*Table 9.* Trend analysis of difficulty estimation across benchmarks. We report the mean regressor-predicted difficulty scores and the corresponding ground-truth difficulty scores. Higher values indicate higher estimated reasoning difficulty.

| Model | Type | Math-500 | AIME24 | AIME25 | GSM8K | Olympiad |
|---|---|---|---|---|---|---|
| Qwen3-4B-Thinking-2507 | Regressor Prediction | 0.69 | 0.78 | 0.79 | 0.49 | 0.80 |
| Qwen3-4B-Thinking-2507 | Ground Truth | 0.69 | 0.78 | 0.80 | 0.51 | 0.81 |
| DeepSeek-R1-Distill-Qwen-7B | Regressor Prediction | 0.76 | 0.85 | 0.86 | 0.61 | 0.81 |
| DeepSeek-R1-Distill-Qwen-7B | Ground Truth | 0.77 | 0.85 | 0.86 | 0.61 | 0.83 |
| Qwen3-14B | Regressor Prediction | 0.65 | 0.77 | 0.76 | 0.47 | 0.77 |
| Qwen3-14B | Ground Truth | 0.65 | 0.75 | 0.77 | 0.47 | 0.77 |
| QwQ-32B | Regressor Prediction | 0.73 | 0.80 | 0.80 | 0.61 | 0.83 |
| QwQ-32B | Ground Truth | 0.73 | 0.82 | 0.82 | 0.61 | 0.86 |

The difficulty estimator in DyCon is designed to provide an online estimate of the current reasoning difficulty during generation. Therefore, a desirable property is that its predictions should not only be accurate at the instance level, but should also preserve meaningful aggregate trends across datasets. In particular, benchmarks that typically require longer and more complex reasoning, such as AIME and Olympiad-style problems, are expected to receive higher difficulty scores, while relatively simpler grade-school arithmetic problems, such as GSM8K, should receive lower scores.

To analyze this property, we train the regressor on the MATH dataset and then evaluate it on five benchmarks: Math-500, AIME24, AIME25, GSM8K, and Olympiad. These benchmarks cover a broad spectrum of mathematical reasoning difficulty. For each benchmark, we compute the mean predicted difficulty score produced by the regressor and compare it with the corresponding ground-truth difficulty score. Higher scores indicate higher estimated reasoning difficulty.

As shown in Table 9, the regressor produces predictions that are highly consistent with the ground-truth difficulty values across different base models. For Qwen3-4B-Thinking-2507, the predicted scores almost exactly match the ground-truth scores on Math-500 and AIME24, and remain very close on AIME25, GSM8K, and Olympiad. Similar patterns can also be observed for DeepSeek-R1-Distill-Qwen-7B, Qwen3-14B, and QwQ-32B. Across all models, GSM8K consistently receives the lowest difficulty score, Math-500 receives a moderate score, and AIME/Olympiad benchmarks receive substantially higher scores.

This trend is important because it suggests that the regressor is not merely memorizing superficial properties of the training set. Instead, it generalizes to unseen benchmarks and preserves the relative difficulty structure among datasets. In particular, the estimated ordering broadly follows the expected pattern: Olympiad and AIME-style benchmarks are more difficult than Math-500, while GSM8K is the easiest among the evaluated datasets. The close agreement between regressor predictions and ground-truth scores further supports the reliability of the learned difficulty estimator used by DyCon.

Overall, these results provide additional evidence that the learned difficulty estimator captures meaningful reasoning difficulty rather than dataset-specific artifacts. The estimator can recover both fine-grained numerical scores and coarse-grained benchmark-level trends, which makes it suitable for dynamically controlling the reasoning behavior of DyCon during inference.

### A.6. Domain Generalizability of Difficulty Estimation

**Summary.** In this section, we evaluate whether the difficulty estimator learned from mathematical reasoning can generalize to non-math domains. We find that a regressor fitted only on MATH already transfers reasonably well to CommonsenseQA and GPQA, indicating that step-level hidden representations encode difficulty-related signals beyond mathematical tasks. However, its calibration degrades on domains with substantially different interaction patterns, such as MultiChallenge. To improve robustness, we refit the regressor on a balanced mixture of MATH, CommonsenseQA, GPQA, and MultiChallenge. The refitted estimator achieves much closer alignment with ground-truth difficulty across both math and non-math benchmarks, while also generalizing to the unseen C4 domain. These results suggest that diverse-domain fitting strengthens the calibration and transferability of difficulty estimation, supporting DyCon as a general difficulty-aware control mechanism rather than a math-specific method.

The difficulty estimator in DyCon is designed to estimate the model's reasoning difficulty during generation. Since our

*Table 10.* Out-of-domain evaluation of the difficulty regressor fitted on the MATH dataset using Qwen3-4B-Thinking-2507. We report the mean regressor-predicted difficulty scores and the corresponding ground-truth difficulty scores. Higher values indicate higher estimated reasoning difficulty.

| Type | CommonsenseQA | GPQA | MultiChallenge |
|---|---|---|---|
| Regressor Prediction | 0.59 | 0.61 | 0.64 |
| Ground Truth | 0.54 | 0.67 | 0.48 |

*Table 11.* Domain generalization of the refitted difficulty regressor using Qwen3-4B-Thinking-2507. The regressor is refitted on a balanced mixture of MATH, CommonsenseQA, GPQA, and MultiChallenge, and evaluated across both math and non-math benchmarks. We report the mean regressor-predicted difficulty scores and the corresponding ground-truth difficulty scores. Higher values indicate higher estimated reasoning difficulty.

| Type | CommonsenseQA | GPQA | MultiChallenge | Math-500 | AIME2024 | AIME2025 | AMC23 | GSM8K | Olympiad | C4 |
|---|---|---|---|---|---|---|---|---|---|---|
| Regressor Prediction | 0.58 | 0.70 | 0.53 | 0.72 | 0.79 | 0.80 | 0.75 | 0.56 | 0.78 | 0.58 |
| Ground Truth | 0.58 | 0.70 | 0.53 | 0.73 | 0.80 | 0.82 | 0.74 | 0.56 | 0.81 | 0.59 |

main experiments fit the regressor on mathematical reasoning data, it is important to test whether the learned signal is specific to math problems or transferable to other reasoning domains. Non-math benchmarks introduce different forms of difficulty: CommonsenseQA relies more on implicit world knowledge, GPQA requires expert-level scientific reasoning, and MultiChallenge evaluates realistic multi-turn conversation abilities such as context tracking and instruction following.

To evaluate this transferability, we apply the MATH-fitted regressor to CommonsenseQA, GPQA, and MultiChallenge using Qwen3-4B-Thinking-2507 as the base model. Table 10 reports the mean regressor-predicted difficulty scores and the corresponding ground-truth difficulty scores. The regressor provides reasonably aligned estimates on CommonsenseQA and GPQA, suggesting that the difficulty signal learned from math data contains transferable information. However, the gap becomes larger on MultiChallenge, where the predicted difficulty is noticeably higher than the ground-truth value. This indicates that single-domain fitting can generalize to some extent, but may not fully capture difficulty patterns in domains with very different interaction structures.

The results in Table 10 motivate a more diverse fitting strategy. If different domains express reasoning difficulty through different generation behaviors, then exposing the regressor to multiple reasoning distributions should improve its calibration. Therefore, we refit the regressor on a balanced dataset spanning MATH, CommonsenseQA, GPQA, and MultiChallenge. The refitting split is separated from the final evaluation split, so the reported results are not obtained by evaluating on the same examples used for fitting.

Table 11 presents the results of the refitted regressor. Compared with the MATH-fitted setting in Table 10, the refitted regressor achieves much closer alignment with ground-truth difficulty scores on CommonsenseQA, GPQA, and MultiChallenge. Importantly, this improvement does not come at the cost of performance on mathematical benchmarks. The refitted regressor remains well aligned with the ground-truth scores on Math-500, AIME2024, AIME2025, AMC23, GSM8K, and Olympiad. It also generalizes well to C4 (Raffel et al., 2020), which is not included in the refitting mixture, suggesting that diverse fitting can improve the robustness of difficulty estimation beyond the training domains.

Overall, the two experiments lead to complementary conclusions. First, the MATH-fitted regressor already transfers reasonably well to several non-math domains, showing that step-level representations contain difficulty-related signals beyond mathematical reasoning. Second, fitting the regressor on diverse reasoning distributions further improves its calibration and robustness, especially for domains whose reasoning patterns differ from math. These results support the use of DyCon as a general difficulty-aware control mechanism rather than a method restricted to mathematical benchmarks.

### A.7. Recovering Token-Space Performance and Cross-Distribution Generalization

**Summary.** In this section, we further evaluate whether the remaining-length regressor preserves its predictive utility after mapping normalized predictions back to the original token space. The results show that the regressor can estimate token-level remaining length with relatively small errors on simpler datasets such as GSM8K, while larger absolute errors arise on harder benchmarks such as AIME2025 and Olympiad due to both the intrinsic uncertainty of difficult reasoning and the compression effect of the logarithmic transformation. Nevertheless, the regressor still captures the coarse scale of reasoning effort and becomes increasingly accurate in the later stages of difficult problems, supporting the view that hard

*Table 12.* Token-space evaluation of the remaining-length regressor for Qwen3-4B-Thinking-2507 across mathematical datasets. We report the average ground-truth remaining length, the average prediction, and the corresponding absolute and percentage errors (lower is better).

| Dataset | Ground Truth | Prediction | Abs. Error | Pct. Error (%) |
|---|---|---|---|---|
| Math-500 | 5963 | 5212 | 751 | 12.60 |
| GSM8K | 1137 | 953 | 184 | 16.18 |
| AIME2025 | 21959 | 16211 | 5748 | 26.17 |
| Olympiad | 14823 | 10059 | 4764 | 32.13 |

instances gradually transition into easier regimes as reasoning progresses.

A key question is whether the strong performance observed in the transformed target space can be retained after mapping predictions back to the original token space. Given the estimated $y_{\min}$ and $y_{\max}$, we invert the normalization to recover predictions in terms of the original remaining-token counts.

$$\hat{y} = \exp\left(\hat{\tilde{y}}\left(y_{\max} - y_{\min}\right) + y_{\min}\right) - 1, \tag{17}$$

We then evaluate the regressor trained on Math on other mathematical datasets with different distributions to assess its cross-distribution generalization. As shown in Table 12, we evaluate the regressor in the original token space. For simpler datasets such as GSM8K, the prediction error is on the order of $\sim$100 tokens, indicating a high level of accuracy. In contrast, for more challenging datasets such as AIME2025 and Olympiad, the prediction error increases, although the regressor still captures the coarse scale of the remaining length. From a modeling perspective, this behavior is partly attributable to the logarithmic transformation, which naturally compresses large values and thus leads to larger absolute errors for long outputs after inverse transformation. From an intuitive perspective, this trend is also expected: for simple problems, the model can reliably anticipate how many tokens are required, whereas for difficult problems, the model primarily recognizes that the problem is hard, but cannot precisely predict the exact number of tokens needed to reach a solution.

As shown in Figure 12, we randomly sample one instance from each dataset for visualization. We observe that the regressor can continuously track the problem difficulty and provide relatively accurate token-length predictions, particularly for simple problems. For challenging problems, the predictions are less precise, consistent with our earlier quantitative results. However, in the later stages of difficult problem solving, the regressor becomes increasingly accurate, indicating that once a difficult problem enters its closing phase, it effectively transitions into an easier regime. This phenomenon further corroborates our earlier hypothesis that hard problems tend to become easy in the later stages of reasoning.

### A.8. From Instruct-Style to Reasoning-Style: Difficulty-Adaptive Generation via a Regressor

**Summary.** In this section, we introduce a regressor-guided adaptive termination mechanism that converts the estimated reasoning difficulty into a continuous logit bias on the `</think>` token. This design selectively increases the probability of terminating reasoning when the regressor predicts low necessity for continued deliberation, while leaving the rest of the token distribution unchanged. As a result, the model can adapt its generation behavior according to task difficulty: on easier benchmarks such as GSM8K, it shifts toward instruct-style fast generation with substantially reduced token usage, whereas on harder benchmarks such as AIME2024, it largely preserves reasoning-style deliberation and maintains accuracy. These results suggest that difficulty-aware control enables reasoning models to dynamically switch between System 1–like and System 2–like behaviors. However, aggressive termination may amplify regressor errors or the model's intrinsic underthinking, causing premature stopping and accuracy degradation. Therefore, we adopt a soft control strategy in the main paper to balance efficiency gains with reasoning reliability.

Leveraging the regressor's ability to estimate task difficulty, we model the decision of whether to terminate reasoning as a continuous, difficulty-aware logit bias applied to the reasoning termination token:

$$\Delta\ell_{\langle/\text{think}\rangle}(t) = \lambda \cdot f\left(1 - r(h_t)\right), \qquad \lambda \geq 0. \tag{18}$$

Here, $h_t \in \mathbb{R}^d$ denotes the hidden state at the $t$-th reasoning checkpoint, and $r(h_t) \in [0,1]$ is the regressor's prediction of the necessity to continue reasoning. The monotonic mapping $f(\cdot) : [0,1] \to [0,1]$ transforms the regressor output into a

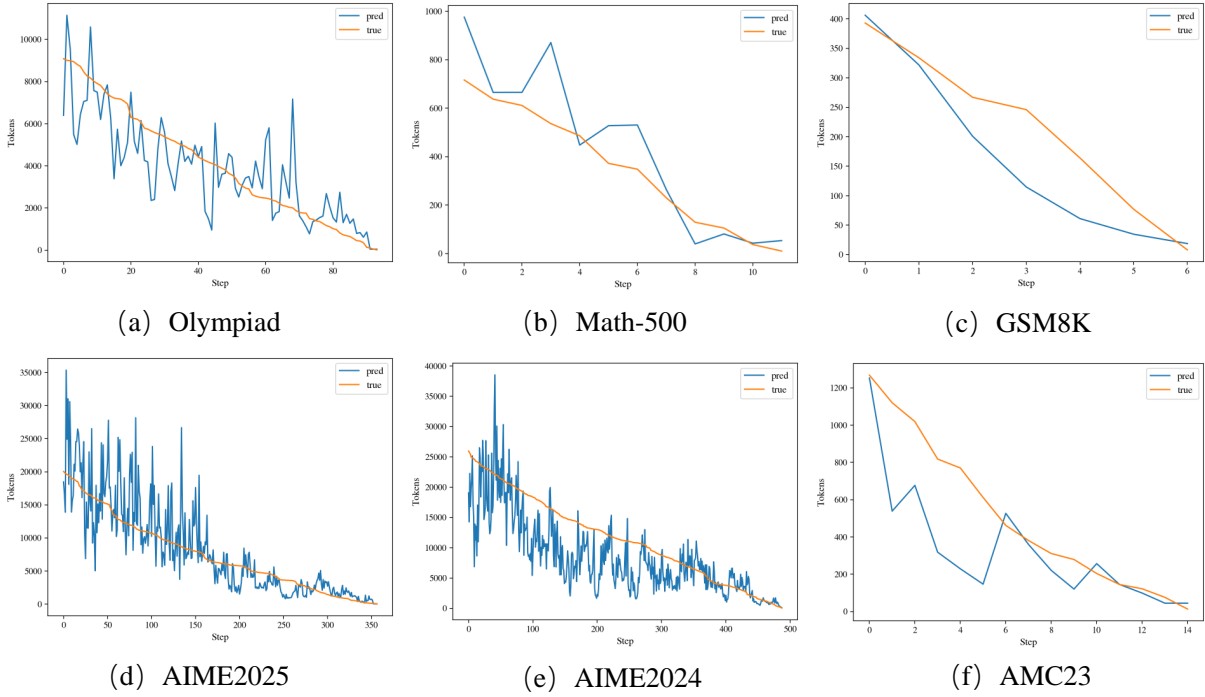

(a) Olympiad

(b) Math-500

(c) GSM8K

(d) AIME2025

(e) AIME2024

(f) AMC23

*Figure 12.* Token-space visualization of remaining-length prediction for Qwen3-4B-Thinking-2507 across datasets.

*Table 13.* Adaptive behavior switching on **Qwen3-4B-Thinking-2507**. On GSM8K, our method induces instruct-style generation with a large reduction in token usage, while on AIME2024 it preserves reasoning-style behavior.

| Model | Dataset | Method | Acc. (%) | Tokens | Behavior |
|---|---|---|---|---|---|
| Qwen3-4B-Thinking-2507 | GSM8K | Baseline | **95.0** | 1494 | Reasoning-like |
| | | Ours | 94.0 | **414** | Instruct-like |
| | AIME2024 | Baseline | 83.3 | 21493 | Reasoning-like |
| | | Ours | 83.3 | **19481** | Reasoning-like |

normalized control signal, while $\lambda$ controls the maximum strength of the termination bias. The resulting $\Delta\ell_{\langle/\text{think}\rangle}(t)$ is added to the logit of the `</think>` token at the next generation step.

During decoding, the model's conditional distribution is modified by injecting the difficulty-aware logit bias into the reasoning termination token:

$$p(y_t \mid y_{<t}) = \text{Softmax}\Big(\ell_t + \Delta\ell_{\langle/\text{think}\rangle}(t)\, e_{\langle/\text{think}\rangle}\Big), \tag{19}$$

where $\ell_t$ denotes the original pre-softmax logits at step $t$, and $e_{\langle/\text{think}\rangle}$ is a one-hot vector with value 1 at the position corresponding to the `</think>` token and 0 elsewhere. This formulation ensures that the difficulty-aware control signal selectively affects only the probability of terminating reasoning, while leaving the remaining token distribution unchanged.

As shown in Table 13, the model adaptively adjusts its generation behavior based on the regressor-assisted estimation of task difficulty. On easier benchmarks, the model exhibits instruct-style behavior, while on more challenging benchmarks it preserves reasoning-style generation. This emergent behavior is particularly encouraging, as it suggests that the reasoning model acquires the ability to autonomously switch between *System 1* (fast, shallow) and *System 2* (slow, deliberate) modes of reasoning.

However, we observe that under any form of hard or aggressive control, the model tends to suffer a non-negligible accuracy drop on easy datasets. For this reason, we adopt a *soft* control mechanism in the main paper. We attribute the observed performance degradation to multiple factors. First, the regressor is not perfectly accurate and may occasionally misclassify

*Table 14.* Stationarity order distribution of the regressor-predicted difficulty signal across reasoning trajectories (with a maximum differencing order of 6). Percentages are computed over all trajectories.

| Model | Traj. | I(0) % | I(1) % | I(2+) % | No stat. % |
|---|---|---|---|---|---|
| QwQ-32B | 1000 | 0.1 | **63.9** | 11.5 | 24.5 |
| Qwen3-14B | 1000 | 0.2 | **63.6** | 12.8 | 23.4 |
| DeepSeek-R1-Distill-Qwen-7B | 1000 | 0.3 | **64.6** | 9.7 | 25.4 |
| Qwen3-4B-Thinking-2507 | 1000 | 0.1 | **50.6** | 12.3 | 37.0 |

hard or medium-difficulty problems as easy, leading to premature termination of reasoning and consequent accuracy loss. Second, the base model itself may exhibit an *underthinking* phenomenon, where an instance is initially judged as easy due to insufficient early deliberation or overconfidence. In such cases, the regressor may further amplify this underthinking behavior, exacerbating premature stopping and increasing the likelihood of errors.

### A.9. Temporal Dynamics and Non-Stationarity of Difficulty Signals

**Summary.** In this section, we analyze the temporal dynamics of the regressor-predicted difficulty signal during reasoning. Rather than treating difficulty as a static property, we examine whether the estimated necessity of continued reasoning evolves across reasoning steps. Through ADF and KPSS stationarity tests, we find that the predicted difficulty signals are predominantly first-order non-stationary across multiple reasoning models, indicating the presence of systematic temporal trends rather than stationary fluctuations around a fixed mean. This provides mechanistic evidence that the model's perceived difficulty is dynamically updated throughout the reasoning process, supporting our view that difficulty-aware control should operate as an online, trajectory-dependent mechanism rather than a one-shot static decision.

Beyond aggregate performance metrics, we seek to understand the temporal behavior of the regressor-predicted difficulty signal during the reasoning process. Since the regressor is designed to estimate the necessity of continued reasoning at each step, this signal is inherently dynamic and may evolve as the model progressively refines its internal understanding of the problem. To characterize this temporal structure, we analyze the stationarity properties of the predicted difficulty signal across reasoning steps.

As shown in Table 14, we conduct time-series stationarity tests using ADF and KPSS on the regressor's per-step difficulty predictions. We find that for the vast majority of trajectories, the predicted signal exhibits first-order non-stationarity with a systematic trend, rather than stationary fluctuations around a constant mean. This observation provides mechanistic evidence that the model's perceived task difficulty evolves systematically over the course of reasoning, reflecting dynamic updates of its internal assessment as reasoning progresses.

### A.10. Analysis of Overthinking Behavior in LLMs

**Summary.** In this section, we perform a step-level analysis of overthinking by identifying the earliest reasoning step at which the correct answer first appears, and using this to define the early-correctness ratio and the corresponding overthinking ratio. The results show that, across multiple reasoning models, correct answers typically emerge well before the end of the full reasoning trajectory, indicating that a substantial proportion of later steps are potentially redundant. This provides direct empirical evidence that overthinking is a systematic and widespread phenomenon in reasoning-oriented language models. At the same time, the distributions reveal clear model-dependent differences in the severity of overthinking, as well as substantial instance-level variability in when correctness first arises. These findings suggest that overthinking cannot be effectively characterized or controlled by a single fixed stopping threshold, and instead motivate a more robust, distribution-aware strategy for adaptive reasoning control.

As shown in Fig. 13, we conduct a systematic step-level analysis of the overthinking phenomenon in LLMs. Specifically, we employ an LLM-based judge to identify the earliest step at which the correct answer first appears in the generated reasoning. Based on this, we first define the *early-correctness ratio* as

$$r_{\text{early}} = \frac{N_{\text{earliest}}}{N_{\text{total}}}, \tag{20}$$

where $N_{\text{earliest}}$ denotes the earliest step at which the correct answer is identified and $N_{\text{total}}$ is the total number of reasoning steps.

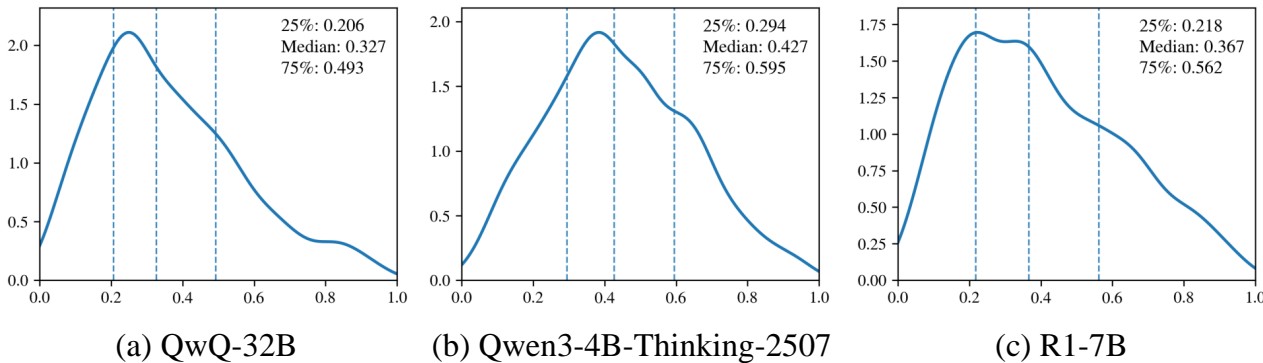

(a) QwQ-32B  (b) Qwen3-4B-Thinking-2507  (c) R1-7B

*Figure 13.* **Kernel density of earliest correctness emergence.** The distribution of $r = \text{earliest\_step}/\text{num\_steps}$ (identified by an LLM-judge) shows substantial variability across instances, indicating that the correct answer can emerge at markedly different reasoning stages.

Intuitively, a smaller $r_{\text{early}}$ indicates that correctness is achieved earlier in the reasoning process, implying that a larger fraction of subsequent steps are potentially redundant. Accordingly, we define the *overthinking ratio* as

$$r_{\text{overthink}} = 1 - r_{\text{early}} = \frac{N_{\text{total}} - N_{\text{earliest}}}{N_{\text{total}}}, \tag{21}$$

which directly quantifies the proportion of reasoning steps generated after correctness is already achieved. This metric therefore serves as a proxy for the degree of redundant reasoning, and hence the extent of overthinking exhibited by the model.

Fig. 13 visualizes the empirical distributions of $r_{\text{early}}$ for three representative models. Across all models, the median $r_{\text{early}}$ values are well below $0.5$, indicating that correct answers typically emerge in the first third to first half of the reasoning process, after which a substantial fraction of generated steps are potentially redundant. This provides direct empirical evidence that overthinking is a systematic phenomenon rather than an isolated case.

Moreover, we observe clear model-dependent differences. In particular, QwQ-32B exhibits the earliest correctness (median $r_{\text{early}} = 0.327$), suggesting the most severe overthinking behavior, while Qwen3-4B-Thinking-2507 reaches correctness later on average (median $r_{\text{early}} = 0.427$), implying relatively milder overthinking. R1-7B lies between these two models.

Importantly, all three models display wide interquartile ranges, reflecting substantial instance-level variability in the stage at which correctness first appears. This distributional spread suggests that overthinking does not occur at a fixed step or ratio, but rather varies significantly across instances. These observations naturally motivate a robust, distribution-aware hyperparameter design, instead of relying on a single fixed threshold for early stopping or suppression.

## B. Additional Experimental Results and Ablations

### B.1. Alternative Efficient Reasoning Strategies

**Summary.** In this section, we discuss alternative strategies for efficient reasoning and show that difficulty awareness can serve as a transferable control signal beyond our soft suppression framework. We evaluate both classifier-based and regressor-based early-exit methods, where reasoning is explicitly terminated once the predicted difficulty falls below a predefined threshold. The results demonstrate that difficulty-aware early exit can substantially reduce token consumption while largely maintaining accuracy across multiple benchmarks. However, compared with our soft control strategy, hard early-exit mechanisms provide coarser control and are more prone to suboptimal termination, leading to weaker overall trade-offs between efficiency and accuracy. These findings suggest that continuous difficulty-aware modulation offers a more fine-grained and reliable approach, while also highlighting promising future directions such as integrating difficulty prediction with steering-based reasoning control.

Efficient reasoning can be achieved through a wide range of strategies. Beyond our soft suppression of reflective transition terms, existing approaches include early exit, steering mechanisms, and prompt-based methods. As a transferable component,

*Table 15.* Comparison of classifier-based and regressor-based early-exit strategies on **Qwen3-4B-Thinking-2507**.

| Method | MATH-500 | | AIME24 | | AIME25 | | GSM8K | | AMC23 | | MMLU$_{algebra}$ | |
|---|---|---|---|---|---|---|---|---|---|---|---|---|
| | Acc. | Tok. | Acc. | Tok. | Acc. | Tok. | Acc. | Tok. | Acc. | Tok. | Acc. | Tok. |
| Baseline (Yang et al., 2025a) | 96.2 | 6749 | 83.3 | 21493 | 76.7 | 22708 | 95.9 | 1494 | 100 | 11073 | 94.0 | 3496 |
| Early-Exit (Classifier-based) | 95.6 | 4427 | 86.7 | 19136 | 83.3 | 16848 | 95.0 | 1264 | 100 | 7443 | 94.0 | 2341 |
| Early-Exit (Regressor-based) | 95.8 | 5283 | 73.3 | 18787 | 76.7 | 20398 | 95.2 | 1113 | 97.5 | 9422 | 94.0 | 2568 |

difficulty awareness can be naturally integrated into these alternative strategies.

As shown in Table 15, we further evaluate early-exit strategies guided by difficulty awareness. In addition to the regressor-based design, we also fit a classifier to predict whether the current problem instance can be safely terminated. In both cases, when the predicted difficulty falls below a predefined threshold, we explicitly terminate the reasoning process by appending the text sequence "</think>". The detailed experimental results are summarized in Table 15. We observe that early-exit methods augmented with difficulty awareness can substantially reduce the number of generated tokens while largely preserving accuracy. However, their performance remains inferior to our soft control approach. We attribute this gap to the finer granularity of soft modulation, which enables more precise control over the reasoning process and better preserves accuracy. As a promising direction for future work, difficulty-aware steering frameworks warrant further investigation. Since the strength of steering is inherently governed by a tunable parameter, it can be naturally coupled with difficulty prediction: the steering strength can be reduced when the model identifies a problem as difficult and increased when the problem is deemed easy. This opens several avenues for future research.

### B.2. Direct Earliest-Correctness Modeling with GRU and Underthinking

**Summary.** In this section, we investigate whether the earliest step at which a model reaches the correct answer can be directly predicted from its hidden-state trajectory. We formulate earliest-correctness prediction as a sequence labeling problem and train a GRU-based model to identify whether each reasoning step occurs after the first correct solution point. Although the GRU achieves high prediction accuracy and can guide early exit with substantial token reductions, further analysis reveals an important limitation: it often learns surface-level conclusion patterns, such as "Final answer" or "In conclusion," rather than the true point at which correctness is achieved. As a result, the model may terminate prematurely when an initial conclusion is incorrect but would have been corrected through later self-reflection, leading to underthinking. These findings suggest that direct earliest-correctness modeling is trainable but unreliable as a standalone control mechanism, motivating our preference for softer, distribution-aware difficulty control rather than hard termination based on a single predicted exit point.

Given that we can identify the earliest step at which the model first produces the ground-truth (GT) answer, and that hidden states are shown to encode evolving difficulty-related signals, we ask whether the hidden states also encode sufficient information to predict the step at which the model solves the problem.

Let $\mathbf{H}_{0:t} = \{\mathbf{h}_0, \mathbf{h}_1, \ldots, \mathbf{h}_t\}$ denote the sequence of hidden states up to step $t$. Given the earliest correct step $i^\star$, we define the binary supervision as

$$y_i = \mathbb{I}[i \geq i^\star], \tag{22}$$

where $\mathbb{I}[\cdot]$ is the indicator function.

We parameterize a sequence model $f_\theta$ with a GRU to predict the earliest-correctness signal:

$$\mathbf{s}_i = \text{GRU}(\mathbf{h}_i, \mathbf{s}_{i-1}), \tag{23}$$

$$\hat{y}_i = \sigma(\mathbf{w}^\top \mathbf{s}_i + b), \tag{24}$$

where $\sigma(\cdot)$ denotes the sigmoid function.

The model is trained with a sequence-wise binary cross-entropy loss:

$$\mathcal{L} = \frac{1}{t+1} \sum_{i=0}^{t} \text{BCE}(\hat{y}_i, y_i). \tag{25}$$

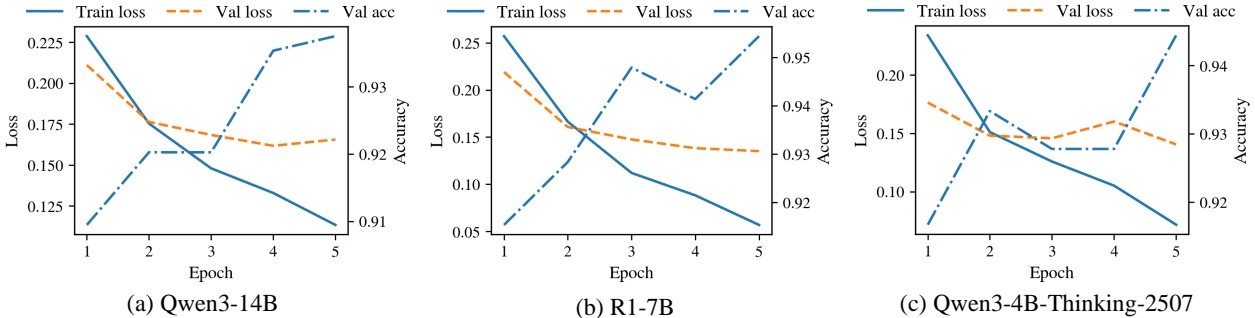

(a) Qwen3-14B      (b) R1-7B      (c) Qwen3-4B-Thinking-2507

*Figure 14.* Training curves of the GRU-based earliest-correctness predictor.

*Table 16.* Comparison between baseline and GRU-based early exit on **Qwen3-4B-Thinking-2507**.

| Method | MATH-500 | | AIME24 | | AIME25 | | GSM8K | | AMC23 | | GPQA | |
|---|---|---|---|---|---|---|---|---|---|---|---|---|
| | Acc. | Tok. | Acc. | Tok. | Acc. | Tok. | Acc. | Tok. | Acc. | Tok. | Acc. | Tok. |
| Baseline (Yang et al., 2025a) | 96.2 | 6749 | 83.3 | 21493 | 76.7 | 22708 | 95.9 | 1494 | 100 | 11073 | 66.2 | 9210 |
| Early-Exit (GRU-based) | 95.2 | 5450 | 83.3 | 17763 | 80.0 | 22227 | 94.9 | 1336 | 100 | 8690 | 67.7 | 6476 |

As shown in Table 17 and Figure 14, we find that this formulation is indeed trainable in practice. The GRU achieves relatively high accuracy in predicting the model's earliest exit point. We then use this trained GRU to guide the execution of the early-exit algorithm.

As shown in Table 16, the GRU achieves strong performance on several benchmarks and attains state-of-the-art results on some datasets. We further observe that the steps selected by the GRU are often highly precise, frequently occurring immediately after conclusion-related patterns such as "Final answer" or "In conclusion."

Compared to approaches that rely on large-model-based judges, such as FlashThinking (Jiang et al., 2025) and TrimR (Lin et al., 2025a), the GRU-based method achieves superior performance. However, we also observe substantial accuracy degradation on relatively simple datasets. We attribute this to the fact that the appearance of conclusion-style phrases does not necessarily indicate that the ground-truth answer has been correctly derived. In many cases, the model may first produce an incorrect conclusion and subsequently enter a self-reflection or correction phase. In such cases, the GRU may incorrectly identify these premature conclusion steps as valid exit points, leading to underthinking (Wang et al., 2025d).

These results suggest that the GRU primarily learns surface-level conclusion patterns rather than the true optimal point of correctness. As a result, it fails to reliably capture the genuine earliest-correctness step. This limitation motivates our use of averaging over optimal points, as there exists no stable and learnable pattern that consistently corresponds to the true optimal solving step.

### B.3. Stability Analysis of the Distance-Based Suppression Signal

**Summary.** In this section, we define and validate a logit-gap distance for measuring the relative dominance of targeted transition tokens in the model's output distribution. By computing mean and maximum positive gaps with respect to the vocabulary-wide mean logit, we show that the proposed distance yields a stable and well-calibrated scale across different reasoning models, while its extreme-value structure reflects meaningful model-dependent capability differences. After normalizing by the global logit standard deviation, the distance remains tightly structured and scale-invariant, indicating that it naturally adapts to each model's intrinsic uncertainty without requiring manually tuned bias terms. We further show that our adjustment precisely re-centers the targeted token subset around the global mean, reducing over-dominant transition probabilities without hard blocking or suppressing model expressivity. Finally, global drift analysis demonstrates that this local correction does not distort the full-vocabulary logit distribution, preserving global scale, extrema, and distributional structure. These results support the proposed logit-gap distance as a robust, localized, and cross-model generalizable control metric.

*Table 17.* Summary of the GRU-based earliest-correctness predictor. We report the best-performing layer and the corresponding accuracy and loss for each model.

| Model | Best Layer | Acc ↑ | Loss ↓ |
|---|---|---|---|
| DeepSeek-R1-Distill-Qwen-7B | 4 | 0.9543 | 0.1351 |
| QwQ-32B | 27 | 0.9241 | 0.1774 |
| Qwen3-14B | 11 | 0.9353 | 0.1617 |
| Qwen3-4B-Thinking-2507 | 15 | 0.9444 | 0.1405 |

*Table 18.* **Raw logit-gap distance statistics across models.** Summary statistics of the mean positive gap $d_m$ and the max positive gap $d_M$ computed over the tracked token subset.

| Model | $d_m$ (mean positive gap) | | | | $d_M$ (max positive gap) | | | |
|---|---|---|---|---|---|---|---|---|
| | mean | std | p95 | max | mean | std | p95 | max |
| **DeepSeek-R1-Distill-Qwen-7B** | 3.7229 | 1.2337 | 5.5625 | 8.1875 | 11.2423 | 4.6427 | 20.0000 | 34.2500 |
| **QwQ-32B** | 3.6900 | 1.9258 | 7.4688 | 11.8750 | 15.4336 | 8.2416 | 30.8750 | 53.0000 |
| **Qwen3-14B** | 3.4822 | 1.8789 | 7.3125 | 14.6875 | 15.5645 | 8.6662 | 34.0000 | 68.0000 |
| **Qwen3-4B-Thinking-2507** | 3.7645 | 1.0638 | 5.5313 | 10.4375 | 13.7327 | 4.7214 | 22.0000 | 50.0000 |

**Logit-Gap Distance Definition.** Let $z_v \in \mathbb{R}$ denote the logit of token $v$ at a given decoding step, and let

$$\mu = \frac{1}{|\mathcal{V}|} \sum_{v \in \mathcal{V}} z_v \tag{26}$$

be the vocabulary-wide mean logit, where $\mathcal{V}$ is the vocabulary. For a token subset $\mathcal{B} \subseteq \mathcal{V}$, we define the per-token positive mean-centered logit gap as

$$\delta_v = (z_v - \mu)_+ = \max(z_v - \mu,\ 0), \qquad v \in \mathcal{B}. \tag{27}$$

**Aggregated Gap Statistics.** Based on $\{\delta_v\}_{v \in \mathcal{B}}$, we summarize the overall gap strength with two statistics:

$$d_m = \frac{1}{|\mathcal{B}|} \sum_{v \in \mathcal{B}} \delta_v, \tag{28}$$

which we refer to as the *mean positive gap* (average positive advantage), and

$$d_M = \max_{v \in \mathcal{B}} \delta_v, \tag{29}$$

which we refer to as the *max positive gap* (maximum positive advantage).

As shown in Table 18, the distance scale remains highly consistent across different models. This indicates that the proposed logit-gap distance defines a well-calibrated scale in logit space and does not require additional normalization.

Moreover, we observe that the extreme-value structure systematically strengthens with increasing model capacity, suggesting a meaningful monotonic relationship between the proposed distance and model capability. This property constitutes a key advantage of our distance over fixed bias-based heuristics, as it eliminates the need to manually tune model-specific bias terms.

Overall, the distance exhibits a two-scale structure, characterized by a stable central tendency and an expressive heavy tail, enabling strong cross-model generalization. These observations provide critical evidence supporting the validity and robustness of the proposed distance metric. We define $\sigma_0$ as the standard deviation of the full-vocabulary logit distribution at each decoding step, which provides an intrinsic measure of the model's instantaneous uncertainty. Formally, let $\{z_v\}_{v \in \mathcal{V}}$ denote the raw logits over the full vocabulary at a given step, and let

$$\sigma_0 = \sqrt{\frac{1}{|\mathcal{V}|} \sum_{v \in \mathcal{V}} (z_v - \mu_0)^2}, \tag{30}$$

*Table 19.* **Logit-gap distance normalized by global logit scale.** Summary statistics of the mean and max positive gaps normalized by the global logit standard deviation $\sigma_0$. The results demonstrate scale invariance of the proposed distance and confirm that large gaps correspond to statistically significant ($\sigma$-level) advantages.

| Model | $d_m/\sigma_0$ | | | | $d_M/\sigma_0$ | | | |
| --- | --- | --- | --- | --- | --- | --- | --- | --- |
| | **mean** | **std** | **p95** | **max** | **mean** | **std** | **p95** | **max** |
| Qwen3-4B-Thinking-2507 | 1.3257 | 0.3841 | 2.0093 | 3.2019 | 4.7956 | 1.5234 | 7.6044 | 14.2786 |
| Qwen3-14B | 1.3504 | 0.6001 | 2.5326 | 4.9683 | 6.0209 | 2.7988 | 11.9977 | 18.4881 |
| QwQ-32B | 1.4038 | 0.6409 | 2.5847 | 4.6849 | 5.8525 | 2.7308 | 10.7761 | 17.3124 |
| DeepSeek-R1-Distill-Qwen-7B | 1.9534 | 0.5249 | 2.7217 | 4.5152 | 5.8896 | 2.1642 | 9.9109 | 20.2415 |

*Table 20.* **Targeted-subset re-centering across models.** We report full distribution statistics for the subset offset $\Delta_{\mathcal{B}}^{\text{pre}}$ and $\Delta_{\mathcal{B}}^{\text{post}}$, as well as the fraction of subset logits above the global mean $\rho_{\mathcal{B}}^{\text{pre}}$ and $\rho_{\mathcal{B}}^{\text{post}}$.

| Model | Metric | mean | std | min | p50 | p90 | p95 | p99 | max |
| --- | --- | --- | --- | --- | --- | --- | --- | --- | --- |
| DeepSeek-R1-Distill-Qwen-7B | $\Delta_{\mathcal{B}}^{\text{pre}}$ | 3.732 | 1.185 | -1.648 | 3.838 | 5.165 | 5.555 | 6.276 | 7.879 |
| | $\Delta_{\mathcal{B}}^{\text{post}}$ | 0.0108 | 0.445 | -2.582 | -0.0125 | 0.00880 | 0.0149 | 3.320 | 6.153 |
| | $\rho_{\mathcal{B}}^{\text{pre}}$ | 0.9516 | 0.0678 | 0.279 | 0.9706 | 1.000 | 1.000 | 1.000 | 1.000 |
| | $\rho_{\mathcal{B}}^{\text{post}}$ | 0.5510 | 0.4108 | 0.000 | 0.7059 | 0.9853 | 1.000 | 1.000 | 1.000 |
| QwQ-32B | $\Delta_{\mathcal{B}}^{\text{pre}}$ | 3.4097 | 1.8954 | -1.790 | 3.1408 | 6.1190 | 7.0394 | 8.5345 | 11.7239 |
| | $\Delta_{\mathcal{B}}^{\text{post}}$ | -0.2787 | 0.4989 | -2.5604 | -0.2381 | -0.0576 | -0.0297 | 2.3850 | 8.6744 |
| | $\rho_{\mathcal{B}}^{\text{pre}}$ | 0.7803 | 0.1171 | 0.1912 | 0.7941 | 0.9118 | 0.9412 | 0.9706 | 1.0000 |
| | $\rho_{\mathcal{B}}^{\text{post}}$ | 0.4763 | 0.2570 | 0.0000 | 0.5000 | 0.7941 | 0.8529 | 0.9118 | 1.0000 |
| Qwen3-14B | $\Delta_{\mathcal{B}}^{\text{pre}}$ | 3.0661 | 1.8847 | -2.8160 | 2.7970 | 5.4719 | 6.8161 | 8.9249 | 13.9850 |
| | $\Delta_{\mathcal{B}}^{\text{post}}$ | -0.4145 | 0.6078 | -3.1722 | -0.3260 | -0.0846 | -0.0515 | 2.4994 | 9.5910 |
| | $\rho_{\mathcal{B}}^{\text{pre}}$ | 0.7379 | 0.1371 | 0.0441 | 0.7647 | 0.8971 | 0.9265 | 0.9559 | 1.0000 |
| | $\rho_{\mathcal{B}}^{\text{post}}$ | 0.4373 | 0.2686 | 0.0000 | 0.4706 | 0.7794 | 0.8235 | 0.8824 | 0.9706 |
| Qwen3-4B-Thinking-2507 | $\Delta_{\mathcal{B}}^{\text{pre}}$ | 3.5193 | 1.1133 | -1.5780 | 3.7297 | 4.7595 | 5.2984 | 6.3913 | 9.5122 |
| | $\Delta_{\mathcal{B}}^{\text{post}}$ | -0.2434 | 0.4956 | -3.5103 | -0.1373 | -0.0522 | -0.0272 | 0.0093 | 8.2898 |
| | $\rho_{\mathcal{B}}^{\text{pre}}$ | 0.8419 | 0.1088 | 0.2941 | 0.8824 | 0.9412 | 0.9559 | 0.9853 | 1.0000 |
| | $\rho_{\mathcal{B}}^{\text{post}}$ | 0.4883 | 0.3154 | 0.0000 | 0.5735 | 0.8676 | 0.8971 | 0.9412 | 1.0000 |

where $\mu_0 = \frac{1}{|\mathcal{V}|} \sum_{v \in \mathcal{V}} z_v$ is the mean logit. As shown in Table 19, after accounting for the step-wise global uncertainty, the proposed distance remains well-structured and tightly concentrated, without exhibiting collapse or explosion. This demonstrates strong stability across decoding steps and varying uncertainty levels. Except for DeepSeek (which exhibits distinct distillation-specific characteristics), the remaining model families show highly consistent normalized distances, indicating that the proposed metric naturally adapts to each model's intrinsic uncertainty. This property constitutes a key advantage over fixed bias-based heuristics, as it eliminates the need for manual tuning of model-specific bias terms.

Let $\mathcal{B} \subseteq \mathcal{V}$ denote the targeted token subset and let $\mu$ denote the mean of the full-vocabulary logits at a given decoding step. We define the subset mean logits before and after adjustment as

$$\bar{z}_{\mathcal{B}}^{\text{pre}} = \frac{1}{|\mathcal{B}|} \sum_{v \in \mathcal{B}} z_v^{\text{pre}}, \qquad \bar{z}_{\mathcal{B}}^{\text{post}} = \frac{1}{|\mathcal{B}|} \sum_{v \in \mathcal{B}} z_v^{\text{post}}. \tag{31}$$

We then define the relative subset offsets as

$$\Delta_{\mathcal{B}}^{\text{pre}} = \bar{z}_{\mathcal{B}}^{\text{pre}} - \mu, \qquad \Delta_{\mathcal{B}}^{\text{post}} = \bar{z}_{\mathcal{B}}^{\text{post}} - \mu. \tag{32}$$

Finally, we define the fraction of subset logits above the global mean as

$$\rho_{\mathcal{B}}^{\text{pre}} = \frac{1}{|\mathcal{B}|} \sum_{v \in \mathcal{B}} \mathbb{I}[z_v^{\text{pre}} > \mu], \qquad \rho_{\mathcal{B}}^{\text{post}} = \frac{1}{|\mathcal{B}|} \sum_{v \in \mathcal{B}} \mathbb{I}[z_v^{\text{post}} > \mu]. \tag{33}$$

As shown in Table 20, all models exhibit a substantial positive subset offset prior to control, indicating that the targeted subset is consistently ranked above the global mean. After applying our adjustment, the subset is precisely re-centered around zero. Importantly, this is not achieved through a global shift, but via fine-grained local correction.

*Table 21.* **Global logit distribution drift across models.** Full distribution statistics for drift components ($\Delta\mu_t$, $\Delta\sigma_t$, $\Delta m_t$, $\Delta M_t$) and the composite score $D_t^{\text{global}} = |\Delta\mu_t| + |\Delta\sigma_t|$.

| Model | Metric | mean | std | min | p50 | p90 | p95 | p99 | max |
|---|---|---|---|---|---|---|---|---|---|
| DeepSeek-R1-Distill-Qwen-7B | $\Delta\mu_t$ | -0.00166479 | 0.00055168 | -0.00367117 | -0.00172186 | -0.000929838 | -0.000774753 | 0 | 0 |
| | $\Delta\sigma_t$ | -0.00248784 | 0.00134572 | -0.0151528 | -0.00241697 | -0.00090766 | -0.000679517 | 0 | 0 |
| | $\Delta m_t$ | 0 | 0 | 0 | 0 | 0 | 0 | 0 | 0 |
| | $\Delta M_t$ | -0.0144071 | 0.266059 | -15.25 | 0 | 0 | 0 | 0 | 0 |
| | $D_t^{\text{global}}$ | 0.00415263 | 0.00185214 | 0 | 0.00415158 | 0.0063448 | 0.00717585 | 0.00944375 | 0.018824 |
| QwQ-32B | $\Delta\mu_t$ | -0.00165009 | 0.000861187 | -0.00530505 | -0.00150046 | -0.000657105 | -0.00050931 | 0 | 0 |
| | $\Delta\sigma_t$ | -0.00290258 | 0.00273083 | -0.0190656 | -0.00192714 | -0.000445461 | -0.000306606 | 0 | 0 |
| | $\Delta m_t$ | 0 | 0 | 0 | 0 | 0 | 0 | 0 | 0 |
| | $\Delta M_t$ | -0.0817131 | 0.611305 | -17.25 | 0 | 0 | 0 | 0 | 0 |
| | $D_t^{\text{global}}$ | 0.00455267 | 0.0035652 | 0 | 0.00343448 | 0.0101979 | 0.012361 | 0.0154463 | 0.0243707 |
| Qwen3-14B | $\Delta\mu_t$ | -0.00155845 | 0.000840938 | -0.00658393 | -0.0014329 | -0.000642896 | -0.000498146 | 0 | 0 |
| | $\Delta\sigma_t$ | -0.00274806 | 0.00277262 | -0.026582 | -0.00190973 | -0.000510097 | -0.000351906 | 0 | 0 |
| | $\Delta m_t$ | 0 | 0 | 0 | 0 | 0 | 0 | 0 | 0 |
| | $\Delta M_t$ | -0.0567881 | 0.565262 | -25 | 0 | 0 | 0 | 0 | 0 |
| | $D_t^{\text{global}}$ | 0.00430652 | 0.00356723 | 0 | 0.00335801 | 0.00848454 | 0.0123584 | 0.0180068 | 0.0331659 |
| Qwen3-4B-Thinking-2507 | $\Delta\mu_t$ | -0.00168482 | 0.000476121 | -0.00466251 | -0.00174904 | -0.00110273 | -0.000955227 | -0.000542369 | 0 |
| | $\Delta\sigma_t$ | -0.00205793 | 0.00112011 | -0.0142033 | -0.00181937 | -0.000981569 | -0.000702155 | -0.000281575 | 0 |
| | $\Delta m_t$ | 0 | 0 | 0 | 0 | 0 | 0 | 0 | 0 |
| | $\Delta M_t$ | -0.0203379 | 0.446352 | -21.625 | 0 | 0 | 0 | 0 | 0 |
| | $D_t^{\text{global}}$ | 0.00374275 | 0.00155197 | 0 | 0.00358558 | 0.00549715 | 0.00649142 | 0.00923532 | 0.0188324 |

The targeted tokens are still permitted to appear; however, their relative ranking is softened such that their average probability mass is reduced from being significantly above $0.5$ to approximately neutral. Compared to hard semantic suppression or blocking, our approach constitutes a substantially softer intervention that preserves model expressivity while effectively controlling over-dominant transitions.

Next, we investigate whether such adjustments induce any unintended changes to the global logit distribution. Let $\mathbf{z}_t^{(0)} = \{z_{t,v}^{(0)}\}_{v=1}^V$ and $\mathbf{z}_t^{(1)} = \{z_{t,v}^{(1)}\}_{v=1}^V$ denote the full-vocabulary logits at decoding step $t$ before and after adjustment, respectively. We define the global mean and standard deviation as

$$\mu_t^{(k)} = \frac{1}{V}\sum_{v=1}^{V} z_{t,v}^{(k)}, \qquad \sigma_t^{(k)} = \sqrt{\frac{1}{V}\sum_{v=1}^{V}\left(z_{t,v}^{(k)} - \mu_t^{(k)}\right)^2}, \quad k \in \{0,1\}. \tag{34}$$

We further define the global extrema as

$$m_t^{(k)} = \min_v z_{t,v}^{(k)}, \qquad M_t^{(k)} = \max_v z_{t,v}^{(k)}, \quad k \in \{0,1\}. \tag{35}$$

The corresponding drift components are defined by

$$\Delta\mu_t = \mu_t^{(1)} - \mu_t^{(0)}, \qquad \Delta\sigma_t = \sigma_t^{(1)} - \sigma_t^{(0)}, \qquad \Delta m_t = m_t^{(1)} - m_t^{(0)}, \qquad \Delta M_t = M_t^{(1)} - M_t^{(0)}. \tag{36}$$

As a scalar summary of potential global distributional side effects, we define the global drift score as

$$\boxed{D_t^{\text{global}} = \left|\Delta\mu_t\right| + \left|\Delta\sigma_t\right|.} \tag{37}$$

As shown in Table 21, our method does not induce harmful changes to the global logit distribution. The extrema structure is preserved, with no evidence of scale collapse or explosion, and no global translation of the distribution. In contrast, for the targeted tokens subject to local adjustment, the proposed distance effectively reduces their relative advantage in ranking. Overall, these results demonstrate that our approach performs safe, localized control with strong generalization across model architectures and data distributions.

*Table 22.* Ablation on $\mu_t$ for **DeepSeek-R1-Distill-Qwen-7B**.

| Setting | Math-500 Pass@1↑ | #Tok↓ | AIME2024 Pass@1↑ | #Tok↓ | AIME2025 Pass@1↑ | #Tok↓ | AMC23 Pass@1↑ | #Tok↓ | GSM8K Pass@1↑ | #Tok↓ | MMLU Pass@1↑ | #Tok↓ |
|---------|------------------|-------|------------------|-------|------------------|-------|---------------|-------|---------------|-------|--------------|-------|
| Baseline | 92.0 | 3955 | 50.0 | 13008 | 36.7 | 15245 | 87.5 | 6193 | 90.6 | 1214 | 90.0 | 2387 |
| ABLATE $\mu_t$ | 90.4 | 2699 | 43.3 | 10107 | 26.7 | 9061 | 85.0 | 3956 | 90.5 | 788 | 89.0 | 1595 |

*Table 23.* **Avg@30 performance with standard deviation on mathematical reasoning benchmarks.**

| Method | AIME2024 Avg@30↑ | Std | #Tok↓ | Std | AIME2025 Avg@30↑ | Std | #Tok↓ | Std | AMC23 Avg@30↑ | Std | #Tok↓ | Std |
|--------|------------------|-----|-------|-----|------------------|-----|-------|-----|---------------|-----|-------|-----|
| **DeepSeek-R1-Distill-Qwen-7B** | | | | | | | | | | | | |
| Baseline | 53.1 | 0.0643 | 13358 | 1205 | 37.8 | 0.0433 | 14471 | 1175 | 90.1 | 0.0313 | 6243 | 646 |
| Ours | 54.7 | 0.04605 | 10183 | 981 | 37.9 | 0.0459 | 11239 | 971 | 91.2 | 0.0294 | 3741 | 480 |
| **Qwen3-4B-Thinking-2507** | | | | | | | | | | | | |
| Baseline | 83.1 | 0.0384 | 21279 | 657 | 80.2 | 0.0501 | 22553 | 958 | 99.8 | 0.0062 | 11145 | 391 |
| Ours | 85.3 | 0.0372 | 18632 | 641 | 82.2 | 0.0398 | 19820 | 863 | 99.9 | 0.0046 | 9034 | 381 |

## B.4. Ablation on $\mu_t$

**Summary.** In this section, we ablate the vocabulary-wide mean logit term $\mu_t$ to examine its role in defining a calibrated suppression distance. When $\mu_t$ is removed, the distance signal is computed directly from raw logits, effectively measuring each targeted token's distance to zero rather than its relative advantage over the global logit distribution. This produces an overly large and poorly calibrated suppression signal. Even when modulated by the difficulty regressor, the resulting control remains too aggressive, substantially reducing token usage but causing clear accuracy degradation, especially on harder benchmarks such as AIME2024 and AIME2025. These results demonstrate that the $\mu_t$ term is essential for converting raw logits into a relative, distribution-aware distance, enabling localized and appropriately scaled suppression rather than crude truncation.

As shown in Table 22, we ablate the entire $\mu_t$ term and retain only the raw logits as the distance signal, with a regressor used to modulate the suppression strength. The results indicate that this distance is overly large, as it corresponds to the linear distance of the logits from their original values to zero. Even with regressor-based scaling, the resulting suppression remains excessively strong, causing the model to degenerate toward conventional efficient reasoning methods with aggressive truncation.

## B.5. Avg@K Performance Analysis

**Summary.** In this section, we evaluate the robustness of our method on small-scale mathematical reasoning benchmarks using Avg@30 with standard deviations. The results show that our approach consistently reduces token consumption across AIME2024, AIME2025, and AMC23 while maintaining or even improving average accuracy for both DeepSeek-R1-Distill-Qwen-7B and Qwen3-4B-Thinking-2507. These gains indicate that the proposed control mechanism improves reasoning efficiency without sacrificing performance, even under repeated sampling evaluation. At the same time, the relatively large standard deviations observed on small benchmarks highlight the importance of Avg@30 evaluation, as single-run results may be unstable and insufficient for reliably characterizing performance on limited test sets.

As shown in Table 23, we further evaluate the Avg@30 performance of our method on small-scale benchmarks. We find that our method is able to consistently reduce token consumption while preserving, and in some cases even improving, accuracy. We also observe relatively large standard deviations on small datasets, which highlights the necessity of Avg@30 evaluation for providing a more reliable assessment.

## B.6. Time Latency Analysis

**Summary.** In this section, we analyze the inference efficiency of our method on MMLU using DeepSeek-R1-Distill-Qwen-7B. As a lightweight and training-free approach, our method achieves strong latency efficiency while maintaining high throughput comparable to prompt-based baselines. Unlike rollback- or trial-and-error-based methods such as DEER and Dynasor-CoT, our approach avoids repeated decoding, leading to more efficient inference. It also does not rely on auxiliary models, unlike TrimR and FlashThinking, thereby avoiding additional memory overhead. Empirically, our method

*Table 24.* Overall comparison of accuracy and inference efficiency on **MMLU** using **DeepSeek-R1-Distill-Qwen-7B**. Lower is better for time and tokens, while higher is better for Pass@1 and throughput.

| Method | Pass@1(%)↑ | Avg. Tokens↓ | Time Per Request (s)↓ | Tokens Per Second↑ |
|---|---|---|---|---|
| Baseline | 90.0 | 2387 | 33.20 | 68.08 |
| CoD | 85.0 | 1091 | 15.22 | 65.49 |
| DEER | 79.0 | 1493 | 24.77 | 60.82 |
| Ours | **91.0** | 1488 | 22.57 | **65.18** |

*Table 25.* Robustness of DyCon under different reflection-token vocabularies. We report Pass@1 and average output tokens in the format of Pass@1 / Tok.

| Model | Method | Math-500 | AIME24 | AIME25 |
|---|---|---|---|---|
| DeepSeek-R1-Distill-Qwen-7B | Baseline | 92.0 / 3955 | 50.0 / 13008 | 36.7 / 15245 |
| | +DyCon | 92.0 / 3216 | 53.3 / 10906 | 36.7 / 12415 |
| | +DyCon-SEAL | 92.2 / 3569 | 56.7 / 11141 | 40.0 / 12775 |
| Qwen3-4B-Thinking-2507 | Baseline | 96.2 / 6749 | 83.3 / 21493 | 76.7 / 22708 |
| | +DyCon | 96.2 / 6092 | 86.7 / 18867 | 76.7 / 21100 |
| | +DyCon-SEAL | 96.4 / 5753 | 83.3 / 18056 | 83.3 / 19147 |

reduces average token usage and latency relative to the baseline, while achieving the best Pass@1 among compared methods. These results demonstrate that our approach offers a favorable balance between accuracy, token efficiency, latency, and deployment simplicity.

We analyze the inference latency of our method on **DeepSeek-R1-Distill-Qwen-7B** evaluated on **MMLU**. As shown in Table 24, as a lightweight, training-free approach, our method achieves throughput comparable to prompt-based baselines, while substantially outperforming lightweight alternatives such as CoD (Xu et al., 2025a) in terms of accuracy. In contrast to methods that rely on iterative rollback or trial-and-error strategies (e.g., DEER (Yang et al., 2025b) and Dynasor-CoT (Fu et al., 2025)), our approach avoids repeated decoding and therefore yields significantly better latency efficiency. Moreover, unlike TrimR (Lin et al., 2025a) and FlashThinking (Jiang et al., 2025), our method does not require any additional auxiliary models, and thus incurs no extra memory overhead.

### B.7. Analysis of Reflection Token Sensitivity

**Summary.** In this section, we further investigate the choice of reflection-token vocabulary. Our goal is to examine whether DyCon depends on a particular manually predefined token list, or whether its effectiveness is preserved under alternative choices of reflection-related tokens. We show that DyCon is not tied to a specific token set: replacing the original vocabulary with the token set proposed by SEAL (Chen et al., 2025) leads to comparable performance across models and benchmarks. We further study whether the reflection-token vocabulary can be optimized in a model-specific manner, and find that an evolutionary refinement strategy can yield additional improvements in both accuracy and inference efficiency.

The predefined reflection-token list used in our main experiments is not an essential component of DyCon. Instead, it serves as a conventional instantiation following prior work on manipulating thinking or reflection-related tokens (Wang et al., 2025a). Conceptually, DyCon only requires a set of tokens that approximately correspond to reflective reasoning behaviors, since its control mechanism operates by dynamically modulating the logits of such tokens. Therefore, the method does not rely on any particular handcrafted vocabulary, but rather on the broader principle that reflection-related token probabilities can be adjusted to regulate the model's reasoning behavior.

To examine the sensitivity of DyCon to different token-set choices, we replace our initial reflection-token list with the token set proposed by SEAL (Chen et al., 2025). As shown in Table 25, DyCon achieves comparable performance under this alternative vocabulary across different models and benchmarks. In several cases, the SEAL-based token set even leads to slightly higher Pass@1 or lower average token consumption. These results suggest that DyCon is robust to reasonable choices of reflection-token vocabularies, and that its gains do not come from overfitting to a particular manually selected token list.

Beyond robustness to existing token sets, we further investigate whether a more suitable reflection-token vocabulary can

*Table 26.* Performance of DyCon with an optimized reflection-token set using **Qwen3-4B-Thinking-2507**. We report Pass@1 and average output tokens in the format of Pass@1 / Tok.

| Method | Math-500 | AIME24 | AIME25 |
|---|---|---|---|
| Baseline | 96.2 / 6749 | 83.3 / 21493 | 76.7 / 22708 |
| DyCon-Optimized | **96.6 / 5898** | **86.7 / 19136** | **83.3 / 16848** |

be automatically identified for a specific model. This is motivated by the observation that different reasoning models may express reflection through slightly different lexical patterns. Therefore, while a general reflection-token list is sufficient for DyCon to be effective, a model-adaptive token set may further improve the controllability of the reasoning process.

Specifically, we adopt an evolutionary strategy initialized with frequently occurring reflection-related tokens. Inspired by dynamic context learning (Li et al., 2025a), the search process iteratively applies token selection, mutation, and crossover. We use model accuracy on a held-out Math validation set as the optimization signal, so that the resulting token set is selected according to its downstream effect on reasoning performance rather than by manual inspection alone.

As reported in Table 26, the optimized reflection-token set further improves DyCon on Qwen3-4B-Thinking-2507. Compared with the baseline, DyCon-Optimized improves Pass@1 on Math-500, AIME24, and AIME25, while also reducing the average number of generated tokens. These results indicate that although DyCon is already robust to different reasonable token vocabularies, model-specific token refinement can provide additional benefits. This also suggests a promising direction for future work: instead of relying on manually designed reflection-token lists, one can develop more principled optimization objectives and search strategies to automatically discover effective control vocabularies.

The optimized token set further improves both accuracy and reasoning efficiency compared with the original list. This suggests that although DyCon is not sensitive to a specific predefined vocabulary, model-specific token refinement can still provide additional benefits. Designing more principled optimization objectives and more advanced token-selection strategies remains a promising direction for future work.

### B.8. Analysis of Noisy Difficulty Proxy

**Summary.** In this section, we provide a detailed analysis of the difficulty proxy used in DyCon. Since fine-grained dynamic difficulty labels are generally unavailable, DyCon uses generation length as a practical proxy for model-perceived reasoning difficulty. We first analyze the stability of this proxy through an outlier study over generation lengths across different difficulty levels. As shown in Table 27, length-based outliers account for only a small proportion of samples, suggesting that the overall distributional signal is stable. We then conduct a complementary Pass@1-based analysis by grouping samples according to output length and measuring the correlation between group-level length and accuracy. As reported in Table 28, longer generations are consistently associated with lower Pass@1, providing further evidence that generation length captures meaningful difficulty-related information.

Reasoning difficulty is inherently dynamic during generation. A problem may appear easy at the beginning but become harder when the model encounters intermediate uncertainty, or conversely become easier after a key reasoning step is resolved. However, most existing datasets only provide static and coarse-grained difficulty annotations, such as the five discrete difficulty levels in MATH. These annotations are useful for interpretability, but they cannot fully describe the model's evolving perception of difficulty during the reasoning process. Therefore, rather than relying on exact difficulty labels, DyCon exploits a statistically meaningful proxy that can be observed during generation.

We use output length as such a proxy. The intuition is that when a model perceives a problem as more difficult, it typically spends more tokens exploring intermediate steps, verifying partial results, correcting mistakes, or searching for alternative reasoning paths. This does not imply that every long response is necessarily difficult or every short response is necessarily easy. Instead, the claim is distributional: across a sufficiently large set of samples, generation length provides a useful signal for estimating model-perceived difficulty.

To quantify the stability of this signal, we conduct an outlier analysis based on generation length. For each model and each MATH difficulty level, we compute the mean generation length, the first quartile $q_{25}$, the third quartile $q_{75}$, and the interquartile range. We define length outliers as samples whose generation length lies outside the standard IQR interval:

$$[q_{25} - 1.5 \cdot \text{IQR}, \ q_{75} + 1.5 \cdot \text{IQR}], \quad \text{where} \quad \text{IQR} = q_{75} - q_{25}. \tag{38}$$

*Table 27.* Outlier analysis of generation length across MATH difficulty levels. For each model and difficulty level, we report the mean generation length, the first quartile, the third quartile, and the outlier ratio computed using the IQR rule.

| Model | Level | Mean | $q_{25}$ | $q_{75}$ | Outlier Ratio |
|---|---|---|---|---|---|
| Qwen3-4B-Thinking-2507 | 1 | 2121 | 903 | 2088 | 0.10 |
| | 2 | 3068 | 1112 | 4454 | 0.03 |
| | 3 | 5322 | 1839 | 7155 | 0.03 |
| | 4 | 6840 | 2930 | 9327 | 0.02 |
| | 5 | 12025 | 6356 | 15386 | 0.04 |
| DeepSeek-R1-Distill-Qwen-7B | 1 | 1932 | 1159 | 2162 | 0.05 |
| | 2 | 2230 | 1302 | 2600 | 0.06 |
| | 3 | 2953 | 1570 | 3149 | 0.10 |
| | 4 | 3482 | 1721 | 3866 | 0.09 |
| | 5 | 5580 | 2551 | 6941 | 0.09 |
| Qwen3-14B | 1 | 1978 | 1324 | 2114 | 0.08 |
| | 2 | 2806 | 1493 | 3022 | 0.09 |
| | 3 | 3570 | 1987 | 4243 | 0.06 |
| | 4 | 4610 | 2429 | 5568 | 0.05 |
| | 5 | 7673 | 3846 | 9326 | 0.06 |
| QwQ-32B | 1 | 1751 | 1191 | 2015 | 0.07 |
| | 2 | 2297 | 1338 | 2832 | 0.04 |
| | 3 | 3303 | 1757 | 4060 | 0.05 |
| | 4 | 4311 | 2045 | 5196 | 0.07 |
| | 5 | 7054 | 3775 | 8374 | 0.06 |

Table 27 reports the statistics for four representative reasoning models. Across models and difficulty levels, the outlier ratio remains relatively small, indicating that the length distribution is not dominated by rare abnormal generations. More importantly, the mean generation length generally increases with the annotated difficulty level, supporting the use of length as a stable aggregate signal.

The results in Table 27 suggest that generation length provides a stable distribution-level signal. Nevertheless, correlation with the manually annotated MATH difficulty levels is limited by the coarse granularity of the labels. The MATH dataset uses only five discrete levels, whereas reasoning length is a continuous variable with substantial natural variance. As a result, a moderate Spearman correlation with static difficulty labels does not necessarily imply that generation length is a weak proxy. It may instead reflect a mismatch between a coarse human annotation scheme and the model's continuous, instance-specific perception of difficulty.

To obtain a more direct measure of difficulty, we further analyze the relationship between generation length and Pass@1. Pass@1 reflects whether the model solves a problem correctly, and thus provides a performance-based view of problem difficulty. We group samples by output length and compute the average length and Pass@1 within each group. For the $k$-th length group $B_k$, we compute:

$$\bar{l}_k = \frac{1}{|B_k|} \sum_{i \in B_k} l_i, \quad \text{Pass@1}(B_k) = \frac{1}{|B_k|} \sum_{i \in B_k} y_i. \tag{39}$$

where $l_i$ denotes the output length of sample $i$, and $y_i$ is a binary correctness indicator. We then compute the Pearson and Spearman correlations between $\bar{l}_k$ and $\text{Pass@1}(B_k)$. For the overall results, we merge samples from all evaluated datasets and compute the correlation on the combined set.

As shown in Table 28, the correlation between grouped output length and Pass@1 is consistently negative across datasets, models, and different group sizes. This indicates that longer generations are generally associated with lower accuracy, which is consistent with the interpretation that longer reasoning often reflects higher model-perceived difficulty. The correlations remain strong under different choices of $|B_k|$, showing that the trend is not an artifact of a particular grouping resolution.

Overall, the analyses in Table 27 and Table 28 provide complementary evidence for using generation length as a proxy for model-perceived difficulty. The outlier analysis shows that the signal is stable at the distribution level, while the Pass@1-based analysis shows that the signal is strongly associated with actual model performance. These results support

*Table 28.* Correlation between grouped output length and Pass@1. Samples are grouped by generation length, and Pearson/Spearman correlations are computed between the average length and Pass@1 of each group. Negative values indicate that longer generations are associated with lower accuracy.

| Model | $|B_k|$ | GPQA | StrategyQA | AIME2024 | AIME2025 | Math500 | Olympiad | Overall |
|---|---|---|---|---|---|---|---|---|
| DeepSeek-R1-Distill-Qwen-7B | 5 | -0.93/-0.90 | -0.97/-0.99 | -0.98/-0.99 | -0.93/-0.99 | -0.96/-0.86 | -0.98/-0.99 | -0.99/-0.99 |
| | 10 | -0.90/-0.89 | -0.96/-0.96 | -0.94/-0.95 | -0.90/-0.94 | -0.94/-0.84 | -0.98/-0.95 | -0.99/-0.97 |
| | 20 | -0.82/-0.81 | -0.93/-0.91 | -0.91/-0.84 | -0.82/-0.83 | -0.90/-0.81 | -0.95/-0.89 | -0.99/-0.90 |
| Qwen3-4B-Thinking-2507 | 5 | -0.96/-0.99 | -0.99/-0.99 | -0.98/-0.97 | -0.95/-0.97 | -0.94/-0.83 | -0.98/-0.97 | -0.99/-0.99 |
| | 10 | -0.90/-0.94 | -0.99/-0.97 | -0.83/-0.87 | -0.85/-0.94 | -0.90/-0.81 | -0.94/-0.95 | -0.91/-0.97 |
| | 20 | -0.86/-0.88 | -0.96/-0.98 | -0.81/-0.83 | -0.82/-0.92 | -0.86/-0.81 | -0.92/-0.89 | -0.87/-0.89 |

the design choice of DyCon: when explicit dynamic difficulty annotations are unavailable, generation length provides a practical, observable, and empirically grounded supervision signal for learning difficulty-aware control.

This analysis also clarifies the role of the length-based proxy. DyCon does not assume that output length is a perfect difficulty label for every individual sample. Instead, it uses length as a scalable statistical signal that reflects the model's reasoning effort in aggregate. Developing more precise supervision for dynamic difficulty estimation remains an important direction for future work, but the current evidence suggests that generation length is already a reliable and useful proxy for difficulty-aware reasoning control.

### B.9. Analysis of Cross-Lingual Generalization

**Summary.** In this section, we analyze whether DyCon can be applied across different languages. Although DyCon modulates reflection-related tokens during generation, the method does not require a fixed English-only token list. For each target language, we replace the original reflection-token vocabulary with a concise set of reflection-related tokens in that language, while keeping the difficulty regressor unchanged. We evaluate this setting on MGSM (Shi et al., 2022) using Qwen3-4B-Thinking-2507. As shown in Table 29, DyCon consistently reduces token usage across English, Chinese, French, German, and Japanese, while maintaining comparable or slightly improved accuracy. We further examine whether an English-fitted difficulty regressor produces similar difficulty estimates on non-English inputs. As reported in Table 30, the predicted difficulty scores on English and Chinese are close to their corresponding ground-truth scores, suggesting that similar difficulty-estimation behavior can emerge across languages.

DyCon contains two components that are relevant to cross-lingual transfer: the difficulty estimator and the reflection-token vocabulary. The difficulty estimator predicts the model's current reasoning difficulty from internal representations, while the reflection-token vocabulary determines which token logits are modulated during generation. The second component is naturally language-dependent, since different languages express reflective reasoning through different surface forms. Therefore, when applying DyCon to a new language, we only replace the reflection-token list with a small set of language-specific reflection-related tokens, without modifying the difficulty regressor.

This setting allows us to test whether DyCon can retain its efficiency benefits under multilingual generation. Importantly, we do not perform additional tuning, refitting, or language-specific calibration for the regressor. The only adaptation is the substitution of the reflection-token vocabulary. Therefore, the results reflect whether the original difficulty estimator can provide a useful control signal when paired with appropriate reflection-token mappings in different languages.

Table 29 reports the multilingual results on MGSM. Across all evaluated languages, DyCon consistently reduces the average number of generated tokens. For English, DyCon improves Pass@1 from 95.6 to 96.8 while reducing the average token count from 1483 to 1116. For Chinese, French, German, and Japanese, DyCon preserves the baseline accuracy while substantially reducing token usage. These results indicate that DyCon can be effectively extended to multilingual reasoning tasks by adapting the reflection-token vocabulary.

To further understand this behavior, we separately analyze the difficulty estimator. Specifically, we evaluate whether a regressor fitted on English data can produce reasonable difficulty estimates when applied to Chinese inputs. Table 30 compares the regressor-predicted difficulty scores with the corresponding ground-truth difficulty scores on English and Chinese. The predicted score is 0.50 for English and 0.47 for Chinese, which closely matches the ground-truth scores of 0.51 and 0.46, respectively. This suggests that, at least empirically, the English-fitted regressor can produce difficulty estimates

*Table 29.* Cross-lingual evaluation of DyCon on MGSM using Qwen3-4B-Thinking-2507. We report Pass@1 and average output tokens in the format of Pass@1 / Tok.

| Method | English | Chinese | French | German | Japanese |
|--------|---------|---------|--------|--------|----------|
| Baseline | 95.6 / 1483 | 90.8 / 1235 | 89.6 / 2284 | 91.0 / 2470 | 89.6 / 2482 |
| DyCon | 96.8 / 1116 | 90.8 / 1053 | 89.8 / 1731 | 91.0 / 1590 | 89.6 / 1823 |

*Table 30.* Cross-lingual evaluation of the English-fitted difficulty regressor. We report the mean regressor-predicted difficulty scores and the corresponding ground-truth difficulty scores on English and Chinese.

| Type | English | Chinese |
|------|---------|---------|
| Regressor Prediction | 0.50 | 0.47 |
| Ground Truth | 0.51 | 0.46 |

on Chinese that are similar to the corresponding ground-truth difficulty values.

Overall, Table 29 shows that DyCon can reduce reasoning length across multiple languages without degrading accuracy, while Table 30 provides preliminary evidence that English and Chinese exhibit similar difficulty-estimation behavior under the same regressor. We do not claim that the underlying difficulty estimator is theoretically language-agnostic. Rather, the empirical results suggest that there may exist a cross-lingual correspondence between difficulty representations in different languages, and that an appropriate reflection-token mapping may be sufficient for DyCon to transfer across languages in practice. Formalizing this correspondence and developing a principled theory of cross-lingual difficulty estimation remain promising directions for future research.

### B.10. Analysis of Regressor Refinement

**Summary.** In this section, we analyze the effect of the fit–refine–refit procedure on difficulty estimation and downstream DyCon performance. The key question is whether the improvement from refinement simply comes from removing redundant or noisy trajectories. Our results suggest that this is not the case. As shown in Table 31, directly removing length-based outlier trajectories does not improve performance and can even degrade accuracy on challenging benchmarks. In contrast, moderate trajectory refinement improves both regressor quality and downstream inference efficiency, as shown in Table 32. However, this improvement is not monotonic: excessive refinement further increases the regressor's $R^2$ but hurts downstream performance. These results indicate that refinement should be understood as a controlled reshaping of the reasoning-trajectory distribution rather than simple denoising.

The difficulty regressor in DyCon is fitted on reasoning trajectories generated by the base model. Therefore, the quality and distribution of these trajectories directly influence what kind of difficulty signal the regressor learns. A natural hypothesis is that long or atypical trajectories may introduce noise, and that removing such trajectories should improve difficulty estimation. However, this interpretation is overly simplistic. Reasoning trajectories with unusually long generations are not necessarily invalid or harmful; they may correspond to harder problems, failed attempts, self-corrections, or atypical reasoning patterns that are important for modeling the full behavior of the base model.

To test whether removing such trajectories is beneficial, we conduct an IQR-based outlier removal experiment. Specifically, we remove length-based outlier trajectories before fitting the difficulty regressor and then evaluate DyCon on downstream benchmarks. Table 31 reports the results. Compared with standard DyCon, removing outliers slightly reduces the average token count on Math-500 but lowers Pass@1. More importantly, it substantially degrades performance on AIME24, where Pass@1 drops from 86.7 to 80.0. This suggests that outlier trajectories are not merely noise. Instead, they may contain informative examples of complex or atypical reasoning behavior. This observation is also consistent with prior findings that abnormal or heavy-tailed samples can carry important learning signals rather than being reducible to simple noise (Gurbuzbalaban et al., 2021).

The results in Table 31 show that refinement should not be viewed as a procedure for simply discarding noisy samples. Instead, the fit–refine–refit procedure modifies the structure of the reasoning trajectories while preserving their connection to the original model behavior. After an initial DyCon pass, the generated trajectories tend to become more concise and structured. Such trajectories may provide a clearer supervision signal for fitting the difficulty regressor, because the remaining reasoning steps are less dominated by unnecessary repetition while still reflecting the model's problem-solving

*Table 31.* Effect of removing length-based outlier trajectories before fitting the difficulty regressor. We report Pass@1 and average output tokens in the format of Pass@1 / Tok. Removing outliers does not consistently improve performance and can hurt accuracy on challenging benchmarks.

| Method | Math-500 | AIME24 | AIME25 |
|---|---|---|---|
| Qwen3-4B-Thinking-2507 | 96.2 / 6749 | 83.3 / 21493 | 76.7 / 22708 |
| + DyCon | 96.2 / 6092 | 86.7 / 18867 | 76.7 / 21100 |
| + DyCon (without outliers) | 96.0 / 5929 | 80.0 / 20794 | 76.7 / 20038 |

*Table 32.* Effect of iterative fit–refine–refit. We report the regressor $R^2$, Pass@1, and average output tokens. Moderate refinement improves both regressor quality and downstream performance, while excessive refinement increases $R^2$ but degrades generalization.

| Method | $R^2$ | Math-500 | AIME24 | AIME25 |
|---|---|---|---|---|
| Qwen3-4B-Thinking-2507 | – | 96.2 / 6749 | 83.3 / 21493 | 76.7 / 22708 |
| + DyCon | 0.8008 | 96.2 / 6092 | 86.7 / 18867 | 76.7 / 21100 |
| + DyCon (Iteration 2) | 0.9073 | 96.6 / 5567 | 86.7 / 17156 | 80.0 / 19351 |
| + DyCon (Iteration 3) | 0.9276 | 96.2 / 5710 | 86.7 / 19051 | 73.3 / 19726 |

process. This is consistent with prior work showing that shorter but complete reasoning traces can serve as effective learning signals (Wu et al., 2025).

To further understand this effect, we evaluate multiple refinement iterations. Table 32 reports the regressor $R^2$ and downstream performance after successive refinement rounds. The second iteration improves the regressor $R^2$ from 0.8008 to 0.9073 and also improves downstream results: Math-500 increases from 96.2 to 96.6, AIME25 increases from 76.7 to 80.0, and token usage is further reduced on all three benchmarks. This indicates that moderate refinement can improve the quality of the fitted difficulty estimator and make DyCon more efficient.

However, the third iteration reveals an important limitation. Although the regressor $R^2$ further increases to 0.9276, downstream performance does not continue to improve. In particular, AIME25 drops from 80.0 to 73.3, and token usage increases compared with the second iteration. This discrepancy indicates that a higher $R^2$ on refined trajectories does not necessarily imply better inference-time control. The reason is that excessive refinement can shift the fitting distribution away from the base model's original reasoning distribution. Since DyCon is ultimately applied during the model's actual inference process, the regressor must remain aligned with the trajectories that the model naturally produces. When refinement becomes too strong, the regressor may fit the refined data better while becoming less suitable for controlling the original inference behavior.

Overall, Table 31 and Table 32 together suggest that the benefit of regressor refinement does not come from simply removing noisy or redundant reasoning. Direct outlier removal can discard useful atypical trajectories and harm downstream performance. In contrast, moderate refinement improves the structure of reasoning trajectories while maintaining sufficient alignment with the base model's inference distribution. Excessive refinement, however, can introduce a distribution mismatch: the regressor becomes better at fitting the refined trajectories, but less effective for controlling the model under its natural inference behavior.

These findings position iterative refinement as an optional enhancement to DyCon rather than a necessary correction to the original pipeline. The standard DyCon procedure already provides strong performance, while one additional refinement round can further improve efficiency and accuracy when the refined trajectories remain close to the original model distribution. Designing principled criteria for determining when to stop refinement is an interesting direction for future work.

## B.11. Analysis of Unidirectional Logit Suppression

**Summary.** In this section, we analyze the effect of the modulation direction in DyCon. The main version of DyCon adopts a unidirectional logit-suppression strategy, where reflection-related token logits are selectively suppressed according to the estimated difficulty. To better understand this design choice, we compare it with a bidirectional variant that can both suppress and amplify reflection-token logits. As shown in Table 33, bidirectional modulation can further improve accuracy on several benchmarks, but it also substantially increases the number of generated tokens. In contrast, the original unidirectional DyCon achieves a more favorable efficiency–accuracy trade-off by preserving comparable accuracy while

*Table 33.* Comparison between unidirectional DyCon and a bidirectional modulation variant. We report Pass@1 and average output tokens in the format of Pass@1 / Tok. Bidirectional modulation improves accuracy in several cases but requires longer reasoning outputs, while the original unidirectional DyCon provides a stronger efficiency–accuracy trade-off.

| Model / Method | Math-500 | AIME24 | AIME25 |
|---|---|---|---|
| DeepSeek-R1-Distill-Qwen-7B | 92.0 / 3955 | 50.0 / 13008 | 36.7 / 15245 |
| + DyCon | 92.0 / 3216 | 53.3 / **10906** | 36.7 / 12415 |
| + Bidirectional-DyCon | **92.6** / 3556 | **56.7** / 12716 | **40.0** / 14796 |
| Qwen3-4B-Thinking-2507 | 96.2 / 6749 | 83.3 / 21493 | 76.7 / 22708 |
| + DyCon | 96.2 / **6092** | 86.7 / **18867** | 76.7 / **21100** |
| + Bidirectional-DyCon | **96.4** / 6562 | **90.0** / 21210 | **90.0** / 22487 |

producing significantly shorter reasoning outputs.

The goal of DyCon is not to maximize accuracy at any computational cost, but to improve reasoning efficiency while maintaining or improving task performance. This objective motivates the use of unidirectional suppression. When the estimated difficulty is low, suppressing reflection-related tokens encourages the model to avoid unnecessary continuation and terminate reasoning more efficiently. When the estimated difficulty is high, the suppression is weakened, allowing the model to preserve sufficient reasoning capacity. This design provides a conservative form of control: it primarily reduces excessive reasoning rather than actively forcing the model to reason more.

A natural alternative is bidirectional modulation, where the method suppresses reflection-related tokens under low estimated difficulty and amplifies them under high estimated difficulty. This variant can encourage more exploration on difficult instances and may therefore improve accuracy. This bidirectional control resembles the design philosophy of ReBalance (Li et al., 2026), which also adjusts reasoning behavior in both directions to balance performance and reasoning cost. However, such bidirectional modulation can also increase the tendency of the model to generate longer reasoning traces, especially when the difficulty estimator assigns high scores. We evaluate this bidirectional variant to understand whether the additional accuracy gain justifies the extra token cost.

Table 33 reports the comparison between the original DyCon and the bidirectional variant on DeepSeek-R1-Distill-Qwen-7B and Qwen3-4B-Thinking-2507. The results show a clear trade-off. For DeepSeek-R1-Distill-Qwen-7B, Bidirectional-DyCon improves Pass@1 from 92.0 to 92.6 on Math-500, from 53.3 to 56.7 on AIME24, and from 36.7 to 40.0 on AIME25 compared with standard DyCon. However, it also generates more tokens on all three benchmarks. A similar pattern is observed for Qwen3-4B-Thinking-2507: Bidirectional-DyCon improves accuracy, especially on AIME25, but its token usage becomes much closer to the baseline.

The results in Table 33 indicate that the modulation direction directly controls the trade-off between accuracy and efficiency. Bidirectional modulation is more aggressive: by amplifying reflection-related tokens on difficult instances, it can increase the chance of solving challenging problems, but this often comes with longer reasoning trajectories. Unidirectional suppression is more efficiency-oriented: it mainly removes unnecessary reflection when the model is estimated to be in a low-difficulty state, while avoiding excessive intervention on difficult problems.

Therefore, the original design of DyCon prioritizes reasoning efficiency under controlled accuracy constraints. This choice is aligned with the central goal of the method: reducing overthinking without substantially sacrificing task performance. The bidirectional variant is still useful as an alternative when the application prioritizes accuracy over token efficiency, but the unidirectional version offers a more balanced default setting for efficient inference.

### B.12. Analysis of Regressor Complexity

**Summary.** In this section, we analyze whether using a more complex difficulty regressor improves DyCon. The main implementation of DyCon adopts a simple linear regressor, which provides an efficient and stable way to decode difficulty from hidden states. To examine whether this design is overly restrictive, we replace the default linear regressor with a two-layer MLP whose hidden dimensions are 1024 and 512. As shown in Table 34, the MLP achieves competitive performance and can further improve accuracy in some cases. However, it does not consistently provide a clearly superior efficiency–accuracy trade-off over the simple linear regressor. This suggests that the difficulty signal used by DyCon is already largely accessible through a simple linear readout from model representations.

*Table 34.* Effect of regressor complexity on DyCon. We compare the base model, DyCon with the default OLS regressor, and DyCon with a two-layer MLP regressor. We report Pass@1 and average output tokens in the format of Pass@1 / Tok.

| Method | Math-500 | AIME2024 | AIME2025 |
|---|---|---|---|
| Qwen3-4B-Thinking-2507 | 96.2 / 6749 | 83.3 / 21493 | 76.7 / 22708 |
| + DyCon (OLS) | 96.2 / 6092 | 86.7 / 18867 | 76.7 / 21100 |
| + DyCon (MLP 1024, 512) | **96.6 / 5505** | 86.7 / **17809** | **80.0 / 19944** |

The difficulty estimator in DyCon maps intermediate hidden states to a scalar difficulty score. A natural question is whether this mapping requires a more expressive nonlinear model. In principle, a deeper regressor may capture more complex interactions among hidden dimensions and thus fit the training trajectories more accurately. However, increased regressor complexity may also introduce additional sensitivity to the fitting distribution, increase implementation cost, and provide limited benefit if the relevant difficulty information is already well organized in the representation space.

To study this question, we compare the default ordinary least squares (OLS) regressor with a two-layer MLP regressor. The MLP uses hidden dimensions of 1024 and 512, while all other components of DyCon remain unchanged. Table 34 reports the downstream performance on Math-500, AIME2024, and AIME2025 using Qwen3-4B-Thinking-2507.

The results in Table 34 show that the MLP regressor is effective: it improves Math-500 from 96.2 to 96.6, reduces token usage from 6092 to 5505 compared with the OLS version, and improves AIME2025 from 76.7 to 80.0 while also reducing the number of generated tokens. These results indicate that nonlinear regressors can serve as a valid alternative within the DyCon framework.

At the same time, the gains from the MLP are moderate rather than transformative. The simple OLS regressor already improves AIME2024 accuracy from 83.3 to 86.7 and substantially reduces token usage across all benchmarks compared with the base model. Moreover, on AIME2024, the MLP obtains the same Pass@1 as OLS, with the main difference being a further reduction in token count. This suggests that most of the useful difficulty signal can already be extracted by a lightweight linear readout.

Overall, Table 34 supports two conclusions. First, DyCon is robust to the choice of regressor: replacing the linear regressor with a more expressive MLP preserves, and in some cases improves, downstream performance. Second, the strong performance of OLS suggests that the model's hidden states already encode difficulty-related information in a largely linearly decodable form. Therefore, we use the simple linear regressor as the default choice because it is lightweight, stable, and sufficient for obtaining strong efficiency–accuracy trade-offs. More complex regressors remain a possible extension, especially when additional validation data are available for controlling overfitting and distribution sensitivity.

## B.13. Analysis of Effectiveness on Non-Reasoning Models

**Summary.** In this section, we analyze the behavior of DyCon-related difficulty estimation on non-reasoning instruction-tuned models. DyCon is primarily designed to mitigate overthinking in reasoning-oriented models, where excessive reflection and unnecessarily long reasoning traces are common. In contrast, non-reasoning models such as Qwen2.5-Instruct typically generate much shorter outputs and often do not exhibit the same degree of redundant reasoning. As shown in Table 35, Qwen2.5-7B-Instruct produces substantially shorter outputs than reasoning models, but its accuracy is also much lower on challenging mathematical benchmarks. This suggests that the main limitation of such models is often insufficient reasoning rather than excessive reasoning. Nevertheless, the difficulty-estimation component of DyCon remains meaningful: as shown in Table 36, regressors trained on Math can still recover reasonable dataset-level difficulty trends for non-reasoning models. These results suggest that while reflection suppression is less beneficial for non-reasoning models, difficulty estimation may still be useful for adaptive model routing or compute allocation.

DyCon targets the overthinking phenomenon in reasoning models. In such models, the generation process often contains long reflective segments, repeated verification, backtracking, and redundant intermediate reasoning. Suppressing reflection-related tokens under low estimated difficulty can therefore reduce unnecessary computation while preserving, or even improving, accuracy. This setting is different for non-reasoning instruction-tuned models. These models usually produce shorter answers and may not generate sufficiently detailed reasoning traces in the first place. Therefore, there is less redundant reflection to suppress.

Table 35 illustrates this behavior using Qwen2.5-7B-Instruct. Compared with reasoning models evaluated in the main

*Table 35.* Behavior of a non-reasoning instruction-tuned model on mathematical reasoning benchmarks. We report Pass@1 and average output tokens in the format of Pass@1 / Tok. The model produces short outputs but achieves much lower accuracy on challenging benchmarks, suggesting that insufficient reasoning is the main bottleneck.

| Model | Math-500 | AIME2024 | AIME2025 |
|---|---|---|---|
| Qwen2.5-7B-Instruct | 76.4 / 607 | 13.3 / 1144 | 6.7 / 1381 |

*Table 36.* Difficulty-estimation trends on non-reasoning instruction-tuned models. We report mean regressor-predicted difficulty scores and the corresponding ground-truth difficulty scores. Higher values indicate higher estimated reasoning difficulty.

| Model | Type | Math-500 | AIME2024 | AIME2025 | GSM8K |
|---|---|---|---|---|---|
| Qwen2.5-7B-Instruct | Prediction | 0.41 | 0.47 | 0.47 | 0.33 |
| Qwen2.5-7B-Instruct | Ground Truth | 0.41 | 0.50 | 0.50 | 0.33 |
| Qwen2.5-1.5B-Instruct | Prediction | 0.44 | 0.46 | 0.46 | 0.36 |
| Qwen2.5-1.5B-Instruct | Ground Truth | 0.44 | 0.48 | 0.46 | 0.36 |

experiments, Qwen2.5-7B-Instruct uses far fewer tokens on Math-500, AIME2024, and AIME2025. However, its Pass@1 is also much lower, especially on AIME2024 and AIME2025. This indicates that short generation alone is not necessarily desirable: for non-reasoning models, shorter outputs often reflect incomplete reasoning rather than efficient reasoning. Consequently, directly applying reflection suppression to such models is expected to bring limited benefit, because their primary bottleneck is not overthinking but under-reasoning.

Although suppression is less suitable for non-reasoning models, the underlying difficulty-estimation assumption still holds to a meaningful extent. Specifically, we examine whether hidden states from non-reasoning models encode information about remaining generation length, which serves as a proxy for model-perceived difficulty. Regressors trained on Math achieve stable fitting quality, with approximately $R^2 \approx 0.64$ and $\text{MAE} \approx 0.06\text{–}0.07$. This indicates that even when the model does not produce long reasoning traces, its hidden representations still contain useful signals related to expected reasoning effort.

To further evaluate this point, we compare the predicted difficulty scores with ground-truth difficulty scores across datasets. Table 36 reports the dataset-level difficulty trends for Qwen2.5-7B-Instruct and Qwen2.5-1.5B-Instruct. For both models, the predicted scores are closely aligned with the ground-truth values on Math-500 and GSM8K, and also reasonably track the higher difficulty of AIME2024 and AIME2025. This suggests that the regressor can still recover meaningful relative difficulty information from non-reasoning models.

Overall, Table 35 and Table 36 show that non-reasoning models differ from reasoning models in two important ways. First, they already generate relatively short outputs, so reflection suppression has limited room to reduce redundant reasoning. Second, despite their shorter and often incomplete reasoning traces, their hidden states still encode useful difficulty-related information. Therefore, while DyCon's suppression mechanism is most effective for reasoning models with pronounced overthinking, its difficulty estimator can still be valuable for non-reasoning models.

One potential application is adaptive routing. For example, a lightweight non-reasoning model could first estimate the difficulty of an input; if the estimated difficulty is low, the system may allow the non-reasoning model to answer directly, while high-difficulty cases can be routed to a stronger reasoning model or allocated more inference compute. In this sense, DyCon's difficulty-estimation component may serve as a general lightweight signal for adaptive inference, even when reflection suppression itself is not the primary intervention.

## C. Related Work

**From Parameter Scaling to Reasoning Scaling.** Classical scaling laws establish that model performance follows power-law relationships with model size, data, and compute (Kaplan et al., 2020). Following this paradigm, recent large models, such as GPT-4o (Hurst et al., 2024) and DeepSeek-V3 (Liu et al., 2024), have achieved remarkable success largely through massive parameter and compute scaling. This scaling momentum has also propagated beyond text-only NLP into multimodal and vision-language domains, reshaping tasks from reasoning segmentation, open-vocabulary perception, and language-driven adaptation to multimodal reasoning, visual-token compression, scene generation, and intervention-based reliability improvement (Lai et al., 2024a; Yang et al., 2023; Shao et al., 2024; Yang et al., 2024; Wang et al., 2025b;c;

Li et al., 2025b; Yang et al., 2025c; Huang et al., 2025a; Peng et al., 2025b). Meanwhile, foundation model designs have further influenced visual perception and representation learning, including semantic segmentation, few-shot segmentation, scene text detection, long-tailed recognition, 3D understanding, contrastive learning, point prompt learning, and 2D-3D representation learning (Tian et al., 2020; Lai et al., 2021; Tian et al., 2019; Cui et al., 2022; Jiang et al., 2021; Peng et al., 2023; Tian et al., 2022b; Cui et al., 2023; Luo et al., 2023; Peng et al., 2024b; Tian et al., 2022a; 2023; Wang et al., 2024; Ning et al., 2023; Wu et al., 2024; Zhang et al., 2025b; Huang et al., 2025b).

However, the marginal gains from parameter scaling often come with prohibitive computational costs. Consequently, a complementary paradigm, reasoning scaling, has emerged, which improves model capability by expanding the depth and structure of inference-time reasoning rather than merely increasing model width. Starting from Chain-of-Thought prompting (Wei et al., 2022), this trajectory has evolved into more structured reasoning and search mechanisms, such as self-correction (Kumar et al., 2024), Tree-of-Thoughts (Yao et al., 2023), and Graph-of-Thoughts (Besta et al., 2024), and has been further refined by preference optimization and continual adaptation techniques for long-chain reasoning (Lai et al., 2024b; Peng et al., 2025a; 2024a). While these methods enable smaller models to approach the performance of larger ones, they often introduce excessive inference overhead due to overthinking. To address this issue, we propose **DyCon**. Instead of further enlarging models or blindly extending reasoning chains, DyCon dynamically estimates residual reasoning demand from hidden states and adaptively regulates reasoning termination, reducing unnecessary thinking tokens while preserving answer quality.

**Large Reasoning Models.** Building on this line of work, a new class of large reasoning models has recently emerged, including the DeepSeek-R1 series (Guo et al., 2025) and OpenAI o1 series (Jaech et al., 2024). These models generate explicit intermediate reasoning before producing final answers, enabling iterative deliberation and improved problem decomposition. As a result, they achieve substantially improved performance on complex reasoning tasks.

**Efficient Reasoning.** Despite their strong reasoning capability, large reasoning models (LRMs) still face notable challenges. In particular, excessively long reasoning processes (often referred to as *overthinking*) introduce substantial computational overhead. A central question is therefore how to preserve strong reasoning ability while reducing reasoning length to improve efficiency. This motivates the line of work on *efficient reasoning*. Among existing approaches, the most direct and widely adopted strategy is prompt-based control of reasoning behavior, including static prompt designs such as BTC (Ding et al., 2024), CoD (Xu et al., 2025a), CCoT (Renze & Guven, 2024), CCoT-2-45 (Nayab et al., 2024), and NoThinking (Ma et al., 2025), as well as dynamic prompt methods such as ThinkPilot (Li et al., 2025a). Beyond prompt-based methods, training-based approaches also constitute an important direction for efficient reasoning. These methods leverage supervised fine-tuning (SFT) or reinforcement learning (RL) to explicitly encourage shorter chains of thought while preserving reasoning accuracy. Representative work along this line includes C3oT (Kang et al., 2025), as well as SFT- and RL-based approaches for chain-of-thought compression and distillation (Arora & Zanette, 2025; Munkhbat et al., 2025; Shen et al., 2025). In addition, leveraging latent reasoning constitutes another important direction for efficient reasoning. Rather than explicitly generating full chains of thought, these methods operate on latent or implicit reasoning representations, aiming to reduce token-level reasoning overhead while retaining reasoning capability. Representative approaches include SoftThinking (Zhang et al., 2025c) and SoftCoT (Xu et al., 2025b). Early-exit methods constitute another active direction for efficient reasoning. For example, TrimR (Lin et al., 2025a) and FlashThinking (Jiang et al., 2025) employ external large models to monitor the reasoning process and trigger early termination. In contrast, DEER (Yang et al., 2025b) leverages the model's internal confidence signals to decide when to exit, while Dynasor-CoT (Fu et al., 2025) uses agreement across multiple sampled answers to guide early termination. These methods demonstrate the effectiveness of early-exit strategies for reducing reasoning cost.

## D. Details On Experimental Settings

### D.1. Decoding and Sampling Settings

To ensure optimal model performance, we follow the original model configurations and experimental settings adopted in (Guo et al., 2025; Team, 2025; Yang et al., 2025a). For the Qwen3-4B-Thinking-2507 model, we set the temperature to 0.6, Top-$p$ to 0.95, Top-$k$ to 20, and Min-$p$ to 0, with the maximum output length fixed at 81,920 tokens. For the DeepSeek-R1-Distill-Qwen-7B, QwQ-32B, Qwen3-8B and Qwen3-14B models, we adopt the same sampling configuration (temperature = 0.6, Top-$p$ = 0.95, Top-$k$ = 20, Min-$p$ = 0), while setting the maximum output length to 32,768 tokens. All experiments are conducted with a fixed random seed of 42 to ensure reproducibility.

*Table 37.* Predefined token phrases used for suppression, following NoWait (Wang et al., 2025a).

| Predefined Token Phrases |
| --- |
| wait, alternatively, hmm, but, however, alternative, another, check, double-check, oh, maybe, verify, other, again, now, ah, any |

### D.2. Token Lists Used for Suppression

For reproducibility, we adopt the same predefined token lists for suppression as used in NoWait (Wang et al., 2025a), as summarized in Table 37, enabling a direct and fair comparison without introducing additional design choices.

### D.3. Implementation Details

We implement our method using both the native HuggingFace Transformers library (Wolf et al., 2019) and vLLM (Kwon et al., 2023). Unless otherwise stated, all experimental results reported in this paper are based on the HuggingFace Transformers implementation (Wolf et al., 2019).

### D.4. Details on Benchmarks

**Math-500** (Lightman et al., 2023): A difficulty-balanced mathematical reasoning benchmark comprising 500 problems, with each instance labeled according to a five-level difficulty hierarchy (Level 1 to Level 5).

**GSM8K** (Cobbe et al., 2021): A grade-school mathematics reasoning benchmark comprising 1,319 problems, on which most instruction-tuned models already achieve high accuracy.

**AIME2024** (AI-MO, 2024a): A set of 30 challenging problems from the American Invitational Mathematics Examination, with difficulty substantially exceeding that of the AMC series and typically requiring extended multi-step reasoning.

**AIME2025** (OpenCompass, 2025): A collection of 30 challenging problems from the American Invitational Mathematics Examination, commonly regarded as an extension of AIME2024 and similarly demanding complex, multi-step reasoning.

**AMC23** (AI-MO, 2024b): Problems from the AMC (American Mathematics Competition), one of the most influential pre-college mathematics competitions worldwide, consisting of 40 problems and typically regarded as lower in difficulty compared to AIME-level benchmarks.

**Olympiad Bench** (He et al., 2024): A collection of 675 challenging Olympiad-style problems drawn from international mathematical olympiad competitions, typically requiring deep and rigorous multi-step reasoning.

**MMLU** (Hendrycks et al., 2020): A large-scale, multi-task benchmark consisting of multiple-choice questions drawn from a wide range of knowledge domains. The benchmark spans the humanities, social sciences, and the hard sciences. In this work, we adopt the abstract mathematics subset to evaluate models' mathematical reasoning abilities, comprising 100 problems of relatively low difficulty.

**GPQA Diamond** (Rein et al., 2024): A challenging scientific multiple-choice benchmark comprising 198 questions authored by domain experts in biology, physics, and chemistry.

**LiveCodeBench** (Jain et al., 2024): A code evaluation benchmark consisting of 400 programming problems drawn from diverse sources, including LeetCode, AtCoder, and Codeforces. We use version v1 in our experiments.

**StrategyQA** (Geva et al., 2021): A creative and diverse yes–no question benchmark that requires implicit multi-step reasoning. The dataset contains 2,290 questions and is generally of low difficulty.

**TriviaQA** (Joshi et al., 2017): A reading comprehension benchmark composed of question–answer–evidence triples. In this work, we disable retrieval-augmented generation (RAG) to assess the model's intrinsic knowledge and reasoning capabilities. From the original test split, we randomly sample 20% of the examples for evaluation, resulting in a subset of 3,589 knowledge-oriented questions of moderate difficulty.

**CommonSenseQA** (Talmor et al., 2019): A multiple-choice question answering benchmark that requires diverse types of commonsense knowledge to identify the correct answer. Each instance consists of one correct option and four distractors, with a total of 1,221 questions.

## D.5. Details of Baseline Methods

In our performance comparison, we evaluate the proposed method against a broad range of representative efficient reasoning approaches across multiple paradigms. Specifically, we consider: (1) *steering-based* methods, including SEAL (Chen et al., 2025), Controlling Thinking Speed (Lin et al., 2025b) and Manifold Steering (Huang et al., 2025d); (2) *prompt-based* methods, including CoD (Xu et al., 2025a), NoThinking (Ma et al., 2025), and ThinkPilot (Li et al., 2025a); (3) *early-exit–based* methods, including DEER (Yang et al., 2025b), TrimR (Lin et al., 2025a), Dynasor-CoT (Fu et al., 2025) and FlashThinking (Jiang et al., 2025); and (4) *output-based* methods, represented by NoWait (Wang et al., 2025a).

## D.6. Details on Prompts.

Math-500, AIME2024, AIME2025, AMC23, GSM8K, Olympiad-Bench, and MMLU:

```
<|System|> Please reason step by step, and place the final answer inside \boxed{}.

<|User|> [question]
```

GPQA Diamond, CommonSenseQA:

```
<|System|> Please reason step by step, and place the final answer inside \boxed{}.

<|User|> [question]

Answer with the choice letter only, in \boxed{}.  Do not include option text.
```

StrategyQA:

```
<|System|> You answer binary commonsense questions.  Think step by step, then output
exactly one final line:  \boxed{Yes} or \boxed{No}.

<|User|> [question]

Answer with \boxed{Yes} or \boxed{No} only.
```

LiveCodeBench:

```
<User> ###### Instruction:  You will be given a question (problem specification)
and will generate a correct Python program that matches the specification and
passes all tests.  You will NOT return anything except for the program.  Question:
[problem] Ensure that when the python program runs, it reads the inputs, runs
the algorithm and writes output to STDOUT. python ## YOUR CODE HERE ######
Response:<|im_end|><|im_start|>assistant<|think|
```

TriviaQA:

```
<|System|> Please answer the question.

Directly provide the final answer inside <answer> and </answer>, without any
explanation or additional text.

Example:  <answer> London </answer>

<|User|> [question]
```

Step-wise Difficulty Self-Assessment Prompt:

```
Let me quickly rate this problem's difficulty (1=almost solved, 2=some uncertainty
remains, 3=missing key step) based on the reasoning so far.  Difficulty =
```

## D.7. Hardware Configuration.

All experiments were performed on NVIDIA RTX PRO 6000 (Blackwell Server Edition) GPUs to ensure a consistent hardware environment.

# E. Case Study

## Case Study 1: Easy Case (From GSM8K)

**Input Problem**: Miguel uses 2 pads of paper a week for his drawing. If there are 30 sheets of paper on a pad of paper, how many sheets of paper does he use every month?

**Ground Truth:** 240                                     **Generate Length**: 983

- - - - - - - - - - - - - - - - - - - - - - - - - - - - - - - - - - - - - - - - - - - -

Okay, let's see. The problem is about Miguel using paper pads. He uses 2 pads a week, and each pad has 30 sheets. We need to find out how many sheets he uses every month. Hmm, first, I need to figure out how many sheets per week, then convert that to a month. But wait, how many weeks are in a month? That's a bit tricky because months have different numbers of days. Maybe the problem assumes a standard month, like 4 weeks? Wait, sometimes in math problems, they approximate a month as 4 weeks for simplicity. Let me check.\n\n          *Regressor Output: 0.43*

First, let's calculate weekly usage. He uses 2 pads per week, each pad has 30 sheets. So per week, that's 2 pads * 30 sheets/pad = 60 sheets per week. Got that.\n\n          *Regressor Output: 0.48*

Then, how many weeks in a month? Well, a month is roughly 4 weeks, right? Like, 4 weeks * 7 days = 28 days, which is close to a month (some months have 30 or 31 days). Since the problem doesn't specify, I guess we go with 4 weeks per month. Let me confirm. If it's a month with 30 days, that's about 4.285 weeks, so approximately 4 weeks. In many math problems, especially at a basic level, they use 4 weeks per month. Let's go with that.\n\n          *Regressor Output: 0.42*

So, if he uses 60 sheets per week, then in 4 weeks (a month), that's 60 * 4 = 240 sheets. Let me make sure. Let's see: 2 pads/week * 30 sheets/pad = 60 sheets/week. Then, 60 sheets/week * 4 weeks/month = 240 sheets/month. Yeah, that seems right. \n\n          *Regressor Output: 0.34*

Key Computation

Then, how many weeks in a month? Well, a month is roughly 4 weeks, right? Like, 4 weeks * 7 days = 28 days, which is close to a month (some months have 30 or 31 days). Since the problem doesn't specify, I guess we go with 4 weeks per month. Let me confirm. If it's a month with 30 days, that's about 4.285 weeks, so approximately 4 weeks. In many math problems, especially at a basic level, they use 4 weeks per month. Let's go with that.\n\n          *Regressor Output: 0.33*

So, if he uses 60 sheets per week, then in 4 weeks (a month), that's 60 * 4 = 240 sheets. Let me make sure. Let's see: 2 pads/week * 30 sheets/pad = 60 sheets/week. Then, 60 sheets/week * 4 weeks/month = 240 sheets/month. Yeah, that seems right.\n\n          *Regressor Output: 0.30*

Is there a chance they mean 30 days a month? Like, 30 days in a month, so 30/7 weeks? Let's try that. 30 days / 7 days per week ≈ 4.2857 weeks. Then 60 sheets/week * 4.2857 weeks ≈ 257.14 sheets. That's a bit more, about 257.14. The problem says \"every month,\" so depending on the context. In school problems, they usually use 4 weeks per month. Let me see if the problem gives a hint. The problem says \"how many sheets of paper does he use every month?\" without specifying the month's length. Since it's a math problem for students, probably expects 4 weeks per month.\n\n          *Regressor Output: 0.28*

Let me recall similar problems. For example, if a problem says \"how many in a month\" without specifying, they often use 4 weeks. So 2 pads/week * 30 sheets = 60 sheets/week. 60 * 4 = 240. So the answer is 240.\n\n I don't see a trick here. Let's just go with 240.\n</think> **[Answer]**

*Figure 15.* Qualitative case study on an easy GSM8K problem for **Qwen3-4B-Thinking-2507**. The difficulty regressor stays low from the beginning and further decreases as the core computation is completed, yielding a short, stable reasoning trajectory.

## Case Study 2: Hard Case (From AIME)

**Input Problem**: There are $n$ values of $x$ in the interval $0 < x < 2\\pi$ where $f(x) = \\sin(7\\pi \\cdot \\sin(5x)) = 0$. For $t$ of these $n$ values of $x$, the graph of $y = f(x)$ is tangent to the $x$-axis. Find $n + t$.

**Ground Truth:** 149                                      **Generate Length**: 12417

- - - - - - - - - - - - - - - - - - - - - - - - - - - - - - - - - - - - - - - - - -

This is a ==complex or challenging== question, and it is difficult to provide a direct and correct answer. I need to think about it.\n Well, so the problem says there are n values of x in (0, 2π) where f(x) = sin(7π sin(5x)) = 0, and then among those n values, t of them are points where the graph is tangent to the x-axis, meaning the derivative is zero there too, right? Then we need n + t. Let's start by recalling when a sine function is zero and when its derivative is zero (i.e., when it's tangent to the x-axis). \n\n                                           Regressor Output: 0.97

==The model begins to recognize this as a difficult problem.==

First, let's handle the zeros of f(x). f(x) = 0 when the argument of the outer sine is an integer multiple of π, right? So sin(θ) = 0 if θ = kπ for integer k. Here, θ = 7π sin(5x), so: 7π sin(5x) = kπ ⇒ 7 sin(5x) = k ⇒ sin(5x) = k/7. \n\n                    Regressor Output: 0.95

Okay, so that's a ==key equation==. Now, sin(5x) has to be between -1 and 1, so k/7 must be in [-1, 1], so k is integer with -7 ≤ k ≤ 7. Let's list possible k values: k = -7, -6, ..., 0, ..., 6, 7. For each k, we can find the number of solutions x in (0, 2π) to sin(5x) = k/7. \n\n         Regressor Output: 0.91

First, let's remember that for the equation sin(y) = c, where y = 5x, so when we solve for y first, then x = y/5. Let's let y = 5x, so as x goes from 0 to 2π, y goes from 0 to 10π. So we can think in terms of y ∈ (0, 10π) (since x ∈ (0, 2π) ⇒ y ∈ (0, 10π)), and then each solution y gives a unique x = y/5 in (0, 2π). So for each k, sin(y) = k/7, y ∈ (0, 10π). How many solutions does sin(y) = c have in (0, 10π) for a given c? \n\n                    Regressor Output: 0.88

 … …

List them out:\n\n1. π/10 ≈0.314\n2. 3π/10≈0.942\n3. π/2≈1.571\n4. 7π/10≈2.199\n5. 9π/10≈2.827\n6. 11π/10≈3.456\n7. 13π/10≈4.084\n8. 3π/2≈4.712\n9. 17π/10≈5.341\n10. 19π/10≈5.969\n\nAll these are between 0 and 2π (≈6.283), yes, 19π/10=1.9π≈5.969 <6.283, next would be 21π/10=2.1π>6.283, so 10 points==, no more, no less. Perfect==, t=10.\n\n So now, we have n=139, t=10, so n + t=149?                    Regressor Output: 0.71

 … …                              ==Critical computation! The problem is reduced to a simple one==

\n\n So n=139, t=10, so n + t=149.\n\n                      Regressor Output: 0.51

Just to recap why t=10: tangency points for f(x)=sin(g(x)) at zeros of f(x) are exactly where g'(x)=0 (since cos(g(x))≠0 at zeros), g'(x)=0 ⇒ cos(5x)=0 ⇒ sin(5x)=±1 ⇒ f(x)=sin(7π*±1)=0, so these are valid zeros, and there are 10 such x in (0,2π), as counted.\n\n  Regressor Output: 0.50

Yes, this all lines up. I thinks the key was realizing through the general rule for f(x)=sin(g(x)) that tangency at zeros requires g'(x)=0, which simplified the problem a lot, and then verifying that those points do satisfy f(x)=0 (which they do, since sin(±7π)=0).\n\n     Regressor Output: 0.41

I don't see a reason to doubt the counts anymore. So n=139, t=10, sum is 149.\n\n**Final Answer**\n\\boxed{149}\n</think>\n\n **[Answer]**

*Figure 16.* Qualitative case study on a hard AIME problem for **Qwen3-4B-Thinking-2507**. The figure shows the step-wise reasoning transcript with difficulty regressor annotations. The regressor remains near 1.0 for most of the trajectory and only drops to ∼0.5 after a late key insight, indicating that the model resolves the core difficulty only near the end of reasoning.

