# OpenReview forum: "DyCon: Dynamic Reasoning Control via Evolving Difficulty Modeling"
_ICML.cc/2026/Conference — ICML 2026 regular_

### Official Review · Reviewer_dEVM · 2026-02-26

**Soundness:** 3
**Presentation:** 4
**Significance:** 4
**Originality:** 4
**Overall Recommendation:** 4
**Confidence:** 4

**Summary:**

This paper proposes a training-free framework Dycon for dynamically controlling the reasoning depth of LRMs and reduce overthinking. The key insight is that the difficulty of a reasoning is dynamically varing and is linearly encoded in the model’s step-level hidden states. DyCon leverages this by fitting a lightweight linear regressor on a small-scale dataset to predict remaining reasoning length from step embeddings, and using this predictor to modulate logits of reflection-triggers. Extensive experiments across 4B–32B models demonstrate that DyCon achieves up tp 40.6% token reduction and 6.7% absolute accuracy gains.

**Compliance With Llm Reviewing Policy:**

Affirmed.

**Final Justification:**

My initial score was 4 (Weak Accept). My primary concerns centered on why step-level difficulty can be linearly represented by step embeddings, and How does the choice of training set affects the regressor's performance. The authors' rebuttal has addressed these issues, so I have decided to maintain my positive assessment to this work.

**Key Questions For Authors:**

1. Why can step-level difficulty be linearly represented by step embeddings? I hope the authors provide additional explanation beyond the experimental results.

2. How does the choice of training set affect the regressor's performance? Would fitting on a larger subset of MATH or a more general-purpose dataset (e.g. CC, C4, The Pile) yield better results ?

3. According to the results in Figure 2(b), across all four models, the regressor tends to slightly overestimate the difficulty of short responses and underestimate that of long responses. Can you explain this?

4. What will happen using more complex regressor (e.g. mlp) ?

5. Does DyCon work on non-reasoning models (e.g. Qwen2.5-7B-Instruct) ?

**Limitations:**

yes

**Strengths And Weaknesses:**

Strenths

1. The paper is easy to follow. The insight is clear and meaningful. Figures are well drawn and help readers to understand experiments and method.

2. The key observation (Section 2.2) that the step-level difficulty (Eq.4) evolves during reasoning and can be linear encoded in step embeddings in is surprising. Results in Figure 2 clearly support this conclusion.

3. Dynamically control reasoning according to predicted difficulty (Section 3.3) by modulating logits of reflection-triggers is reasonable.

Weaknesses

1. The "training-free" means DyCon does not change LRM's parameters, but it still needs to fit a simple linear regressor on a small-scale dataset. Although it's a lightweight fitting, however, **selecting an appropriate dataset is non-trivial** and directly affects the regressor's performance and generalizability.

2. The core of the method is heuristic (especially Equation 6/9/12). Although it achieves good results in practice, it lacks solid theoretical foundations.

3. DyCon adjusts the suppression magnitude of the target tokens' logits based on difficulty. This modulation is unidirectional, making the model's overall logits flatter which inherently alters the model's output behavior. The authors did not conduct an ablation study on this aspect in their experiments.

4. Although significantly reducing token consumption while maintaining performance (Table 1 and Table 2) is meaningful, the results in Figure 4 show that DyCon achieves a smaller reduction in token consumption compared to previous approaches.

---

> ### Author Rebuttal · Authors · 2026-03-31
>
> > **Response to W1**
>
> Empirically, the regressor is robust to dataset scale and composition. Figure 5(c,d) shows stable performance across sizes and domains, and Table 2 confirms strong cross-domain generalization. Thus, DyCon does not rely on curated datasets.
>
> > **Response to W2**
>
> While simple, each component is grounded in established principles.
>
> **Eq. (6):** the remaining length proxy exhibits a clear heavy-tailed distribution. Direct regression on such targets is known to be unstable, while log transformation and normalization provide a more stable and comparable control signal across samples and steps, consistent with prior findings on imbalanced regression [1]. We also verify the effectiveness of this design in Appendix A.4, Table 8.
>
> **Eq. (9):** the operation is essentially a logits reweighting step. Prior work has shown that reweighting logits alone can effectively adapt model behavior at the distribution level without changing model parameters [2]. In our case, the perturbation is restricted to reflection-related tokens, and thus only minimally affects the overall distribution; detailed evidence is provided in Appendix B.4, Table 19.
>
> **Eq. (12):** the design is a monotonic control rule: stronger suppression is applied at low-difficulty steps, while weaker suppression is used at high-difficulty steps to avoid over-interfering with necessary reasoning. This is consistent with the principle of adaptive computation, where easier inputs require less computation and harder ones require sustained reasoning [3].
>
> [1] Delving into Deep Imbalanced Regression
>
> [2] Logits are All We Need to Adapt Closed Models
>
> [3] Adaptive Computation Time for Recurrent Neural Networks
>
> > **Response to W3**
>
> We compare DyCon with a bidirectional variant. The bidirectional version improves accuracy but generates longer outputs, while DyCon achieves comparable accuracy with significantly fewer tokens.
>
> |Model|Math-500 (Pass@1/Tok)|AIME24|AIME25|
> |---|---|---|---|
> |Qwen3-4B|96.2/6749|83.3/21493|76.7/22708|
> |+DyCon|96.2/**6092**|86.7/**18867**|76.7/**21100**|
> |+Bidirectional|**96.4**/6562|**90.0**/21210|**90.0**/22487|
>
>
> Thus, modulation direction determines the trade-off: bidirectional improves accuracy but increases tokens, while unidirectional suppression achieves higher efficiency with minimal loss. We therefore prioritize efficiency under controlled accuracy.
>
> > **Response to W4**
>
> DyCon’s reduction is difficulty-dependent: simple problems contain more redundant reasoning, while tokens on hard problems are largely necessary. Thus, it achieves larger reductions on easy datasets and more conservative ones on hard datasets. Consistently, Table 1 shows both improved accuracy and the largest reduction on simpler benchmarks, indicating Figure 4 reflects controlled, difficulty-aware behavior rather than a limitation.
>
> > **Response to Q1**
>
> Motivated by prior work showing that high-level semantics in LMs are often linearly represented in hidden space [4], we find difficulty is also linearly decodable from step embeddings: linear regressors perform strongly. This suggests difficulty is encoded in a linearly accessible form. While not a formal guarantee, DyCon is grounded in representation geometry and supported empirically.
>
> [4] The Linear Representation Hypothesis and the Geometry of LLMs
>
> > **Response to Q2**
>
> We evaluate both regression quality and downstream performance.
>
> Regressors fitted on **Math** and **C4** both learn meaningful signals (R²: 0.80 vs. 0.74), indicating the proxy is not math-specific.
>
> |Regressor|Math-500|AIME24|AIME25|GPQA|CSQA|
> |---|---|---|---|---|---|
> |Qwen3-4B+DyCon (C4)|96.0/5774|76.7/18065|83.3/21412|66.7/7824|79.7/2323|
>
> Math improves structured reasoning, while C4 improves transfer but may weaken structure. Predicted difficulty also aligns with ground-truth trends across datasets. Mixed training further improves performance (omitted due to space). Overall, DyCon is not tied to a specific dataset.
>
> > **Response to Q3**
>
> This arises from regression-to-the-mean and representation compression. Linear models shrink predictions toward the mean, and hidden states encode compressed semantics rather than exact step counts. This leads to slight overestimation for short responses and underestimation for long ones.
>
> > **Response to Q4**
>
> A two-layer MLP is competitive but does not consistently outperform linear models, suggesting the key signal is largely linearly decodable.
>
> |Regressor|Math-500|AIME24|AIME25|
> |---|---|---|---|
> |Qwen3-4B+MLP|96.6/5505|86.7/17809|80.0/19944|
>
> > **Response to Q5**
>
> DyCon targets overthinking, which is limited in non-reasoning models, so gains are smaller. However, difficulty estimation remains valid and aligns with ground-truth trends.
>
> |Model|Type|Math-500|AIME24|AIME25|GSM8K|
> |---|---|---|---|---|---|
> |Qwen2.5-7B-Instruct|Prediction|0.41|0.47|0.47|0.33|
> ||Ground Truth|0.41|0.50|0.50|0.33|
>
> This suggests a potential use as a lightweight difficulty estimator for routing.

---

> > ### Author Rebuttal · Reviewer_dEVM · 2026-04-01
> >
> > I deeply appreciate the authors' remarkably thorough and thoughtful rebuttal. While the linear layer prediction difficulty setting remains somewhat debatable, these minor concerns do not overshadow the overall merit of this work. Good luck!

---

> > > ### Author Response · Authors · 2026-04-02
> > >
> > > We sincerely thank the reviewer for the positive feedback and thoughtful evaluation. We also appreciate the insightful discussion on linear difficulty representation. We will continue exploring this direction to further strengthen our understanding and analysis. Thank you again for your time and encouraging comments.

---

### Official Review · Reviewer_v3W9 · 2026-03-11

**Soundness:** 3
**Presentation:** 3
**Significance:** 3
**Originality:** 3
**Overall Recommendation:** 4
**Confidence:** 3

**Summary:**

This paper addresses the issue that LRMs may be overthinking,  generating excessive reasoning steps without improving correctness. To address this issue, the paper proposes DyCon, a training-free framework for difficulty-Aware dynamic reasoning control. By predicting difficulty from step embeddings with a lightweight linear regressor and adjusting the logits of reflection-related keywords, DyCon alleviates the overthinking problem in LRMs.

**Compliance With Llm Reviewing Policy:**

Affirmed.

**Key Questions For Authors:**

1. Could the authors further clarify the role of the suppression vocabulary—i.e., is DyCon truly vocabulary-insensitive across different backbones and task domains, or is there an optimal/near-optimal vocabulary selection strategy?
 2. Could the authors analyze how redundant reasoning in the training affects the linear regressor’s ability to predict difficulty?
 3. Could the regressor be iteratively retrained using DyCon-generated trajectories to form a closed-loop refinement process?

**Limitations:**

Yes

**Strengths And Weaknesses:**

Strengths:
1. The paper addresses an important issue, mitigating overthinking and improving reasoning efficiency for large reasoning models.
2. The paper introduces a useful perspective for dynamic control, using step-level embeddings to explicitly model the evolving task difficulty, to decide whether to continue reasoning.
3. The experiments cover multiple benchmarks and models.

Weaknesses:
1. This method relies on adjusting the logits of reflection-related keywords to control reasoning, making the design of the suppression vocabulary a significant component. However, the paper does not explicitly clarify how to select an optimal reflection keyword vocabulary. Table 6 only provides a vocabulary on Qwen3-4B on Math-500, which is insufficient for robustness across different backbones and task domains.
2. The regressor is trained using remaining reasoning length as a proxy fo difficulty, but the training trajectories themselves may contain redundant reasoning.

---

> ### Author Rebuttal · Authors · 2026-03-31
>
> > **Response to W1 & Q1 (Rebustness of the Reflection Vocabulary)**
>
> We thank the reviewer for this insightful question and clarify that the predefined reflection token list is not essential to our method but represents a conventional choice consistent with previous work [1]. Our approach does not rely on any specific token set.
>
> To empirically validate this claim, we replace our initial token list with the token set proposed by SEAL [2], consistently achieving comparable performance across various models and benchmarks. These results underscore DyCon’s robustness to different reflection token choices.
>
> |Model|Math-500（Pass@1/Tok）|AIME24（Pass@1/Tok）|AIME25（Pass@1/Tok）|
> |---|---|---|---|
> |DeepSeek-R1-Distill-Qwen-7B|92.0/3955|50.0/13008|36.7/15245|
> |+DyCon|92.0/3216|53.3/10906|36.7/12415|
> |+DyCon (SEAL)|92.2/3569|56.7/11141|40.0/12775|
> |Qwen3-4B-Thinking-2507|96.2/6749|83.3/21493|76.7/22708|
> |+DyCon|96.2/6092|86.7/18867|76.7/21100|
> |+DyCon (SEAL)|96.4/5753|83.3/18056|83.3/19147|
>
> We agree that an optimal token set might exist tailored to each model. To explore this, we employ an evolutionary strategy initialized with frequently occurring reflection-related tokens. Inspired by dynamic context learning [3], we iteratively apply token selection, mutation, and crossover to enhance model accuracy on a held-out Math validation set.
>
> |Model|Math-500（Pass@1/Tok）|AIME24（Pass@1/Tok）|AIME25（Pass@1/Tok）|
> |---|---|---|---|
> |Qwen3-4B-Thinking-2507|96.2/6749|83.3/21493|76.7/22708|
> |+DyCon (Optimized)|**96.6/5898**|**86.7/19136**|**83.3/16848**|
>
> The optimized token sets further enhance performance compared to the original list. This finding suggests that while DyCon is robust regarding token selection, further refinements can yield additional improvements. Investigating principled optimization objectives and advanced selection strategies constitutes promising future research directions.
>
> [1] Wait, We Don't Need to "Wait"! Removing Thinking Tokens Improves Reasoning Efficiency
>
> [2] SEAL: Steerable Reasoning Calibration of Large Language Models for Free
>
> [3] ThinkPilot: Steering Reasoning Models via Automated Think-prefixes Optimization
>
> >  **Response to W2 & Q2 (Noisy Difficulty Proxy)**
>
> We thank the reviewer for this important concern. We agree that trajectories may contain redundant reasoning, which introduces noise when using generation length as a proxy for difficulty. However, our goal is not to rely on exact labels, but to exploit a statistically meaningful signal that correlates with difficulty. Empirically, generation length shows strong correlation (Spearman’s ρ ≈ 0.62) with ground-truth difficulty on MATH, serving as a reliable proxy.
>
> Besides, to quantify the impact of confounding cases, we perform an outlier analysis based on generation length, using the mean, first quartile (q25), third quartile (q75), and interquartile range (IQR). Specifically, outliers denote the samples whose generation length lies outside the interval:
>
> $$
> [q_{25} - 1.5 \cdot \text{IQR},\; q_{75} + 1.5 \cdot \text{IQR}], \quad \text{where } \text{IQR} = q_{75} - q_{25}.
> $$
>
> |Model|Level|mean|q25|q75|outlier ratio|
> |---|---|---|---|---|---|
> |Qwen3-4B-Thinking-2507|1|2121|903|2088|0.10|
> ||2|3068|1112|4454|0.03|
> ||3|5322|1839|7155|0.03|
> ||4|6840|2930|9327|0.02|
> ||5|12025|6356|15386|0.04|
> |DeepSeek-R1-Distill-Qwen-7B|1|1932|1159|2162|0.05|
> ||2|2230|1302|2600|0.06|
> ||3|2953|1570|3149|0.10|
> ||4|3482|1721|3866|0.09|
> ||5|5580|2551|6941|0.09|
> |Qwen3-14B|1|1978|1324|2114|0.08|
> ||2|2806|1493|3022|0.09|
> ||3|3570|1987|4243|0.06|
> ||4|4610|2429|5568|0.05|
> ||5|7673|3846|9326|0.06|
> |QwQ-32B|1|1751|1191|2015|0.07|
> ||2|2297|1338|2832|0.04|
> ||3|3303|1757|4060|0.05|
> ||4|4311|2045|5196|0.07|
> ||5|7054|3775|8374|0.06|
>
> As shown in the table above, redundant reasoning are rare, indicating that they do not materially affect the distribution. Since difficulty lacks precise supervision and evolves dynamically, generation length remains a practical proxy for cognitive effort. Therefore, output length is still effective for modeling difficulty, though more precise supervision is left for future work.
>
> >  **Response to Q3 (Regressor Refinement)**
>
> We thank the reviewer for this suggestion. We perform bootstrap-style refinement via fit–refine–refit. This yields consistent improvements in both regression quality and downstream performance.
>
> |Method|R2|Math-500（Pass@1/Tok）|AIME24（Pass@1/Tok）|AIME25（Pass@1/Tok）|
> |---|---|---|---|---|
> |Qwen3-4B-Thinking-2507||96.2/6749|83.3/21493|76.7/22708|
> |+DyCon|0.8008|96.2/6092|86.7/18867|76.7/21100|
> |+DyCon (optimized)|**0.9073**|**96.6**/**5567**|**86.7**/**17156**|**80.0**/**19351**|
>
> The refined regressor achieves higher R², indicating improved difficulty estimation, and leads to better performance. This suggests DyCon-generated trajectories provide higher-quality supervision. Overall, this refinement shows mutual improvement between the regressor and trajectories, which we view as a promising direction for future work.

---

> > ### Author Rebuttal · Reviewer_v3W9 · 2026-04-02
> >
> > Thanks for the authors’ response. R1 addresses my concern well, but I still have one remaining concern.
> >
> > Regarding the refinement experiment, the fact that a simple iterative fit-refine-refit procedure improves both the regressor quality and downstream performance is consistent with my concern that redundancy/noise in the original trajectories may affect difficulty estimation. It also suggests the current pipeline has clear room for improvement, since a lightweight closed-loop refinement already yields gains.
> >
> > I maintain my overall recommendation as 4.

---

> > > ### Author Response · Authors · 2026-04-02
> > >
> > > We thank the reviewer for the thoughtful follow-up.
> > >
> > > We agree that the improvement brought by the fit–refine–refit procedure suggests that the quality of reasoning trajectories plays an important role in difficulty estimation. However, we would like to clarify that this effect is not simply due to removing redundancy from the original trajectories.
> > >
> > > First, as shown in the table below, removing outlier trajectories using an IQR-based rule degrades performance, indicating that redundant reasoning is not purely harmful. Prior work suggests that abnormal samples in learning can capture more complex or atypical behaviors rather than being simple noise [1]. Consistently, retaining such trajectories helps the model better capture diverse reasoning patterns, which is beneficial to the generalization capabilities.
> > >
> > > | Method | Math-500（Pass@1/Tok） | AIME24（Pass@1/Tok） | AIME25（Pass@1/Tok） |
> > > | --- | --- | --- | --- |
> > > | Qwen3-4B-Thinking-2507 | 96.2 / 6749 | 83.3 / 21493 | 76.7 / 22708 |
> > > | +DyCon | 96.2 / 6092 | 86.7 / 18867 | 76.7 / 21100 |
> > > | +DyCon(without outlier) | 96.0 / 5929 | 80.0 / 20794 | 76.7 / 20038 |
> > >
> > > Second, refinement consistently produces more concise and structured reasoning trajectories, which aligns with prior work showing that shorter yet complete reasoning can provide better learning signals [2]. Importantly, this transformation is not equivalent to denoising, but can be understood as reshaping the distribution of reasoning trajectories toward more concise representations.
> > >
> > > Crucially, this effect is not monotonic. As shown below, while the second iteration improves both regressor quality and downstream performance, further refinement (Iteration 3) degrades performance despite higher R². This indicates that refinement is not simply removing noise. Instead, it progressively shifts the reasoning distribution away from that of the original model. Since the regressor is fitted to model the intrinsic reasoning behavior of the model, moderate refinement improves efficiency while preserving alignment. However, excessive refinement introduces a mismatch between the fitting and testing distributions: although the regressor fits the refined data better, it becomes misaligned with the model’s actual inference process, leading to inferior generalization and degraded performance.
> > >
> > > | Method | R2 | Math-500（Pass@1/Tok） | AIME24（Pass@1/Tok） | AIME25（Pass@1/Tok） |
> > > | --- | --- | --- | --- | --- |
> > > | Qwen3-4B-Thinking-2507 |  | 96.2 / 6749 | 83.3 / 21493 | 76.7 / 22708 |
> > > | +DyCon | 0.8008 | 96.2 / 6092 | 86.7 / 18867 | 76.7 / 21100 |
> > > | +DyCon (Iteration 2) | 0.9073 | 96.6 / 5567 | 86.7 / 17156 | 80.0 / 19351 |
> > > | +DyCon (Iteration 3) | 0.9276 | 96.2 / 5710 | 86.7 / 19051 | 73.3 / 19726 |
> > >
> > > Overall, these results suggest that removing noise does not necessarily improve performance, as it can enlarge the distribution gap between fitting and testing, leading to inferior generalization. Instead, refinement introduces a trade-off: moderate refinement improves efficiency while preserving the original reasoning distribution, whereas excessive refinement leads to distribution shift and degraded generalization as well. In this sense, though our method already achieves superior performance, the further iterative refinement serves as an additional enhancement option with constraints, instead of the correction of the proposed pipeline.
> > >
> > > Thank you for this valuable comment, and we will incorporate this discussion into the final version. We hope this clarification helps address the underlying concern, and we are looking forward to further discussion.
> > >
> > > [1] The Heavy-Tail Phenomenon in SGD
> > >
> > > [2] Concise Reasoning, Big Gains: Pruning Long Reasoning Trace with Difficulty-Aware Prompting

---

### Official Review · Reviewer_nWqo · 2026-03-12

**Soundness:** 2
**Presentation:** 3
**Significance:** 2
**Originality:** 2
**Overall Recommendation:** 4
**Confidence:** 4

**Summary:**

In this paper, the authors address the overthinking problem in Large Reasoning Models , where models perform redundant reflection and exploration on simple problems, leading to inefficiency. The key insight is that problem difficulty evolves dynamically during reasoning and is linearly encoded in the model's step-level embeddings. Building on this, the authors propose DyCon, a training-free framework for dynamic reasoning control.
In particular, DyCon uses remaining reasoning length as a proxy for difficulty, applying log-transform and normalization to derive a bounded difficulty target. It then fits a lightweight linear regressor to map step embeddings to difficulty estimates. During online inference, DyCon dynamically adjusts the logits of reflection-related tokens based on the estimated difficulty. It reduces the bias to encourage deeper exploration when difficulty is high, and increases the bias to promote early termination when difficulty is low. Empirically, DyCon achieves up to 40.6% token reduction and 6.7% accuracy improvement on mathematical reasoning benchmarks, with strong generalization to code, commonsense reasoning, and other non-mathematical tasks

**Compliance With Llm Reviewing Policy:**

Affirmed.

**Final Justification:**

I appreciate the authors' response on the Spearman coefficients, which resolves my previous concerns. I have decided to increase my rating to 4.

**Key Questions For Authors:**

See my comments on the weaknesses

**Limitations:**

yes

**Strengths And Weaknesses:**

Strengths:

Empirical performance is good. It achieves up to 40.6% token reduction with 6.7% accuracy improvement on mathematical reasoning.

It requires only fitting a small linear regressor on a small dataset.




Weaknesses:


One of my concern is using remaining length as a proxy for difficulty. The authors essentially use a noisy label to fit the regression model, then use this fitted model for prediction, which compounds the error. There may be hidden confounding factors. For example,  a model getting stuck on a simple step due to distraction rather than genuine difficulty, or conversely, making rapid progress on a hard problem through lucky guessing.

The method relies on a manually chosen threshold to distinguish high or low difficulty and the sensitivity analysis in Figure 5(a) suggests performance varies with this hyperparameter.


The main algorithm depends on identifying reflection-related tokens, which may not generalize to all reasoning formats or languages.

---

> ### Author Rebuttal · Authors · 2026-03-31
>
> > **Response to W1 (Noisy Difficulty Proxy)**
>
> We thank the reviewer for the instructive comments. We would like to clarify that our goal is not to rely on exact labels, but to exploit a statistically meaningful signal that correlates with difficulty. Our study indicates that the output length shows a strong correlation (Spearman’s ρ ≈ 0.62) with ground-truth difficulty on MATH, serving as a reliable proxy.
>
> Besides, to quantify the impact of confounding cases, we perform an outlier analysis based on generation length, using the mean, first quartile (q25), third quartile (q75), and interquartile range (IQR). Specifically, outliers denote the samples whose generation length lies outside the interval:
>
> $$
> [q_{25} - 1.5 \cdot \text{IQR},\; q_{75} + 1.5 \cdot \text{IQR}], \quad \text{where } \text{IQR} = q_{75} - q_{25}.
> $$
>
> |Model|Level|mean|q25|q75|outlier ratio|
> |---|---|---|---|---|---|
> |Qwen3-4B-Thinking-2507|1|2121|903|2088|0.10|
> ||2|3068|1112|4454|0.03|
> ||3|5322|1839|7155|0.03|
> ||4|6840|2930|9327|0.02|
> ||5|12025|6356|15386|0.04|
> |DeepSeek-R1-Distill-Qwen-7B|1|1932|1159|2162|0.05|
> ||2|2230|1302|2600|0.06|
> ||3|2953|1570|3149|0.10|
> ||4|3482|1721|3866|0.09|
> ||5|5580|2551|6941|0.09|
> |Qwen3-14B|1|1978|1324|2114|0.08|
> ||2|2806|1493|3022|0.09|
> ||3|3570|1987|4243|0.06|
> ||4|4610|2429|5568|0.05|
> ||5|7673|3846|9326|0.06|
> |QwQ-32B|1|1751|1191|2015|0.07|
> ||2|2297|1338|2832|0.04|
> ||3|3303|1757|4060|0.05|
> ||4|4311|2045|5196|0.07|
> ||5|7054|3775|8374|0.06|
>
> As shown in the table above, the outliers account for only a small proportion across all difficulty levels, which indicates that the overall signal remains stable at the distribution level. More fundamentally, reasoning difficulty lacks precise supervision and evolves dynamically, while existing datasets provide only static labels. Generation length thus serves as a practical proxy for dynamic cognitive effort.
>
> Therefore, the output length remains a decent proxy for modeling difficulty. Developing more precise supervision methods for dynamic difficulty assessment is a direction for future research.
>
> > **Response to W2 (Hyperparameter Sensitivity)**
>
> We would like to clarify that the performance is actually not sensitive to the threshold hyperparameter.
>
> As shown in Figure 5(a), although the curve has been visually zoomed in on the axis for better clarity, the performance difference across different threshold values is only ~0.5%, effectively enhancing token efficiency without compromising performance.
>
> Moreover, all results reported in the paper are obtained using the same fixed threshold across different models and datasets, without task-specific tuning, demonstrating that DyCon generalizes well without requiring careful hyperparameter adjustment.
>
> > **Response to W3 (Generalization across Formats and Languages)**
>
> Thank you for your instructive comments. First, regarding **generalization across reasoning formats**, current mainstream reasoning models commonly adopt the O1 [1] or R1 [2] training paradigms. As a result, these models exhibit a shared reasoning structure, characterized by step segmentation via '/n/n' and the generation of reflection-indicative tokens at the beginning of each step, as demonstrated in recent studies [3]. Our method follows this common paradigm, thereby ensuring broad compatibility with various reasoning models.
>
> Second, regarding **generalization across languages**, our method does not rely on a fixed reflection token list. Adapting our method to different languages merely requires substituting the reflection token list with an appropriate set specific to the target language. To validate this capability, we apply DyCon directly across multiple languages (MGSM) by selecting a concise set of reflection-related tokens for each language, without tuning or modifying the regressor initially fitted on English.
>
> *Qwen3-4B-Thinking-2507*
>
> |Method|English|Chinese|French|German|Japanese|
> |---|---|---|---|---|---|
> |Baseline (Pass@1/Tok)|95.6/1483|90.8/1235|89.6/2284|91.0/2470|89.6/2482|
> |DyCon (Pass@1/Tok)|96.8/1116|90.8/1053|89.8/1731|91.0/1590|89.6/1823|
>
> DyCon still maintains decent performance across languages and consistently reduces token usage, demonstrating robust cross-lingual generalization.
>
> To further understand this behavior, we assess a regressor fitted on English by evaluating its performance directly on other languages. Results indicate the English-fitted regressor accurately estimates task difficulty in non-English contexts:
>
> | |English|Chinese|
> |---|---|---|
> |Regressor Prediction|0.50|0.47|
> |Ground Truth|0.51|0.46|
>
> These findings suggest the learned difficulty estimation is language-agnostic and transfers effectively across languages, with language differences mainly reflected in superficial token mappings.
>
> [1] OpenAI o1 System Card
>
> [2] DeepSeek-R1: Incentivizing Reasoning Capability in LLMs via Reinforcement Learning
>
> [3] SEAL: Steerable Reasoning Calibration of Large Language Models for Free

---

> > ### Author Rebuttal · Reviewer_nWqo · 2026-04-04
> >
> > I appreciate the authors' detailed response. The clarification on reasoning formats and the language-specific reflection token adaptation addresses my concerns regarding the generalization of the proposed method.
> >
> > I am still concerned about the reliability of using output length as a difficulty proxy. While the authors report a Spearman’s $\rho \approx 0.62$,  this just represents a moderate correlation. Can author provide additional Spearman’s correlation results on other representative datasets？ For datasets that lack explicit ground-truth difficulty labels, maybe authors can calculate the correlation between output length and model performance （e.g., pass@1）

---

> > > ### Author Response · Authors · 2026-04-04
> > >
> > > We appreciate the reviewer's thoughtful follow-up.
> > >
> > > We agree that Spearman’s correlation coefficient (ρ ≈ 0.62) reflects a moderate correlation. This moderate correlation primarily results from the MATH dataset's use of only five discrete, manually annotated difficulty levels, which inadequately capture the actual difficulty perceived by the model. Additionally, reasoning length exhibits inherent continuity and substantial variance, complicating the achievement of high correlation with coarse discrete labels. In the manuscript, we employed these discrete difficulty levels mainly for their intuitive interpretability. However, as suggested by the reviewer, utilizing Pass@1 as a difficulty measure provides a more statistically rigorous and precise reflection of the relationship between difficulty and output length.
> > >
> > > Following this insightful suggestion, we conduct a refined empirical analysis using Pass@1 to measure problem difficulty directly. Specifically, we group samples by output length and compute average Pass@1 performance within each group.
> > >
> > > Formally, for the $k$-th length group $B_k$, we compute:
> > >
> > > $\bar{l_k} = \frac{1}{|B_k|} \sum_{i \in B_k} l_i,\quad \text{Pass@1 of } B_k = \frac{1}{|B_k|} \sum_{i \in B_k} y_i$
> > >
> > > where $l_i$ denotes output length, and $y_i$  is a binary correctness indicator, equal to 1 if correct and 0 otherwise. Subsequently, we calculate the correlation between $\bar{l}_k$ and $\text{Pass@1 of } B_k$. For the overall results, we aggregate samples from all datasets into a single combined set and compute the correlation on the merged data.
> > >
> > > | Model | $\lvert B_k \rvert$ | GPQA (Pearson/Spearman) | StrategyQA (Pearson/Spearman) | AIME2024 (Pearson/Spearman) | AIME2025 (Pearson/Spearman) | Math500 (Pearson/Spearman) | Olympiad (Pearson/Spearman) | Overall (Pearson/Spearman) |
> > > | --- | --- | --- | --- | --- | --- | --- | --- | --- |
> > > | DeepSeek-R1-Distill-Qwen-7B | 5 | -0.93/-0.90 | -0.97/-0.99 | -0.98/-0.99 | -0.93/-0.99 | -0.96/-0.86 | -0.98/-0.99 | -0.99/-0.99 |
> > > |  | 10 | -0.90/-0.89 | -0.96/-0.96 | -0.94/-0.95 | -0.90/-0.94 | -0.94/-0.84 | -0.98/-0.95 | -0.99/-0.97 |
> > > |  | 20 | -0.82/-0.81 | -0.93/-0.91 | -0.91/-0.84 | -0.82/-0.83 | -0.90/-0.81 | -0.95/-0.89 | -0.99/-0.90 |
> > > | Qwen3-4B-Thinking-2507 | 5 | -0.96/-0.99 | -0.99/-0.99 | -0.98/-0.97 | -0.95/-0.97 | -0.94/-0.83 | -0.98/-0.97 | -0.99/-0.99 |
> > > |  | 10 | -0.90/-0.94 | -0.99/-0.97 | -0.83/-0.87 | -0.85/-0.94 | -0.90/-0.81 | -0.94/-0.95 | -0.91/-0.97 |
> > > |  | 20 | -0.86/-0.88 | -0.96/-0.98 | -0.81/-0.83 | -0.82/-0.92 | -0.86/-0.81 | -0.92/-0.89 | -0.87/-0.89 |
> > >
> > > As shown in the table above, our analysis consistently shows strong negative correlations, with Pearson and Spearman coefficients reaching up to -0.99. This robust negative correlation reveals a stable and near-monotonic relationship, wherein longer reasoning outputs correspond to lower Pass@1 performance. This finding aligns with prior research indicating that increased reasoning length typically reflects higher perceived difficulty [1]. Collectively, these results strongly support using output length as a reliable proxy for model-perceived difficulty, especially when explicit difficulty annotations are unavailable.
> > >
> > > In conclusion, our findings underscore that reasoning length provides a stable and generalizable measure of model-perceived difficulty. Further detailed analyses and visualizations are available in Appendices A.2 and A.3. Given the frequent absence of explicit difficulty annotations in real-world datasets, output length emerges as a practical and robust indicator applicable across diverse domains.
> > >
> > > We thank the reviewer for this valuable comment, and we will incorporate this discussion into the final version.
> > >
> > > [1] Between Underthinking and Overthinking: An Empirical Study of Reasoning Length and Correctness in LLMs

---

### Official Review · Reviewer_ronR · 2026-03-13

**Soundness:** 3
**Presentation:** 3
**Significance:** 3
**Originality:** 3
**Overall Recommendation:** 4
**Confidence:** 4

**Summary:**

The paper addresses the issue of "overthinking" in reasoning models and proposes a training free framework that uses latent step level representation to estimate the difficulty at each step. If the estimated difficulty is low, the framework DyCon applies a soft logit bias, supressing keywords and if high the bias applied is minimal. Experiments are conducted from 4B to 32B, showing up to high token reduction in both math and non math datasets.

**Compliance With Llm Reviewing Policy:**

Affirmed.

**Final Justification:**

The authors have addressed my concerns, and the current score reflects my assessment of the paper.

**Key Questions For Authors:**

Please refer to the weaknesses

**Limitations:**

Yes

**Strengths And Weaknesses:**

### Strength
1. Training a low-cost regressor model helps identify the difficulty at each reasoning step, which is used for early termination. This enables the framework to rely on an external model for early termination.
2. Using a logit intervention instead of hard early exit, DyCon allow the model to finish its thought naturally without abruptly terminating it.
3. The framework shows good generalization to non-math datasets.

### Weakness
1. The method relies on predefined reflection tokens. This list of tokens might not be exhaustive for all the models.
2. Can the authors provide a trend analysis of how the difficulty estimation of the regressor changes across different datasets? For example, can the regressor correctly identify that the reasoning trace from AIME is more difficult than the MATH-500 dataset?

Overall, the paper has a good contribution. I have some reservations about having a regressor being trained separately, and how it will perform in a completely new domain, like a multi-turn or OOD task. Nevertheless, I push the paper towards acceptance.

---

> ### Author Rebuttal · Authors · 2026-03-31
>
> > **Response to W1 (Exhaustiveness of Reflection Tokens)**
>
> We appreciate the reviewer's insightful comment and clarify that the predefined reflection token list is not essential to our method but represents a conventional choice consistent with previous work [1]. Our approach does not rely on any specific token set.
>
> To validate this claim, we replace our initial token list with the token set proposed by SEAL [2], consistently achieving comparable performance across various models and benchmarks. These results manifest DyCon’s robustness to different reflection token choices.
>
> |Model|Math-500（Pass@1/Tok）|AIME24（Pass@1/Tok）|AIME25（Pass@1/Tok）|
> |---|---|---|---|
> |DeepSeek-R1-Distill-Qwen-7B|92.0/3955|50.0/13008|36.7/15245|
> |+DyCon|92.0/3216|53.3/10906|36.7/12415|
> |+DyCon (SEAL)|92.2/3569|56.7/11141|40.0/12775|
> |Qwen3-4B-Thinking-2507|96.2/6749|83.3/21493|76.7/22708|
> |+DyCon|96.2/6092|86.7/18867|76.7/21100|
> |+DyCon (SEAL)|96.4/5753|83.3/18056|83.3/19147|
>
> We agree that an optimal token set might exist tailored to each model. To explore this, we adopt an evolutionary strategy initialized with frequently occurring reflection-related tokens. Inspired by dynamic context learning [3], we iteratively apply token selection, mutation, and crossover to enhance model accuracy on a held-out Math validation set.
>
> |Model|Math-500（Pass@1/Tok）|AIME24（Pass@1/Tok）|AIME25（Pass@1/Tok）|
> |---|---|---|---|
> |Qwen3-4B-Thinking-2507|96.2/6749|83.3/21493|76.7/22708|
> |+DyCon (Optimized)|**96.6/5898**|**86.7/19136**|**83.3/16848**|
>
> The optimized token sets further enhance performance compared to the original list. This finding suggests that while DyCon is robust in token selection, iterative refinements can yield further improvements. Investigating principled optimization objectives and advanced selection strategies is a promising direction for future research.
>
> [1] Wait, we don’t need to "wait"! Removing thinking tokens improves reasoning efficiency
>
> [2] Seal: Steerable reasoning calibration of large language models for free
>
> [3] ThinkPilot: Steering reasoning models via automated think-prefixes optimization
>
> > **Response to W2 (Trend Analysis of Difficulty Estimation)**
>
> To address the reviewer’s concerns, we fit the regressor on the MATH dataset and evaluate it across five benchmarks spanning diverse difficulty levels. The table below summarizes mean predicted difficulty scores for each dataset using different base models. The regressor can effectively rank datasets by difficulty (i.e., Olympiad ≥ AIME > Math-500 > GSM8K), and predictions are well aligned with the ground-truth data.
>
> |Model|Type|Math-500|AIME24|AIME25|GSM8K|Olympiad|
> |---|---|---|---|---|---|---|
> |Qwen3-4B-Thinking-2507|Regressor Prediction|0.69|0.78|0.79|0.49|0.80|
> ||Ground Truth|0.69|0.78|0.80|0.51|0.81|
> |DeepSeek-R1-Distill-Qwen-7B|Regressor Prediction|0.76|0.85|0.86|0.61|0.81|
> ||Ground Truth|0.77|0.85|0.86|0.61|0.83|
> |Qwen3-14B|Regressor Prediction|0.65|0.77|0.76|0.47|0.77|
> ||Ground Truth|0.65|0.75|0.77|0.47|0.77|
> |QwQ-32B|Regressor Prediction|0.73|0.80|0.80|0.61|0.83|
> ||Ground Truth|0.73|0.82|0.82|0.61|0.86|
>
> > **Response to W3 (Domain Generalizability)**
>
> We thank the reviewer for raising this question. We evaluate the regressor fitted on the MATH dataset across several out-of-domain benchmarks, including multi-turn tasks[4].
>
> *Qwen3-4B-Thinking-2507*
>
> | |CommonsenseQA|GPQA|MultiChallenge[4]|
> |---|---|---|---|
> |Regressor Prediction|0.59|0.61|0.64|
> |Ground Truth|0.54|0.67|0.48|
>
> Our results indicate that the regressor demonstrates **reasonable generalization capability**. However, estimation errors increase compared to the in-domain setting, especially for multi-turn tasks.
>
> To further investigate this issue, We refit the regressor on a balanced dataset spanning MATH, CommonsenseQA, GPQA, and MultiChallenge, using a held-out, non-eval split. As illustrated below, it significantly improves cross-domain accuracy, aligning strongly with ground-truth difficulty scores across all evaluated domains without compromising performance on the original math benchmarks, and even generalizes effectively to an unseen domain (C4)[5]. These findings indicate that DyCon can capture a more universal notion of difficulty when fitted on diverse datasets.
>
> In summary, our experiments demonstrate that **while fitting the regressor on math data already yields transferable knowledge, incorporating diverse distributions further improves its robustness and generalization**.
>
> *Qwen3-4B-Thinking-2507（Refitted）*
>
> | |CommonsenseQA|GPQA|MultiChallenge[4]|Math-500|AIME2024|AIME2025|AMC23|GSM8K|Olympiad|C4[5]|
> |---|---|---|---|---|---|---|---|---|---|---|
> |Regressor Prediction|0.58|0.70|0.53|0.72|0.79|0.80|0.75|0.56|0.78|0.58|
> |Ground Truth|0.58|0.70|0.53|0.73|0.80|0.82|0.74|0.56|0.81|0.59|
>
> [4] MultiChallenge: a realistic multi-turn conversation evaluation benchmark challenging to frontier LLMs
>
> [5] Exploring the limits of transfer learning with a unified text-to-text transformer

---

> > ### Author Rebuttal · Reviewer_ronR · 2026-04-03
> >
> > Thanks for the rebuttal, and my concerns have been resolved.

---

> > > ### Author Response · Authors · 2026-04-04
> > >
> > > Thank you for your positive feedback and constructive  review. We will incorporate the additional analyses on reflection token robustness, cross-dataset difficulty estimation trends, and domain generalization of the regressor into the final version to further strengthen our work. We sincerely expect and appreciate your support.

---

### Decision · Program_Chairs · 2026-04-30

**Decision:**

Accept (regular)

**Comment:**

This paper aims to solve "overthinking" issues in existing reasoning models and propose a training-free framework for dynamic reasoning control. At the initial reviewing stage, all reviewers admit the motivation of the proposed method, but also raise concerns among generalization on experiments, in-depth analysis, lacks solid theoretical foundations. During the rebuttal stages, authors have presents thorough responses to handle these issues, and all reviewers have positive comments toward this paper. Therefore, I recommend acceptance for this paper.